# HESSIAN-FREE HIGH-RESOLUTION NESTEROV ACCELERATION FOR SAMPLING

## ABSTRACT

It is known (Shi et al., 2021) that Nesterov's Accelerated Gradient (NAG) for optimization differs from its continuous time limit (noiseless kinetic Langevin) when its stepsize becomes finite. This work explores the sampling counterpart of this phenonemon and proposes an accelerated-gradient-based MCMC method, based on the optimizer of NAG for strongly convex functions (NAG-SC): we reformulate NAG-SC as a Hessian-Free High-Resolution ODE, change its high-resolution coefficient to a hyperparameter, inject appropriate noise, and discretize the resulting diffusion process. Accelerated sampling enabled by the new hyperparameter is quantified and it is not a false acceleration created by time-rescaling. At continuous-time level, additional acceleration over underdamped Langevin in $W_2$ distance is proved. At discrete algorithm level, a dedicated discretization is proposed to simulate the Hessian-Free High-Resolution SDE in a cost-efficient manner. For log-strong-concave-and-smooth target measures, the proposed algorithm achieves $\tilde{\mathcal{O}}(\sqrt{d}/\epsilon)$ iteration complexity in $W_2$ distance, same as underdamped Langevin dynamics, but with a reduced constant. Empirical experiments are conducted to numerically verify our theoretical results.

## 1 INTRODUCTION

Optimization is a major machinery that drives both the theory and practice of machine learning in recent years. Since the seminal work of Nesterov (1983), acceleration has played a key role in gradient-based optimization methods. A notable example is Nesterov's Accelerated Gradient (NAG), which is an instance of a more general family of "momentum methods". NAG in fact consists of multiple methods, including NAG-C and NAG-SC, respectively for convex and strongly convex functions. Both of them provably converge faster than vanilla gradient descent (GD) in their corresponding setups (Nesterov, 1983; 2013). Newer perspectives of acceleration continue to be revealed, e.g., Su et al. (2014); Wibisono et al. (2016); Wilson et al. (2021); Hu & Lessard (2017); Attouch et al. (2018); Shi et al. (2021), many based on the interplay between continuous and discrete times. This work aims at turning NAG-SC into a sampler based on this interplay.

In fact, approaches for sampling statistical distributions, such as gradient-based Markov Chain Monte Carlo (MCMC) methods, are also of great importance in machine learning, for example due to their links to statistical inference and abilities to represent uncertainties lacking in optimization-based methods. Although not entirely the same thing, optimization and sampling are closely related: besides seeing a large class of sampling dynamics as optimization dynamics with additional noise, viewing sampling as optimization in probability space is another profound perspective that led to fruitful discoveries (e.g., Jordan et al. (1998); Liu & Wang (2016); Dalalyan (2017a); Wibisono (2018); Zhang et al. (2018); Frogner & Poggio (2020); Chizat & Bach (2018); Chen et al. (2018a); Ma et al. (2021); Erdogdu & Hosseinzadeh (2021)). In fact, an unadjusted Euler-Maruyama discretization of overdamped Langevin dynamics (abbreviated as OLD here) is commonly considered as the analog of GD in sampling (although many other discretizations are also possible), and often referred to as Unadjusted Langevin Algorithm (ULA) (Roberts et al., 1996) and/or Langevin Monte Carlo (LMC). The convergence properties of the continuous dynamics of OLD, as well as asymptotic and non-asymptotic analyses of its discretizations have been extensively studied (e.g., Roberts et al. (1996); Villani (2008); Pavliotis (2014); Dalalyan (2017b); Durmus & Moulines (2016); Dalalyan (2017a); Durmus et al. (2019a;b); Vempala & Wibisono (2019); Cheng & Bartlett (2018); Dwivedi et al. (2019); Ma et al. (2019); Chewi et al. (2021); Erdogdu & Hosseinzadeh (2021)).

Meanwhile, the notion of acceleration is less quantified in sampling compared to that in optimization, although attention has been rapidly building up. Along this direction, one line is based on diffusion processes such as underdamped Langevin dynamics (ULD). For example, the convergence and nonasymptotics of discretized ULD have been studied in Cheng et al. (2018); Dalalyan & Riou-Durand (2020); Ma et al. (2021), and were demonstrated provably faster than discretized OLD in suitable setups. These are not only great progresses but also forming perspectives complementary to the extensive studies of the convergence of continuous ULD in the mathematical community (e.g, Mattingly et al. (2002); Cao et al. (2019); Dolbeault et al. (2009; 2015); Villani (2009); Eckmann & Hairer (2003); Baudoin (2017); Eberle et al. (2019)). Another important line of research is related to accelerating particle-based approaches for optimization in probability spaces (Liu et al., 2019; Taghvaei & Mehta, 2019; Wang & Li, 2019), although we note there is no clear boundary between these two lines (e.g., Leimkuhler et al. (2018)). Additional interesting ideas also include Chen et al. (2018b); Deng et al. (2020). In general, it has been known that adding an irreversible part to the reversible dynamics of OLD[1] accelerates its convergence (e.g., Hwang et al. (2005); Lelievre et al. (2013); Ohzeki & Ichiki (2015); Rey-Bellet & Spiliopoulos (2015); Duncan et al. (2016)), and this work can be viewed to be under this umbrella. Note, though, the discretization of an accelerated continuous process is also important, and it will be analyzed.

Specifically, we propose an accelerated gradient-based MCMC algorithm termed HFHR. It is motivated by a simple question: how to appropriately inject noise to NAG algorithm **in discrete time**, so that it is turned into an algorithm for momentum-accelerated sampling? Note we don't add noise to the learning-rate$\to$ 0 limit of NAG (this has been well studied in Ma et al. (2021)), because a finite-step-size discretization of this limiting ODE may not converge as fast as NAG with the same learning rate. However, we will still use continuous dynamics as intermediate steps.

More precisely, our first step is to combine existing tools to prepare a non-asymptotic formulation for the later steps. The goal is to better account for NAG's behavior when a finite (not infinitesimal) learning rate is used. As pointed out in Shi et al. (2021), a low-resolution limiting ODE (Su et al., 2014), albeit being a milestone leading to a new venue of research (e.g, Wibisono et al. (2016)), does not fully capture the acceleration enabled by NAG — for example, it can't distinguish between NAG and another momentum method of heavy ball (Polyak, 1964). The main reason is, the low-resolution ODE describes the $h \to 0$ limit of NAG, but in practice NAG uses a finite (nonzero) $h$. High-resolution ODE was thus proposed to include additional $\mathcal{O}(h)$ terms to account for the finite $h$ effect (Shi et al., 2021). The original form of high-resolution ODE involves Hessian of the objective function, which is computationally expensive to evaluate and store for high-dimensional problems, but this difficulty can be overcome using techniques introduced in, e.g., Alvarez et al., 2002; Attouch et al., 2020, which allows us to derive a High-Resolution and Hessian-Free limiting ODE for NAG.

Then we replace the high-resolution term's coefficient in the HFHR ODE by a hyperparameter $\alpha \geq 0$, and then add noise to the resulting ODE in a specific way, which turns it into an SDE suitable for the sampling purpose. This SDE will be termed as HFHR dynamics.

To obtain an actual algorithm, the HFHR SDE is then discretized. We will see, both theoretically and empirically, that nonzero $\alpha$ can lead to accelerated convergence of the sampling algorithm; this acceleration is **not** an artificial consequence of time-rescaling, which would not give acceleration after discretization with an appropriate step size. Meanwhile, note our discretization is just one of the many possible schemes. It was known that high-order discretizations can improve statistical accuracy and even the speed of convergence (see e.g., Chen et al. (2015); Li et al. (2019); Shen & Lee (2019)), although such improvements often come with more computations per iteration. The discretization considered here is just a simple first-order scheme that uses one (full-)gradient evaluation per step, but it better utilizes the structure of HFHR dynamics than Euler-Maruyama.

Our presentation will be structured as follows. After detailing the construction of HFHR, we will analyze its convergence, at both the continuous level (HFHR dynamics) and the discrete level (HFHR algorithm). For precise theoretical results, we will consider the setup of log-strongly-concave target distributions, which are commonly considered in the literature (Kim et al., 2016; Bubeck et al., 2018; Dalalyan, 2017b; Dalalyan & Riou-Durand, 2020; Dwivedi et al., 2019; Shen & Lee, 2019). The additional acceleration of HFHR when compared to ULD in continuous time will be demonstrated explicitly in Thm.5.1. For our discretized HFHR algorithm, a non-asymptotic error bound will

---

[1]For irreversible-acceleration *not* from OLD, see e.g., Bierkens et al. (2019); Bouchard-Côté et al. (2018).

be obtained (Thm.5.2), which confirms that the additional acceleration in continuous time carries through to the discrete territory. Finally, numerical experiments are provided, verifying the validity and tightness of our theoretical results, and empirically showing HFHR remains advantageous for the nonconvex and high-dimensional example of Bayesian Neural Networks.

The main contribution of this article is the idea of turning NAG-SC optimization algorithm into a sampler, which also introduces a new dynamics that is neither overdamped or underdamped Langevin. Nevertheless, theoretical analyses (e.g., Thm.5.2, Cor.5.4 & Rmk.5.5) and numerical experiments (Sec.6) are also provided to quantify the effectiveness of this idea.

## 2 BACKGROUND: LANGEVIN DYNAMICS FOR SAMPLING

Consider sampling from Gibbs measure $\mu$ whose density is $d\mu = \frac{1}{\int e^{-f(\boldsymbol{y})}d\boldsymbol{y}}e^{-f(\boldsymbol{x})}d\boldsymbol{x}$, where $f :$ $\mathbb{R}^d \mapsto \mathbb{R}$ will be called the potential function. Two diffusion processes popular for sampling (and modeling important physical processes too) are named after Langevin. One is overdamped Langevin dynamics (OLD), and the other is kinetic Langevin dynamics (abbreviated as ULD to comply with a convention of calling it underdamped Langevin). They are respectively given by

$$\text{(OLD)} \quad d\boldsymbol{q}_t = -\nabla f(\boldsymbol{q}_t)dt + \sqrt{2}d\boldsymbol{W}_t \qquad \text{(ULD)} \quad \begin{cases} d\boldsymbol{q}_t = \boldsymbol{p}_t dt \\ d\boldsymbol{p}_t = -\gamma\boldsymbol{p}_t dt - \nabla f(\boldsymbol{q}_t)dt + \sqrt{2\gamma}d\boldsymbol{B}_t \end{cases}$$

where $\boldsymbol{q}_t, \boldsymbol{p}_t \in \mathbb{R}^d$, $\boldsymbol{W}_t, \boldsymbol{B}_t$ are i.i.d. Wiener processes in $\mathbb{R}^d$, and $\gamma > 0$ is a friction coefficient. Under mild conditions (e.g., Pavliotis (2014)), OLD converges to $\mu$ and ULD converges to $d\pi(\boldsymbol{q}, \boldsymbol{p}) = d\mu(\boldsymbol{q})\nu(\boldsymbol{p})d\boldsymbol{p}$, where $\nu(\boldsymbol{p}) = (2\pi)^{-\frac{d}{2}}e^{-\|\boldsymbol{p}\|^2/2}$ , so its $\boldsymbol{q}$ marginal follows $\mu$.

OLD and ULD are closely related. In fact, OLD is the $\gamma \to \infty$ overdamping limit of ULD after time dilation (e.g., Pavliotis (2014)). However, OLD is a reversible Markov process but ULD is irreversible, and thus both their equilibrium and non-equilibrium statistical mechanics are different, although closely related too. We will only focus on the convergence to statistical equilibrium (see e.g., Souza & Tao (2019) for non-equilibrium aspects).

Many celebrated approaches exist for establishing the exponential convergence (a.k.a. geometric ergodicity) of OLD, including the seminal work of Roberts et al. (1996), the ones using spectral gap (e.g., Dalalyan, 2017b, Lemma 1), synchronous coupling (Villani, 2008, p33-35)(Durmus et al., 2019b, Proposition 1), functional inequalities such as Poincaré's inequality (Pavliotis, 2014, Theorem 4.4) and log Sobolev inequality (Vempala & Wibisono, 2019, Theorem 1). There are also fruitful results for ULD, including the ones leveraging Lyapunov function (Mattingly et al., 2002, Theorem 3.2), hypocoercivity (Villani, 2009; Dolbeault et al., 2009; 2015; Roussel & Stoltz, 2018), coupling (Cheng et al., 2018, Theorem 5)(Dalalyan & Riou-Durand, 2020, Theorem 1)(Eberle et al., 2019, Theorem 2.3), LSI (Ma et al., 2021, Section 3.1), modified Poincaré's inequality (Cao et al., 2019, Theorem 1), and spectral analysis (Kozlov, 1989; Eckmann & Hairer, 2003).

The study of asymptotic convergence of discretized OLD dates back to at least the 1990s (Meyn et al., 1994; Roberts et al., 1996). The non-asymptotic analysis of LMC discretization of OLD can be found in Dalalyan (2017b) and it shows the discretization achieves $\epsilon$ error, in TV distance, in $\tilde{\mathcal{O}}(d/\epsilon^2)$ steps. Subsequent results include $\tilde{\mathcal{O}}(d/\epsilon^2)$ in $W_2$ (Durmus & Moulines, 2016), $\tilde{\mathcal{O}}(d/\epsilon)$ in KL (Cheng & Bartlett, 2018), $\tilde{\mathcal{O}}(d/\epsilon)$ in $W_2$ under additional 3rd-order regularity (Durmus et al., 2019b), and $\tilde{\mathcal{O}}(\sqrt{d}/\epsilon)$ in $W_2$ under additional 3rd-order regularity (Li et al., 2021). For discretized ULD, one has $\tilde{\mathcal{O}}(\sqrt{d}/\epsilon)$ iteration complexity in $W_2$ (Cheng et al., 2018; Dalalyan & Riou-Durand, 2020) and $\tilde{\mathcal{O}}(\sqrt{d}/\sqrt{\epsilon})$ in KL (Ma et al., 2021). ULD is still generally conceived to be advantageous over OLD and sometimes understood as its momentum-accelerated version.

## 3 NOTATIONS AND CONDITIONS

We will use 2-Wasserstein distance to quantify convergence, i.e. $W_2(\mu_1, \mu_2) = \left(\inf_{\pi \in \Pi(\mu_1, \mu_2)} \mathbb{E}_{(\boldsymbol{X},\boldsymbol{Y})\sim\pi}\|\boldsymbol{X} - \boldsymbol{Y}\|^2\right)^{\frac{1}{2}}$ where $\Pi(\mu_1, \mu_2)$ is the set of all couplings of $\mu_1$ and $\mu_2$.

Assume WLOG that $\boldsymbol{0} \in \arg\min_{\boldsymbol{x}\in\mathbb{R}^d} f(\boldsymbol{x})$. The following condition will also be frequently used.

**A 1.** *(**Standard Strong-Convexity and Smoothness Condition**) Function $f \in \mathcal{C}^1(\mathbb{R}^d) : \mathbb{R}^d \mapsto \mathbb{R}$ is m-stronly-convex and L-smooth, if there exist constants $m, L > 0$ such that $\forall \boldsymbol{x}, \boldsymbol{y} \in \mathbb{R}^d$, we have*

$$\|\nabla f(\boldsymbol{y}) - \nabla f(\boldsymbol{x})\| \leq L\|\boldsymbol{y} - \boldsymbol{x}\| \text{ and } f(\boldsymbol{y}) \geq f(\boldsymbol{x}) + \langle \nabla f(\boldsymbol{x}), \boldsymbol{y} - \boldsymbol{x} \rangle + \frac{m}{2}\|\boldsymbol{y} - \boldsymbol{x}\|^2$$

For $f \in \mathcal{C}^2$, this condition is equivalent to $mI \preceq \nabla^2 f \preceq LI$.

## 4 THE CONSTRUCTION OF HFHR DYNAMICS

HFHR is obtained by formulating NAG-SC as a Hessian free high-resolution ODE, lifting the high-resolution term's coefficient as a free parameter, and adding appropriate noises.

More precisely, let's start with NAG-SC algorithm:

$$\boldsymbol{x}_{k+1} = \boldsymbol{y}_k - s\nabla f(\boldsymbol{y}_k) \tag{1}$$
$$\boldsymbol{y}_{k+1} = \boldsymbol{x}_{k+1} + c(\boldsymbol{x}_{k+1} - \boldsymbol{x}_k) \tag{2}$$

where $s$ is the learning rate (also known as step size), and $c = \frac{1-\sqrt{ms}}{1+\sqrt{ms}}$ is a constant based on $s$ and the strong convexity coefficient $m$ of $f$; the method also works for non-strongly-convex $f$ though.

A high-resolution ODE description of Eq.(1) & (2) is obtained in (Shi et al., 2021, Section 2)

$$\ddot{\boldsymbol{y}} + \sqrt{s}\left(\frac{2(1-c)}{s(1+c)} + \nabla^2 f(\boldsymbol{y})\right)\dot{\boldsymbol{y}} + \frac{2}{1+c}\nabla f(\boldsymbol{y}) = \boldsymbol{0}, \tag{3}$$

which can better account for the effect of non-infinitesimal $s$ than the $s \to 0$ limit (note $c$ depends on $s$). However, in this original form, Hessian of $f$ is involved, which is expensive to compute and store especially for high-dimensional problems.

To obtain a Hessian-free high-resolution ODE description of Equation (1) and (2), we first turn the iteration into a 'mechanical' version by introducing position variable $\boldsymbol{q}_k = \boldsymbol{y}_k$ and momentum variable $\boldsymbol{p}_k = (\boldsymbol{y}_k - \boldsymbol{x}_k)/h$. Replacing $\boldsymbol{x}_{k+1}$ in (1) and the first $\boldsymbol{x}_{k+1}$ in (2) by $\boldsymbol{q}_{k+1}$ and $\boldsymbol{p}_{k+1}$, the second $\boldsymbol{x}_{k+1}$ in (2) by $\boldsymbol{q}_k - s\nabla f(\boldsymbol{q}_k)$, and the $\boldsymbol{x}_k$ in (2) by $\boldsymbol{q}_k$ and $\boldsymbol{p}_k$, we obtain

$$\begin{cases} \boldsymbol{q}_{k+1} = \boldsymbol{q}_k + h\boldsymbol{p}_{k+1} - s\nabla f(\boldsymbol{q}_k) \\ \boldsymbol{p}_{k+1} = c\boldsymbol{p}_k - c\frac{s}{h}\nabla f(\boldsymbol{q}_k) \end{cases}$$

Now, choose $\gamma$, $\alpha$ and $h$ as $h = \sqrt{cs}, \gamma = \frac{1-c}{h}, \alpha = \frac{s}{h}$. It is easy to see that $\gamma > 0$, $\alpha > 0$, then NAG-SC exactly rewrites as

$$\begin{cases} \boldsymbol{q}_{k+1} = \boldsymbol{q}_k + h\boldsymbol{p}_{k+1} - h\alpha\nabla f(\boldsymbol{q}_k) \\ \boldsymbol{p}_{k+1} = \boldsymbol{p}_k - h\gamma\boldsymbol{p}_k - h\nabla f(\boldsymbol{q}_k) \end{cases}. \tag{4}$$

Note the technique for bypassing the Hessian without introducing any approximation is already well studied in the literature (e.g., Alvarez et al. (2002); Attouch et al. (2020)).

So far, both $h$ and $\alpha$ are actually determined by the hyperparameter $s$ of NAG-SC. However, if we now consider $\alpha$ as an independent variable (i.e., 'lift' it) and let $h \to 0$, we see (4) is a 1st-order discretization (with step size $h$) of the dynamics

$$\begin{cases} \dot{\boldsymbol{q}} = \boldsymbol{p} - \alpha\nabla f(\boldsymbol{q}) \\ \dot{\boldsymbol{p}} = -\gamma\boldsymbol{p} - \nabla f(\boldsymbol{q}) \end{cases}. \tag{5}$$

Note $\alpha$, if inherited from NAG-SC, should be $\alpha = \sqrt{s/c} = \mathcal{O}(h)$, which, in a low-resolution ODE will be discarded, and this eventually leads to ULD rather than HFHR. However, we now allow it to be a free parameter and will see that $\alpha \neq \mathcal{O}(h)$ can be advantageous.

Before quantifying these advantages later on, we finish the construction by appropriately injecting Gaussian noises to Equation (5). This is just like how OLD can be obtained by adding noise to the

gradient flow. The right amount and structure of noise turn the ODE into a Markov process that can serve the purpose of sampling, and the detailed form of our noise is given by:

$$
\begin{cases}
d\boldsymbol{q}_t = (\boldsymbol{p}_t - \alpha \nabla f(\boldsymbol{q}_t))dt + \sqrt{2\alpha}d\boldsymbol{W}_t \\
d\boldsymbol{p}_t = (-\gamma \boldsymbol{p}_t - \nabla f(\boldsymbol{q}_t))dt + \sqrt{2\gamma}d\boldsymbol{B}_t
\end{cases}. \tag{6}
$$

Here $\alpha \geq 0, \gamma > 0$ are constant parameters, and $\boldsymbol{W}_t, \boldsymbol{B}_t$ are independent standard Brownian motions in $\mathbb{R}^d$. This irreversible process will be named as **Hessian-Free High-Resolution(HFHR)** dynamics. We write it as HFHR$(\alpha, \gamma)$ to emphasize the dependence on $\alpha$ and $\gamma$ when needed.

Substitution into Fokker-Planck PDE shows HFHR dynamics is unbiased (proof in Appendix B.1):

**Theorem 4.1.** $\pi$ *is the invariant distribution of HFHR described in Eq.(6), just like ULD.*

## 5 THEORETICAL ANALYSIS OF THE HFHR DYNAMICS/ALGORITHM

### 5.1 HFHR DYNAMICS IN CONTINUOUS TIME

Let's establish the exponential convergence of HFHR dynamics and its additional acceleration when compared to ULD, when the target measure has a strongly-convex and smooth potential.

**Theorem 5.1.** *Assume Conditions A1 holds and further assume $\gamma^2 > L + m$ and $\alpha \leq \frac{\gamma^2 - L - m}{m\gamma}$. Denote the law of $\boldsymbol{q}_t$ by $\mu_t$. Then there exists $\kappa' > 0$ depending only on $\alpha$ and $\gamma$, such that*

$$
W_2(\mu_t, \mu) \leq \kappa' e^{-(\frac{m}{\gamma} + m\alpha)t} W_2(\mu_0, \mu).
$$

*Detailed expression of $\kappa'$ can be found in Appendix A.*

Thm. 5.1 state that HFHR converges to the target distribution exponentially fast in log-strongly-concave and smooth setup. For ULD, Dalalyan & Riou-Durand (2020, Theorem 1) obtained exponential convergence result in 2-Wasserstein distance with rate $\frac{\sqrt{m}}{\sqrt{\kappa} + \sqrt{\kappa - 1}}$ using a simple and elegant coupling approach, and showed this rate is optimal as it is achieved by the bivariate function $f(x, y) = \frac{m}{2}x^2 + \frac{L}{2}y^2$. In Thm 5.1, we use the same coupling approach to obtain an (asymptotically) equivalent rate $\frac{\sqrt{m}}{2\sqrt{\kappa}}$. Since ULD is HFHR$(0, \gamma)$ and our bound agrees with existing result when $\alpha = 0$ and shows faster convergence for $\alpha > 0$, the acceleration of HFHR in continuous time is evidenced. For example, if we set $\gamma = 2\sqrt{L}$ and push $\alpha$ to the upper bound specified in Thm. 5.1, we obtain an $\mathcal{O}(\sqrt{L})$ rate in the log-strongly-concave setup. Compared with the rate in Dalalyan & Riou-Durand (2020), this is a speed-up of order $\kappa$.

### 5.2 HFHR ALGORITHM IN DISCRETE TIME

To obtain an implementable method, we now discretize the time of HFHR dynamics. As our main goal is to show the acceleration enabled by $\alpha$ won't disappear after discretization (unlike a fake acceleration due to time rescaling), we'll just analyze a 1st-order discretization (but a high-accuracy discretization adapted from RMA (Shen & Lee, 2019) will also be provided, in Appendix F).

For simplicity, we'll work with constant step size $h$. Inspired by Strang splitting for differential equations (Strang, 1968; McLachlan & Quispel, 2002), we consider a symmetric composition for updating over each time interval $[kh, (k+1)h]$: $\boldsymbol{x}_{k+1} := \phi^{\frac{h}{2}} \circ \psi^h \circ \phi^{\frac{h}{2}}(\boldsymbol{x}_k)$ where $\boldsymbol{x}_k = \begin{bmatrix} \boldsymbol{q}_{kh} \\ \boldsymbol{p}_{kh} \end{bmatrix}$, $\phi$ and $\psi$ correspond to solution flows of split SDEs, respectively given by

$$
\phi : \begin{cases} d\boldsymbol{q} = \boldsymbol{p}dt \\ d\boldsymbol{p} = -\gamma \boldsymbol{p}dt + \sqrt{2\gamma}d\boldsymbol{B} \end{cases} \qquad \psi : \begin{cases} d\boldsymbol{q} = -\alpha \nabla f(\boldsymbol{q})dt + \sqrt{2\alpha}d\boldsymbol{W} \\ d\boldsymbol{p} = -\nabla f(\boldsymbol{q})dt \end{cases},
$$

and $\phi^t(\boldsymbol{x}_0)$ and $\psi^t(\boldsymbol{x}_0)$ mean $x$'s value after evolving $\phi$ and $\psi$ for $t$ time with initial condition $\boldsymbol{x}_0$.

Note that $\phi$ flow can be solved explicitly since the second equation is an Ornstein-Unlenbeck process and integrating the second equation followed by integrating the first one gives us an explicit solution

$$\begin{cases} \boldsymbol{q}_t = \boldsymbol{q}_0 + \frac{1-e^{-\gamma t}}{\gamma}\boldsymbol{p}_0 + \sqrt{2\gamma}\int_0^t \frac{1-e^{-\gamma(t-s)}}{\gamma}d\boldsymbol{B}(s), \\ \boldsymbol{p}_t = e^{-\gamma t}\boldsymbol{p}_0 + \sqrt{2\gamma}\int_0^t e^{-\gamma(t-s)}d\boldsymbol{B}(s). \end{cases} \tag{7}$$

For an implementation of the stochastic integral part in Equation 7, denoting $\boldsymbol{X} = \sqrt{2\gamma}\int_0^t \frac{1-e^{-\gamma(t-s)}}{\gamma}d\boldsymbol{B}(s)$ and $\boldsymbol{Y} = \sqrt{2\gamma}\int_0^t e^{-\gamma(t-s)}d\boldsymbol{B}(s)$, and the covariance matrix of $(\boldsymbol{X}, \boldsymbol{Y})$

is $\mathrm{Cov}(\boldsymbol{X}, \boldsymbol{Y}) = \begin{bmatrix} \frac{\gamma h + 4e^{-\gamma\frac{h}{2}} - e^{-\gamma h} - 3}{\gamma^2}I_d & \frac{(1-e^{-\gamma\frac{h}{2}})^2}{\gamma}I_d \\ \frac{(1-e^{-\gamma\frac{h}{2}})^2}{\gamma}I_d & (1-e^{-\gamma h})I_d \end{bmatrix}$. As mean and covariance fully determine

a Gaussian distribution, $\begin{bmatrix} \boldsymbol{X} \\ \boldsymbol{Y} \end{bmatrix} = M\boldsymbol{\xi}$ where $M$ is the Cholesky decomposition of $\mathrm{Cov}(\boldsymbol{X}, \boldsymbol{Y})$, $\boldsymbol{\xi}$ is

a $2d$ standard Gaussian random vector, i.i.d. at each step, and $\phi^t$ can thus be exactly simulated.

However, $\psi$ flow is generally not explicitly solvable unless $f$ is a quadratic function in $\boldsymbol{q}$. We simply choose to approximate $\psi^h(\boldsymbol{x}_0)$ with one-step Euler-Maruyama integration $\psi^h(\boldsymbol{x}_0) \approx \widetilde{\psi}^h(\boldsymbol{x}_0)$ given by $\begin{cases} \boldsymbol{q}_h = \boldsymbol{q}_0 - \alpha\nabla f(\boldsymbol{q}_0)h + \sqrt{2\alpha h}\boldsymbol{\eta} \\ \boldsymbol{p}_h = \boldsymbol{p}_0 - \nabla f(\boldsymbol{q}_0)h \end{cases}$ where $\boldsymbol{\eta}$ is a standard $d$-dimensional Gaussian random vector, again i.i.d. each time $\widetilde{\psi}$ is called.

Altogether, one step of an implementable Strang's splitting of HFHR is hence $\phi^{\frac{h}{2}} \circ \widetilde{\psi}^h \circ \phi^{\frac{h}{2}}$ and we call this numerical scheme the HFHR algorithm, which is summarized in Algorithm 1.

---

**Algorithm 1** HFHR Algorithm

---

1: **Input**: potential function $f$ and its gradient $\nabla f$, damping coefficients $\alpha$ and $\gamma$, step size $h$, initial condition $(\boldsymbol{q}_0, \boldsymbol{p}_0)$

2: **procedure** DISCRETIZED HFHR$(f, \nabla f, \alpha, \gamma, h, \boldsymbol{q}_0, \boldsymbol{p}_0)$

3:     $k = 0$ and initialize $\begin{bmatrix} \boldsymbol{q}_0 \\ \boldsymbol{p}_0 \end{bmatrix}$

4:     **while** not converge **do**

5:         Generate independent standard Gaussian random vectors $\boldsymbol{\eta}_{k+1} \in \mathbb{R}^d, \boldsymbol{\xi}_{k+1}^1, \boldsymbol{\xi}_{k+1}^2 \in \mathbb{R}^{2d}$

6:         Run $\phi^{\frac{h}{2}}$ : $\begin{bmatrix} \boldsymbol{q}_1 \\ \boldsymbol{p}_1 \end{bmatrix} = \begin{bmatrix} \boldsymbol{q}_{kh} + \frac{1-e^{-\gamma\frac{h}{2}}}{\gamma}\boldsymbol{p}_{kh} \\ e^{-\gamma\frac{h}{2}}\boldsymbol{p}_{kh} \end{bmatrix} + M\boldsymbol{\xi}_{k+1}^1$

7:         Run $\widetilde{\psi}^h$ : $\begin{bmatrix} \boldsymbol{q}_2 \\ \boldsymbol{p}_2 \end{bmatrix} = \begin{bmatrix} \boldsymbol{q}_1 - \alpha\nabla f(\boldsymbol{q}_1)h + \sqrt{2\alpha h}\boldsymbol{\eta}_{k+1} \\ \boldsymbol{p}_1 - \nabla f(\boldsymbol{q}_1)h \end{bmatrix}$

8:         Run $\phi^{\frac{h}{2}}$ : $\begin{bmatrix} \boldsymbol{q}_3 \\ \boldsymbol{p}_3 \end{bmatrix} = \begin{bmatrix} \boldsymbol{q}_2 + \frac{1-e^{-\gamma\frac{h}{2}}}{\gamma}\boldsymbol{p}_2 \\ e^{-\gamma\frac{h}{2}}\boldsymbol{p}_2 \end{bmatrix} + M\boldsymbol{\xi}_{k+1}^2$

9:         $\begin{bmatrix} \boldsymbol{q}_{(k+1)h} \\ \boldsymbol{p}_{(k+1)h} \end{bmatrix} \leftarrow \begin{bmatrix} \boldsymbol{q}_3 \\ \boldsymbol{p}_3 \end{bmatrix}$

10:         $k \leftarrow k + 1$

11:     **end while**

12: **end procedure**

---

As $\psi$ in Strang splitting is replaced by a 1st-order approximation $\widetilde{\psi}$, the method is of order 1, however with good constant. This is rigorously established by the following theorem (interested readers are referred to Appendix.D.5-D.7 and Li et al. (2021) for more technical details):

**Theorem 5.2.** *Under Assumption 1, we further assume $\gamma - \frac{L+m}{\gamma} \geq m\alpha$ and the function $\nabla\Delta f$ satisfies a third-order growth condition, i.e., $\left\|\nabla\Delta f(\boldsymbol{q})\right\| \leq G\sqrt{1 + \|\boldsymbol{q}\|^2}, \forall \boldsymbol{q} \in \mathbb{R}^d$ for some $G > 0$. If $(\boldsymbol{q}_0, \boldsymbol{p}_0) \sim \pi_0$, then there exists $h_0, C > 0$ such that when $0 < h < h_0$, we have*

$$W_2(\mu_k, \mu) \leq \sqrt{2}\kappa' e^{-(\frac{m}{\gamma}+m\alpha)kh}W_2(\pi_0, \pi) + \sqrt{2}Ch \tag{8}$$

*where $\kappa'$ is a constant depending only on $L, m, \gamma, \alpha$ (see Appendix A for detail), $\mu_k$ is the law of the $q$ marginal of the $k$-th iterate in Algorithm 1, and $\mu$ is the $q$ marginal of the invariant distribution $\pi$.*

*In particular, $C = \mathcal{O}(\sqrt{d})$ and there exists $b > 0$, independent of $\alpha$ and is of order $\mathcal{O}(\sqrt{d})$, s.t.*

$$C \leq \frac{b}{m}(\alpha^2 - \frac{\alpha}{\gamma} + \frac{1}{\gamma^2}). \tag{9}$$

**Remark 5.3.** *The linear growth (at infinity) condition on $\nabla \Delta f$ is actually not as restrictive as it appears. For example, for monomial potentials, i.e., $f(x) = x^p, p \in \mathbb{Z}_+$, our linear growth condition is met when $p \leq 4$, whereas a standard condition (Pavliotis, 2014, Theorem 3.1) for the existence of SDE solutions holds only when $p \leq 2$. In addition, our condition is related to the Hessian Lipschitz condition commonly used in the literature (e.g., Durmus et al. (2019b); Ma et al. (2021)). Smoothness and Hessian Lipschitzness imply our condition. Meanwhile, examples that satisfy linear growth condition but are not Hessian Lipschitz exist, e.g., $f(x) = x^4$, and thus linear growth condition is not necessarily stronger than Hessian Lipschitzness.*

Inspecting the role of $\alpha$ in Equation (8), we see that $\alpha$ clearly increases the rate of exponential decay, but at the same time it can also increase the discretization error (see (9); assuming $h$ is fixed). However, as the following Corollary 5.4 and its remark will show, the net effect of having a positive $\alpha > 0$, at least for some $\alpha^\star$, is reduced iteration complexity.

**Corollary 5.4.** *Consider the same assumption as in Thm. 5.2. If $(\boldsymbol{q}_0, \boldsymbol{p}_0) \sim \pi_0$, then there exists $h_0, C > 0$ (same as that in Theorem 5.2; recall $C = \mathcal{O}(\sqrt{d})$) such that for any target error tolerance $\epsilon > 0$, if we choose $h = h^\star \triangleq \min\{h_0, \frac{\epsilon}{2\sqrt{2}C}\}$, then for $\epsilon < 2\sqrt{2}Ch_0$, after*

$$k^\star = 2\sqrt{2} \frac{C}{\frac{m}{\gamma} + m\alpha} \frac{1}{\epsilon} \log \frac{2\sqrt{2}\kappa' W_2(\pi_0, \pi)}{\epsilon} = \tilde{\mathcal{O}}\left(\frac{\sqrt{d}}{\epsilon}\right). \tag{10}$$

*steps, we have $W_2(\mu_k, \mu) \leq \epsilon$.*

**Remark 5.5.** *Recall from Thm.5.2 that $C \leq \frac{b}{m}(\alpha^2 - \frac{\alpha}{\gamma} + \frac{1}{\gamma^2})$, so if we consider the minimizer $\alpha^\star$ of an upper bound of $\frac{C}{\frac{m}{\gamma} + m\alpha}$, $\alpha^\star = \arg\min_{\alpha \geq 0} \frac{b}{m^2} \frac{\alpha^2 - \frac{\alpha}{\gamma} + \frac{1}{\gamma^2}}{\frac{1}{\gamma} + \alpha} = \frac{\sqrt{3}-1}{\gamma}$. This suggests that by choosing an optimal $\alpha > 0$, one could effectively reduce iteration complexity. Note, however, that this $\alpha^\star$ may not be the true optimal one as bounds may not be tight. If they were, $k^\star_{\alpha^\star} = (2\sqrt{3} - 3)k^\star_{\alpha=0} \approx 0.46 k^\star_{\alpha=0}$; i.e., steps needed by ULD (discretized by Alg.1 with $\alpha = 0$) can be halved by HFHR.*

Rmk.5.5 shows HFHR algorithm can lead to a similar bound on iteration complexity as ULD algorithm but with an improved constant, and thus is a more efficient algorithm. This improvement also shows that the acceleration of HFHR can be carried through from continuous time to discrete time. The same conclusion has been consistently observed in numerical experiments too.

**Remark 5.6.** *Readers interested in more explicit condition number dependence are referred to Appendix E, where we show, for 2D Gaussian target with condition number $\kappa \gg 1$, the convergences of Euler discretization of ULD under optimal parameters and HFHR under suboptimal parameters are, respectively, given by $(1 - 1/\kappa + o(1/\kappa))^n$ and $(1 - 2/\kappa + o(1/\kappa))^n$, where $n$ is the number of iterations. The latter (HFHR) is faster despite that its hyperparameters may not be optimal.*

## 6 NUMERICAL EXPERIMENTS

We now empirically study the acceleration enabled by $\alpha \neq 0$ by comparing HFHR algorithm and the popular KLMC discretization of ULD (Dalalyan & Riou-Durand, 2020). For fairness, discretizations of the same order and cost are compared (Appendix F has an additional comparison).

### 6.1 VERIFICATION OF THEORETICAL RESULTS IN SECTION 5.2

This subsection numerically verifies the $d$ and $h$ dependence of HFHR algorithm as well as the genuine acceleration enabled by $\alpha$. For this purpose, we will *not* use Gaussian targets, because otherwise HFHR will decouple across different (orthogonal) dimensions, and hence its discretization error having a $\mathcal{O}(\sqrt{d})$ dependence would just be a consequence of using 2-Wasserstein distance for quantifying statistical accuracy. To inspect a more interesting example, we consider a potential that is no longer additive across different dimensions, namely $f(\boldsymbol{x}) = \log(e^{x_1} + \cdots + e^{x_d}) + \frac{1}{2}\|\boldsymbol{x}\|^2$.

It's not hard to see that all dimensions will be coupled in the HFHR dynamics (and ULD too). Moreover, the new potential $f$ is still a strongly convex function and satisfies the assumption in Theorem 5.2. When the target measure is non-Gaussian, we no longer have a closed form expression for 2-Wasserstein distance and it is computationally expensive to approximate the 2-Wasserstein distance by samples. Therefore, we use the error of mean instead as a surrogate because $\left\|\mathbb{E}_{\mu_k}\boldsymbol{q} - \mathbb{E}_{\mu}\boldsymbol{q}\right\| \leq W_2(\mu_k, \mu)$ and hence the bound in Equation (8) also applies to the error in mean, and so does the iteration complexity bound in Equation (10).

Theorem 5.2 says the final sampling error is upper bounded the discretization error that is linear in $h$ and $\sqrt{d}$. To numerically verify the linear dependence on $h$, we work with $d = 2$ and ran ULD algorithm for sufficiently long time with tiny step size ($h = 0.0005$) to obtain $10^8$ independent realizations and use them (as benchmark) to empirically estimate $\mathbb{E}_{\mu}\boldsymbol{q}$. We then set $\gamma = 2, \alpha = 1$ and use Algorithm 1, with $h \in \{2^k | -7 \leq k \leq 0\}$. For each $h$, we run $\frac{T}{h}$ (with $T = 50$) iterations in Algorithm 1 to ensure the Markov chains are well-mixed and the contribution to final error from exponential decay is order-of-magnitude smaller than discretization error. The results are plotted in Figure 1a. The observed linear dependence on the step size $h$ is consistent with Theorem 5.2.

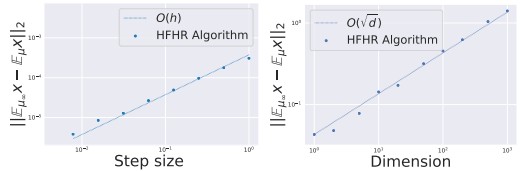
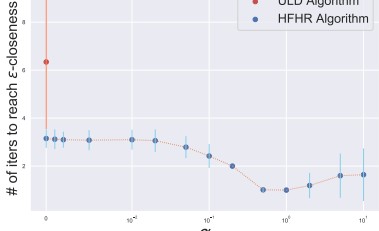

(a) Linear dependence of discretization error of Algorithm 1 on $h$

(b) Linear dependence of discretization error of Algorithm 1 on $\sqrt{d}$

Figure 1: Illustration of the consistency between the theoretical bound in Theorem 5.2 and experiment results.

Figure 2: Improvement of Algorithm 1 over ULD algorithm in iteration complexity. (vertical bar = 1 std.)

To numerically verify the $\mathcal{O}(\sqrt{d})$ dependence, we extensively experiment with $d \in \{1, 2, 5, 10, 20, 50, 100, 200, 500, 1000\}$. For each $d$, we run 1,000 independent realizations of ULD algorithm until well converged with tiny step size ($h = 0.005$) and use their empirical average as the 'true' mean. Then we fix $\gamma = 2, \alpha = 1, h = 0.1, T = 10$ and for each $d$, we run 1,000 independent realizations of HFHR algorithm for $\frac{T}{h} = 100$ iterations. Experiment results are plotted in Figure 1b and the linear trend demonstrates that the bound in Thm. 5.2 is tight in terms of $d$-dependence.

The final experiment compares Algorithm 1 with ULD algorithm in terms of iteration complexity. The goal is to demonstrate the genuine acceleration of HFHR is not an artifact due to time rescaling, which would disappear after discretization as the stability limit changes accordingly. To do so, we push both ULD and HFHR to their respective largest $h$ values that still allow monotonic convergence at a large scale, and compare their mixing times. For general nonlinear problems like the one here, Remark 5.5 suggests that with appropriately chosen $\alpha$, HFHR algorithm effectively reduces the constant factor of the iteration complexity, implying accelerated sampling. To just provide one empirical verification of this improvement, we choose $d = 10$ and use the error of mean $\left\|\mathbb{E}_{\mu_k}\boldsymbol{q} - \mathbb{E}_{\mu}\boldsymbol{q}\right\|$ to measure sampling accuracy. The benchmark, i.e., $\mathbb{E}_{\mu}\boldsymbol{q}$, is again obtained from 1,000 independent realizations of ULD algorithm with tiny step size ($h = 0.005$), ran for long enough to ensure the corresponding Markov chain is well-mixed. The initial measure is chosen as a Dirac measure at $(100 \times \mathbf{1}_d, \mathbf{0}_d)$, where $\mathbf{1}_d, \mathbf{0}_d$ are $d$-dimensional vectors filled with 1 and 0 respectively. We pick threshold $\epsilon = 0.1$, and for each $\alpha \in \{0, 0.001, 0.002, 0.005, 0.01, 0.02, 0.05, 0.1, 0.2, 0.5, 1, 2, 5, 10, 20, 50, 100\}$, we try all combinations of $(\gamma, h) \in \{0.1, 0.2, 0.5, 1, 2, 5, 10, 20, 50, 100\} \times \{0.1 \times [50]\}$ for Algorithm 1 (we also run ULD algorithm when $\alpha = 0$), and empirically find the best combination that requires the fewest iterations to meet $\left\|\mathbb{E}_{\mu_k}\boldsymbol{q} - \mathbb{E}_{\mu}\boldsymbol{q}\right\| \leq \epsilon$. We find that $h = 5$ already surpasses the stability limit of ULD algorithm, hence the range of step size covers the largest step size that are practically useable for ULD algorithm. Experiments are repeated with 100 different seeds to further reduce variance.

The results are shown in Figure 2. When $\alpha > 0$, HFHR algorithm consistently outperforms ULD algorithm (note it also does so when $\alpha = 0$ because HFHR uses a efficiency-wise-comparable but more refined discretization than ULD algorithm). In particular, when $\alpha = 0.5$ and 1, which are

empirically the best values we found for this experiment, HFHR algorithm achieves the specified $\epsilon$-closeness nearly $6\times$ times faster than ULD algorithm, and its decreased mixing time (compared to $\alpha = 0$ for the same algorithm) is consistent with the $\approx 0.46$ factor in Rmk.5.5). This empirical study corroborates that the acceleration HFHR dynamics creates also carries through to its discretization, and the acceleration of HFHR algorithm over ULD algorithm can be significant.

## 6.2 BAYESIAN NEURAL NETWORK

Now consider Bayesian neural network (BNN) which is a compelling learning model (Wilson, 2020); however, the focus won't be on its learning capability, and instead we just consider its training, which amounts to a practical, high-dim., multi-modal example of sampling tasks. It no longer satisfies the conditions of our analysis, and our goal is to show HFHR still accelerates. We use fully-connected network with [22, 10, 2] neurons, ReLU, standard Gaussian prior for all parameters, and compare ULD and HFHR on data set Parkinson from UCI repository (Dua & Graff, 2017).

Choices of hyper-parameter for Algorithm 1 and ULD algorithm are systematically investigated. For each pair $(\gamma, \alpha) \in \{0.1, 0.5, 1, 5, 10, 50, 100\}^2$, we empirically tune the step size to the stability limit of ULD algorithm, simulate 10,000 independent realizations, and use the ensemble to conduct Bayesian posterior prediction. HFHR will then use the same step size. For each $\gamma$, we plot the negative log likelihood of HFHR algorithm (with different $\alpha$ choices) and ULD algorithm on training and test data in Figure 3. Cases where $\alpha$ is too large for numerical stability are not drawn.

From Figure 3, we find that HFHR converges significantly faster than ULD in a wide range of setups. In general, the log-strongly-concave assumption required in Theorem 5.2 does not hold for multimodal target distributions. However, this numerical result shows that HFHR still accelerates ULD for highly complex models such as BNN, even when there is no obvious theoretical guarantee. It showcases the applicability and effectiveness of HFHR as a general sampling algorithm.

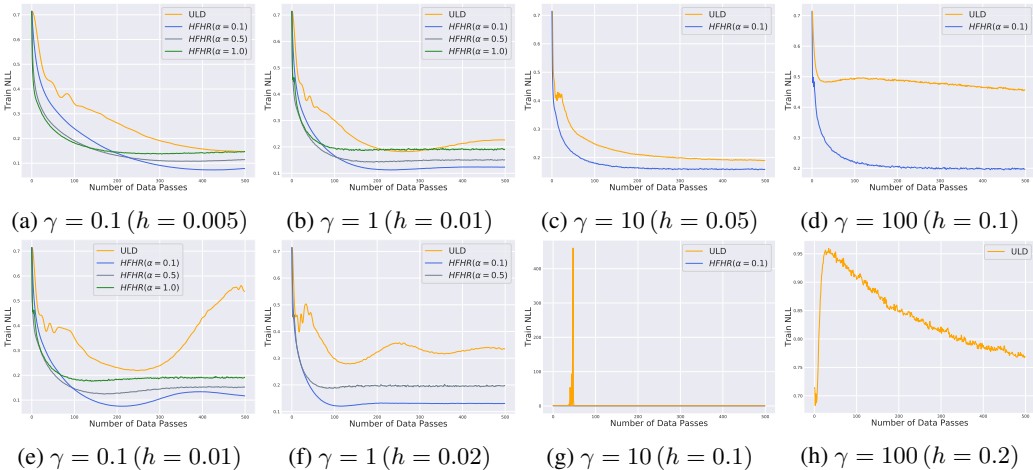

(a) $\gamma = 0.1$ ($h = 0.005$)    (b) $\gamma = 1$ ($h = 0.01$)    (c) $\gamma = 10$ ($h = 0.05$)    (d) $\gamma = 100$ ($h = 0.1$)

(e) $\gamma = 0.1$ ($h = 0.01$)    (f) $\gamma = 1$ ($h = 0.02$)    (g) $\gamma = 10$ ($h = 0.1$)    (h) $\gamma = 100$ ($h = 0.2$)

Figure 3: Training Negative Log-Likelihood (NLL) for various $\gamma$. Row 1: step sizes are close to the stability limit of ULD algorithm; Row 2: further increased step size exceeds that stability limit.

## 7 CONCLUSION AND DISCUSSION

This paper proposes HFHR, an accelerated gradient-based MCMC method for sampling. To demonstrate the acceleration enabled by HFHR, the geometric ergodicity of HFHR (both the continuous and the discretized versions) is quantified, and its convergence is provably faster than Underdamped Langevin Dynamics, which by itself is often already considered as an acceleration of Overdamped Langevin Dynamics. As HFHR is based on a new perspective, which is to turn NAG-SC optimizer with **finite** learning rate into a sampler, there are a number of interesting directions in which this work can be extended. Besides further theoretical investigations that aim at refining the error bounds, examples also include the followings: to scale HFHR up to large data sets, full gradient may be replaced by stochastic gradient (SG) — how to quantify, and hence optimize the performance of SG-HFHR? Can the generalization ability of HFHR trained learning models (e.g., BNN) be quantified, and how does it compare with that by LMC and/or KLMC? These will be future work.

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

# A  ADDITIONAL NOTATIONS

We introduce a few notations that are used in the main text as well as some proof. When $\nabla f$ is $L$-Lipschitz, the drift term $\begin{bmatrix} \boldsymbol{p} - \alpha \nabla f(\boldsymbol{q}) \\ -\gamma \boldsymbol{p} - \nabla f(\boldsymbol{q}) \end{bmatrix}$ in HFHR dynamics is also $L'$-Lipschitz, as proved in Lemma D.3, where

$$L' = \sqrt{2} \max \left\{ \sqrt{1 + \alpha^2} \max \left\{ \frac{1}{\sqrt{2}}, L \right\}, \sqrt{1 + \gamma^2} \right\}.$$

We show in Lemma D.5 that a linear-transformed HFHR dynamics satisfies the nice contraction property, the linear transformation $P$ we use is defined as

$$P = \begin{bmatrix} \gamma I & I \\ 0 & \sqrt{1 + \alpha\gamma} I \end{bmatrix} \in \mathbb{R}^{2d \times 2d}.$$

Denote the largest and the smallest singular value of $P$ by

$$\sigma_{\max} = \sqrt{\frac{\alpha\gamma}{2} + \frac{\gamma^2}{2} + \frac{\sqrt{\alpha^2\gamma^2 - 2\alpha\gamma^3 + 4\alpha\gamma + \gamma^4 + 4}}{2} + 1},$$

$$\sigma_{\min} = s\sqrt{\frac{\alpha\gamma}{2} + \frac{\gamma^2}{2} - \frac{\sqrt{\alpha^2\gamma^2 - 2\alpha\gamma^3 + 4\alpha\gamma + \gamma^4 + 4}}{2} + 1}$$

and its condition number by

$$\kappa' = \frac{\sigma_{\max}}{\sigma_{\min}} = \sqrt{\frac{\frac{\alpha\gamma}{2} + \frac{\gamma^2}{2} + \frac{\sqrt{\alpha^2\gamma^2 - 2\alpha\gamma^3 + 4\alpha\gamma + \gamma^4 + 4}}{2} + 1}{\frac{\alpha\gamma}{2} + \frac{\gamma^2}{2} - \frac{\sqrt{\alpha^2\gamma^2 - 2\alpha\gamma^3 + 4\alpha\gamma + \gamma^4 + 4}}{2} + 1}}.$$

The rate $\lambda'$ of exponential convergence of transformed HFHR dynamics is characterized in Lemma D.5 and is defined as

$$\lambda' = \min \left\{ \frac{m}{\gamma} + \alpha m, \frac{\gamma^2 - L}{\gamma} \right\}$$

given that $\gamma^2 > L$.

# B  PROOFS FOR THE CONTINUOUS DYNAMICS

*Notations and definitions can be found in Sec.3.*

## B.1  PROOF OF THEOREM 4.1

*Proof.* The Fokker-Plank equation of HFHR is given by

$$\partial_t \rho_t = -\nabla_{\boldsymbol{x}} \cdot \left( \begin{bmatrix} \boldsymbol{p} \\ -\nabla f(\boldsymbol{q}) \end{bmatrix} \rho_t \right) + \alpha \left( \nabla_{\boldsymbol{q}} \cdot (\nabla f(\boldsymbol{q}) \rho_t) + \Delta_{\boldsymbol{q}} \rho_t \right) + \gamma \left( \nabla_{\boldsymbol{p}} \cdot (\boldsymbol{p} \rho_t) + \Delta_{\boldsymbol{p}} \rho_t \right)$$

where $\nabla_{\boldsymbol{x}} = (\nabla_{\boldsymbol{q}}, \nabla_{\boldsymbol{p}})$. For $\pi \propto e^{-f(\boldsymbol{q}) - \frac{1}{2}\|\boldsymbol{p}\|^2}$, we have

$$\nabla_{\boldsymbol{x}} \cdot \left( \begin{bmatrix} \boldsymbol{p} \\ -\nabla f(\boldsymbol{q}) \end{bmatrix} \pi \right) = \left\langle \begin{bmatrix} \boldsymbol{p} \\ -\nabla f(\boldsymbol{q}) \end{bmatrix}, \nabla_{\boldsymbol{x}} \pi \right\rangle = 0,$$

$$\Delta_{\boldsymbol{q}} \pi = -\nabla_{\boldsymbol{q}} \cdot (\pi \nabla f(\boldsymbol{q}))$$

$$\Delta_{\boldsymbol{p}} \pi = -\nabla_{\boldsymbol{p}} \cdot (\pi \boldsymbol{p})$$

Therefore $\partial_t \pi = 0$ and hence $\pi$ is the invariant distribution of HFHR. □

## B.2 PROOF OF THEOREM 5.1

*Proof.* Consider two copies of HFHR that are driven by the same Brownian motion

$$\begin{cases} d\boldsymbol{q}_t = (\boldsymbol{p}_t - \alpha\nabla f(\boldsymbol{q}_t))dt + \sqrt{2\alpha}d\boldsymbol{B}_t^1 \\ d\boldsymbol{p}_t = (-\gamma\boldsymbol{p}_t - \nabla f(\boldsymbol{q}_t))dt + \sqrt{2\gamma}d\boldsymbol{B}_t^2 \end{cases}, \quad \begin{cases} d\tilde{\boldsymbol{q}}_t = (\tilde{\boldsymbol{p}}_t - \alpha\nabla f(\tilde{\boldsymbol{q}}_t))dt + \sqrt{2\alpha}d\boldsymbol{B}_t^1 \\ d\tilde{\boldsymbol{p}}_t = (-\gamma\tilde{\boldsymbol{p}}_t - \nabla f(\tilde{\boldsymbol{q}}_t))dt + \sqrt{2\gamma}d\boldsymbol{B}_t^2 \end{cases},$$

where we set $(\tilde{\boldsymbol{q}}_0, \tilde{\boldsymbol{p}}_0) \sim \pi$, $\boldsymbol{p}_0 = \tilde{\boldsymbol{p}}_0$ and $\boldsymbol{q}_0$ such that

$$W_2^2(\mu_0, \mu) = \mathbb{E}\left[\|\boldsymbol{q}_0 - \tilde{\boldsymbol{q}}_0\|_2^2\right], \quad \boldsymbol{q}_0 \sim \mu_0$$

Denote $\begin{bmatrix} \boldsymbol{\phi}_t \\ \boldsymbol{\psi}_t \end{bmatrix} = P \begin{bmatrix} \boldsymbol{q}_t - \tilde{\boldsymbol{q}}_t \\ \boldsymbol{p}_t - \tilde{\boldsymbol{p}}_t \end{bmatrix}$ where $P$ is defined in Appendix A. By Lemma D.5 and the assumption on $\alpha, \gamma$, we have

$$\left\| \begin{bmatrix} \boldsymbol{\phi}_t \\ \boldsymbol{\psi}_t \end{bmatrix} \right\|^2 \le e^{-2(\frac{m}{\gamma}+m\alpha)t} \left\| \begin{bmatrix} \boldsymbol{\phi}_0 \\ \boldsymbol{\psi}_0 \end{bmatrix} \right\|^2.$$

Therefore we obtain

$$W_2^2(\mu_t, \mu) = \inf_{(\boldsymbol{q}_t, \tilde{\boldsymbol{q}}_t) \sim \Pi(\mu_t, \mu)} \mathbb{E}\|\boldsymbol{q}_t - \tilde{\boldsymbol{q}}_t\|^2$$

$$\le \inf_{(\boldsymbol{q}_t, \tilde{\boldsymbol{q}}_t) \sim \Pi(\mu_t, \mu), (\boldsymbol{p}_t, \tilde{\boldsymbol{p}}_t) \sim \Pi(\nu_t, \nu)} \mathbb{E}\left\| \begin{bmatrix} \boldsymbol{q}_t - \tilde{\boldsymbol{q}}_t \\ \boldsymbol{p}_t - \tilde{\boldsymbol{p}}_t \end{bmatrix} \right\|^2$$

$$\le \mathbb{E}\|P^{-1}\|_2^2 \left\| \begin{bmatrix} \boldsymbol{\phi}_t \\ \boldsymbol{\psi}_t \end{bmatrix} \right\|^2$$

$$\le \mathbb{E}\|P^{-1}\|_2^2 e^{-2(\frac{m}{\gamma}+m\alpha)t} \left\| \begin{bmatrix} \boldsymbol{\phi}_0 \\ \boldsymbol{\psi}_0 \end{bmatrix} \right\|^2$$

$$\le (\kappa')^2 e^{-2(\frac{m}{\gamma}+m\alpha)t} \left\| \begin{bmatrix} \boldsymbol{q}_0 - \tilde{\boldsymbol{q}}_0 \\ \boldsymbol{p}_0 - \tilde{\boldsymbol{p}}_0 \end{bmatrix} \right\|^2$$

$$= (\kappa')^2 e^{-2(\frac{m}{\gamma}+m\alpha)t} W_2^2(\mu_0, \mu)$$

Taking square root yields the desired result.

$\square$

## C ARBITRARY LONG TIME DISCRETIZATION ERROR OF ALGORITHM 1

**Theorem C.1.** *Under Conditions A1 and further assume the function $\nabla\Delta f$ grows at most linearly, i.e., $\|\nabla\Delta f(\boldsymbol{q})\| \le G\sqrt{1 + \|\boldsymbol{q}\|^2}, \forall \boldsymbol{q} \in \mathbb{R}^d$. Also suppose $\gamma$ in HFHR dynamics satisfy $\gamma^2 > L$. Then there exist $C, h_0 > 0$, such that for $0 < h \le h_0$, we have*

$$\left(\mathbb{E}\|\boldsymbol{x}_k - \bar{\boldsymbol{x}}_k\|^2\right)^{\frac{1}{2}} \le Ch$$

*where $\bar{\boldsymbol{x}}_k$ is the $k$-th iterate of Algorithm 1 with step size $h$ starting from $\boldsymbol{x}_0$, $\boldsymbol{x}_k$ is the solution of HFHR dynamics at time $kh$, starting from $\boldsymbol{x}_0$. This result holds uniformly for all $k \ge 0$ and $k$ can go to $\infty$. In particular, $C = \mathcal{O}(\sqrt{d})$ and if $\gamma - \frac{L+m}{\gamma} \ge m\alpha$, then there exists $b > 0$, independent of $\alpha$ and is of order $\mathcal{O}(\sqrt{d})$, such that*

$$C \le \frac{b}{m}(\alpha^2 - \frac{\alpha}{\gamma} + \frac{1}{\gamma^2}). \tag{11}$$

*Proof.* Denote $t_k = kh$, the solution of the HFHR dynamics at time $t$ by $\boldsymbol{x}_{0,\boldsymbol{x}_0}(t)$, the $k$-th iterates of the Strang's splitting method of HFHR dynamics by $\bar{\boldsymbol{x}}_{0,\boldsymbol{x}_0}(kh)$. Both $\boldsymbol{x}_{0,\boldsymbol{x}_0}(t)$ and $\bar{\boldsymbol{x}}_{0,\boldsymbol{x}_0}(kh)$

start from the same initial value $\boldsymbol{x}_0$. The linear transformation $P$ defined in Appendix A, transforms the solution of HFHR dynamics into $\boldsymbol{y}_{0,P\boldsymbol{x}_0}(t) = P\boldsymbol{x}_{0,\boldsymbol{x}_0}(t)$ and the Strang's splitting discretization of HFHR into $\bar{\boldsymbol{y}}_{0,P\boldsymbol{x}_0}(t) = P\bar{\boldsymbol{x}}_{0,\boldsymbol{x}_0}(t)$.

For the ease of notation, we write $\boldsymbol{y}_{0,\boldsymbol{y}_0}(t_k)$ as $\boldsymbol{y}_k$ and $\bar{\boldsymbol{y}}_{0,\boldsymbol{y}_0}(t_k)$ as $\bar{\boldsymbol{y}}_k$. We have the following identity

$$
\begin{aligned}
\mathbb{E}\big\|\boldsymbol{y}_{k+1} - \bar{\boldsymbol{y}}_{k+1}\big\|^2 =& \mathbb{E}\Big\|\boldsymbol{y}_{t_k,\boldsymbol{y}_k}(h) - \bar{\boldsymbol{y}}_{t_k,\bar{\boldsymbol{y}}_k}(h)\Big\|^2 \\
=& \mathbb{E}\Big\|\boldsymbol{y}_{t_k,\boldsymbol{y}_k}(h) - \boldsymbol{y}_{t_k,\bar{\boldsymbol{y}}_k}(h) + \boldsymbol{y}_{t_k,\bar{\boldsymbol{y}}_k}(h) - \bar{\boldsymbol{y}}_{t_k,\bar{\boldsymbol{y}}_k}(h)\Big\|^2 \\
=& \underbrace{\mathbb{E}\Big\|\boldsymbol{y}_{t_k,\boldsymbol{y}_k}(h) - \boldsymbol{y}_{t_k,\bar{\boldsymbol{y}}_k}(h)\Big\|^2}_{①} + \underbrace{\mathbb{E}\Big\|\boldsymbol{y}_{t_k,\bar{\boldsymbol{y}}_k}(h) - \bar{\boldsymbol{y}}_{t_k,\bar{\boldsymbol{y}}_k}(h)\Big\|^2}_{②} \\
& + \underbrace{2\,\mathbb{E}\Big\langle \boldsymbol{y}_{t_k,\boldsymbol{y}_k}(h) - \boldsymbol{y}_{t_k,\bar{\boldsymbol{y}}_k}(h), \boldsymbol{y}_{t_k,\bar{\boldsymbol{y}}_k}(h) - \bar{\boldsymbol{y}}_{t_k,\bar{\boldsymbol{y}}_k}(h)\Big\rangle}_{③}
\end{aligned}
$$

By Lemma D.5, when $0 < h < \frac{1}{2\lambda'}$, term $①$ can be upper bounded as

$$
\begin{aligned}
\mathbb{E}\Big\|\boldsymbol{y}_{t_k,\boldsymbol{y}_k}(h) - \boldsymbol{y}_{t_k,\bar{\boldsymbol{y}}_k}(h)\Big\|^2 \le& e^{-2\lambda'h}\mathbb{E}\|\boldsymbol{y}_k - \bar{\boldsymbol{y}}_k\|^2 \\
\le& \left(1 - 2\lambda'h + 2(\lambda')^2h^2\right)\mathbb{E}\|\boldsymbol{y}_k - \bar{\boldsymbol{y}}_k\|^2 \\
\le& \left(1 - \lambda'h\right)\mathbb{E}\|\boldsymbol{y}_k - \bar{\boldsymbol{y}}_k\|^2
\end{aligned}
$$

where the second inequality is due to $e^{-x} \le 1 - x + \frac{x^2}{2}, \forall x > 0$.

For term $②$, we have by Lemma D.8 that

$$
\mathbb{E}\Big\|\boldsymbol{y}_{t_k,\bar{\boldsymbol{y}}_k}(h) - \bar{\boldsymbol{y}}_{t_k,\bar{\boldsymbol{y}}_k}(h)\Big\|^2 \le \sigma_{\max}^2\,\mathbb{E}\Big\|\boldsymbol{x}_{t_k,\bar{\boldsymbol{x}}_k}(h) - \bar{\boldsymbol{x}}_{t_k,\bar{\boldsymbol{x}}_k}(h)\Big\|^2 \le \sigma_{\max}^2 C_2^2 h^3
$$

where $\sigma_{\max}$ is the largest singular value of matrix $P$.

For term $③$, we have by Lemma D.1 that

$$
\begin{aligned}
& 2\mathbb{E}\Big\langle \boldsymbol{y}_{t_k,\boldsymbol{y}_k}(h) - \boldsymbol{y}_{t_k,\bar{\boldsymbol{y}}_k}(h), \boldsymbol{y}_{t_k,\bar{\boldsymbol{y}}_k}(h) - \bar{\boldsymbol{y}}_{t_k,\bar{\boldsymbol{y}}_k}(h)\Big\rangle \\
=& 2\mathbb{E}\Big\langle \boldsymbol{y}_k - \bar{\boldsymbol{y}}_k + \boldsymbol{z}, \boldsymbol{y}_{t_k,\bar{\boldsymbol{y}}_k}(h) - \bar{\boldsymbol{y}}_{t_k,\bar{\boldsymbol{y}}_k}(h)\Big\rangle \\
=& \underbrace{2\mathbb{E}\Big\langle \boldsymbol{y}_k - \bar{\boldsymbol{y}}_k, \boldsymbol{y}_{t_k,\bar{\boldsymbol{y}}_k}(h) - \bar{\boldsymbol{y}}_{t_k,\bar{\boldsymbol{y}}_k}(h)\Big\rangle}_{3a} + \underbrace{2\mathbb{E}\Big\langle \boldsymbol{z}, \boldsymbol{y}_{t_k,\bar{\boldsymbol{y}}_k}(h) - \bar{\boldsymbol{y}}_{t_k,\bar{\boldsymbol{y}}_k}(h)\Big\rangle}_{3b}
\end{aligned}
$$

For term $\boxed{3a}$, by the tower property of conditional expectation, we have

$$
\begin{aligned}
2\mathbb{E}\left\langle \boldsymbol{y}_k - \bar{\boldsymbol{y}}_k, \boldsymbol{y}_{t_k,\bar{\boldsymbol{y}}_k}(h) - \bar{\boldsymbol{y}}_{t_k,\bar{\boldsymbol{y}}_k}(h)\right\rangle =& 2\mathbb{E}\left[\mathbb{E}\left[\left\langle \boldsymbol{y}_k - \bar{\boldsymbol{y}}_k, \boldsymbol{y}_{t_k,\bar{\boldsymbol{y}}_k}(h) - \bar{\boldsymbol{y}}_{t_k,\bar{\boldsymbol{y}}_k}(h)\right\rangle \Big| \mathcal{F}_k\right]\right] \\
=& 2\mathbb{E}\left\langle \boldsymbol{y}_k - \bar{\boldsymbol{y}}_k, \mathbb{E}\left[\boldsymbol{y}_{t_k,\bar{\boldsymbol{y}}_k}(h) - \bar{\boldsymbol{y}}_{t_k,\bar{\boldsymbol{y}}_k}(h)\Big| \mathcal{F}_k\right]\right\rangle \\
\leq& 2\sqrt{\mathbb{E}\|\boldsymbol{y}_k - \bar{\boldsymbol{y}}_k\|^2}\sqrt{\mathbb{E}\left\|\mathbb{E}\left[\boldsymbol{y}_{t_k,\bar{\boldsymbol{y}}_k}(h) - \bar{\boldsymbol{y}}_{t_k,\bar{\boldsymbol{y}}_k}(h)\Big| \mathcal{F}_k\right]\right\|^2} \\
\leq& 2\sqrt{\mathbb{E}\|\boldsymbol{y}_k - \bar{\boldsymbol{y}}_k\|^2}\sqrt{\sigma_{\max}^2\mathbb{E}\left\|\mathbb{E}\left[\boldsymbol{x}_{t_k,\bar{\boldsymbol{x}}_k}(h) - \bar{\boldsymbol{x}}_{t_k,\bar{\boldsymbol{x}}_k}(h)\Big| \mathcal{F}_k\right]\right\|^2} \\
\leq& 2\sqrt{\mathbb{E}\|\boldsymbol{y}_k - \bar{\boldsymbol{y}}_k\|^2}\sqrt{\sigma_{\max}^2 C_1^2 h^4} \\
\leq& 2\sigma_{\max}C_1\sqrt{\mathbb{E}\|\boldsymbol{y}_k - \bar{\boldsymbol{y}}_k\|^2}h^2.
\end{aligned}
$$

For term $\boxed{3b}$, when $0 < h < \frac{1}{4L''}$ we have by Lemma D.1 and Lemma D.8

$$
\begin{aligned}
2\mathbb{E}\left\langle \boldsymbol{z}, \boldsymbol{y}_{t_k,\bar{\boldsymbol{y}}_k}(h) - \bar{\boldsymbol{y}}_{t_k,\bar{\boldsymbol{y}}_k}(h)\right\rangle \leq& 2\sqrt{\mathbb{E}\|\boldsymbol{z}\|^2}\sqrt{\mathbb{E}\left\|\boldsymbol{y}_{t_k,\bar{\boldsymbol{y}}_k}(h) - \bar{\boldsymbol{y}}_{t_k,\bar{\boldsymbol{y}}_k}(h)\right\|^2} \\
=& 2\sqrt{\mathbb{E}\|\boldsymbol{z}\|^2}\sqrt{\mathbb{E}\left[\mathbb{E}\left[\left\|\boldsymbol{y}_{t_k,\bar{\boldsymbol{y}}_k}(h) - \bar{\boldsymbol{y}}_{t_k,\bar{\boldsymbol{y}}_k}(h)\right\|^2\Big| \mathcal{F}_k\right]\right]} \\
=& 2\sqrt{\mathbb{E}\|\boldsymbol{z}\|^2}\sqrt{\sigma_{\max}^2\mathbb{E}\left[\mathbb{E}\left[\left\|\boldsymbol{x}_{t_k,\bar{\boldsymbol{x}}_k}(h) - \bar{\boldsymbol{x}}_{t_k,\bar{\boldsymbol{x}}_k}(h)\right\|^2\Big| \mathcal{F}_k\right]\right]} \\
\leq& 2\sigma_{\max}\sqrt{\tilde{C}\mathbb{E}\|\boldsymbol{y}_k - \bar{\boldsymbol{y}}_k\|^2 h^2}\sqrt{C_2^2 h^3} \\
\leq& 2\sigma_{\max}C_2\sqrt{\tilde{C}}\sqrt{\mathbb{E}\|\boldsymbol{y}_k - \bar{\boldsymbol{y}}_k\|^2}h^{\frac{5}{2}}
\end{aligned}
$$

where $\tilde{C} = 2\left(L''\right)^2 = 2(\kappa')^2\left(L'\right)^2$ is from Lemma D.1 and Lemma D.3.

Recall both $C_1$ and $C_2$ depend on $\|\boldsymbol{x}_k\|$ and we would like to upper bound this term. To this end, consider $\tilde{\boldsymbol{x}}(t)$, a solution of HFHR dynamics with initial value $\tilde{\boldsymbol{x}}_0$ that follows the invariant distribution $\tilde{\boldsymbol{x}}_0 \sim \pi$ and realizes $W_2(\pi_0, \pi)$, i.e., $\mathbb{E}\|\tilde{\boldsymbol{x}}_0 - \boldsymbol{x}_0\|^2 = W_2^2(\pi_0, \pi)$.

Denote $\tilde{\boldsymbol{x}}_k = \tilde{\boldsymbol{x}}(kh)$ and $e_k = \left( \mathbb{E}\|\boldsymbol{y}_k - \bar{\boldsymbol{y}}_k\|^2 \right)^{\frac{1}{2}}$, we then have

$$
\begin{aligned}
\mathbb{E}\|\bar{\boldsymbol{x}}_k\|^2 =& \mathbb{E}\|\boldsymbol{x}_k + \bar{\boldsymbol{x}}_k - \boldsymbol{x}_k\|^2 \\
\leq& 2\mathbb{E}\|\boldsymbol{x}_k\|^2 + 2\mathbb{E}\|\bar{\boldsymbol{x}}_k - \boldsymbol{x}_k\|^2 \\
\leq& 4\mathbb{E}\|\tilde{\boldsymbol{x}}_k\|^2 + 4\mathbb{E}\|\tilde{\boldsymbol{x}}_k - \boldsymbol{x}_k\|^2 + 2\mathbb{E}\|\bar{\boldsymbol{x}}_k - \boldsymbol{x}_k\|^2 \\
=& 4\mathbb{E}\|\tilde{\boldsymbol{x}}_k\|^2 + 4\mathbb{E}\left\|P^{-1}P(\tilde{\boldsymbol{x}}_k - \boldsymbol{x}_k)\right\|^2 + 2\mathbb{E}\left\|P^{-1}P(\bar{\boldsymbol{x}}_k - \boldsymbol{x}_k)\right\|^2 \\
\leq& 4\left( \int_{\mathbb{R}^d} \|\boldsymbol{q}\|^2 \, d\mu + d \right) + \frac{4}{\sigma_{\min}^2}\mathbb{E}\left\|P(\tilde{\boldsymbol{x}}_k - \boldsymbol{x}_k)\right\|^2 + \frac{2}{\sigma_{\min}^2}\mathbb{E}\|\bar{\boldsymbol{y}}_k - \boldsymbol{y}_k\|^2 \\
\overset{(i)}{\leq}& 4\left( \int_{\mathbb{R}^d} \|\boldsymbol{q}\|^2 \, d\mu + d \right) + \frac{4}{\sigma_{\min}^2}e^{-2\lambda'kh}\mathbb{E}\left\|P(\tilde{\boldsymbol{x}}_0 - \boldsymbol{x}_0)\right\|^2 + \frac{2}{\sigma_{\min}^2}e_k^2 \\
\leq& 4\left( \int_{\mathbb{R}^d} \|\boldsymbol{q}\|^2 \, d\mu + d \right) + 4\kappa^2 W_2^2(\pi_0, \pi) + \frac{2}{\sigma_{\min}^2}e_k^2 \\
\triangleq& Fe_k^2 + G
\end{aligned}
$$

where $(i)$ is due to Lemma D.5. Recall from Lemma D.8, we have

$$
\begin{aligned}
C_1 \leq A_1\sqrt{\mathbb{E}\|\bar{x}_k\|^2} + B_1 \leq A_1\sqrt{F}e_k + (A_1\sqrt{G} + B_1) \triangleq U_1 e_k + V_1 \\
C_2 \leq A_2\sqrt{\mathbb{E}\|\bar{x}_k\|^2} + B_2 \leq A_2\sqrt{F}e_k + (A_2\sqrt{G} + B_2) \triangleq U_2 e_k + V_2
\end{aligned}
$$

where

$$
\begin{aligned}
A_1 =& (L+G)\max\{\alpha + 1.25, \gamma + 1\}(1.74 + 0.71\alpha) \\
B_1 =& (L+G)\max\{\alpha + 1.25, \gamma + 1\}\left[0.5\alpha + (1.26\sqrt{\alpha} + 1.14\alpha\sqrt{\alpha} + 2.32\sqrt{\gamma})\sqrt{hd}\right] \\
A_2 =& L\max\{\alpha + 1.25, \gamma + 1\}(1.92 + 2.30\alpha L)\sqrt{h} \\
B_2 =& L\max\{\alpha + 1.25, \gamma + 1\}(2.60\sqrt{\alpha} + 3.34\sqrt{\gamma}h)\sqrt{d}
\end{aligned}
$$

Combine the above and bounds for terms $\boxed{1}$, $\boxed{2}$, $\boxed{3a}$ and $\boxed{3b}$, we then obtain

$$
\begin{aligned}
e_{k+1}^2 \leq &(1-\lambda'h)e_k^2 + \sigma_{\max}^2 C_2^2 h^3 + 2\sigma_{\max}C_1 e_k h^2 + 2\sigma_{\max}C_2\sqrt{\tilde{C}}e_k h^{\frac{5}{2}} \\
\leq &(1-\lambda'h)e_k^2 + \sigma_{\max}^2 2(U_2^2 e_k^2 + V_2^2)h^3 + 2\sigma_{\max}(U_1 e_k + V_1)e_k h^2 + 2\sigma_{\max}(U_2 e_k + V_2)\sqrt{\tilde{C}}e_k h^{\frac{5}{2}} \\
= &\left(1 - \lambda'h + 2\sigma_{\max}^2 U_2^2 h^3 + 2\sigma_{\max}U_1 h^2 + 2\sigma_{\max}U_2\sqrt{\tilde{C}}h^{\frac{5}{2}}\right)e_k^2 \\
&+ \left(2\sigma_{\max}V_1 + 2\sigma_{\max}V_2\sqrt{\tilde{C}h}\right)e_k h^2 + 2\sigma_{\max}^2 V_2^2 h^3 \\
\leq &\left(1 - \lambda'h + 2\sigma_{\max}^2 U_2^2 h^3 + 2\sigma_{\max}U_1 h^2 + 2\sigma_{\max}U_2\sqrt{\tilde{C}}h^{\frac{5}{2}}\right)e_k^2 + \frac{\lambda'}{8}he_k^2 \\
&+ \frac{2\left(2\sigma_{\max}V_1 + 2\sigma_{\max}V_2\sqrt{\tilde{C}h}\right)^2}{\lambda'}h^3 + 2\sigma_{\max}^2 V_2^2 h^3 \\
= &\left(1 - \frac{7}{8}\lambda'h + 2\sigma_{\max}^2 U_2^2 h^3 + 2\sigma_{\max}U_1 h^2 + 2\sigma_{\max}U_2\sqrt{\tilde{C}}h^{\frac{5}{2}}\right)e_k^2 \\
&+ \left(\frac{2\left(2\sigma_{\max}V_1 + 2\sigma_{\max}V_2\sqrt{\tilde{C}h}\right)^2}{\lambda'} + 2\sigma_{\max}^2 V_2^2\right)h^3 \\
\overset{(i)}{\leq} &(1 - \frac{1}{2}\lambda'h)e_k^2 + \left(\frac{2\left(2\sigma_{\max}V_1 + 2\sigma_{\max}V_2\sqrt{\tilde{C}h}\right)^2}{\lambda'} + 2\sigma_{\max}^2 V_2^2\right)h^3 \\
\triangleq &(1 - \frac{1}{2}\lambda'h)e_k^2 + Kh^3
\end{aligned}
$$

where $(i)$ is due to $h < \min\{h_1, h_2, h_3\}$ and

$$
\begin{aligned}
h_1 =& \frac{\sqrt{\lambda'}}{4\sqrt{2}\kappa'L\max\{\alpha + 1.25, \gamma + 1\}(1.92 + 2.30\alpha L)}, \\
h_2 =& \frac{\lambda'}{16\sqrt{2}\kappa'(L+G)\max\{\alpha + 1.25, \gamma + 1\}(1.74 + 0.71\alpha)}, \\
h_3 =& \frac{\lambda'}{8\kappa'L\max\{\alpha + 1.25, \gamma + 1\}(1.92 + 2.30\alpha L)}.
\end{aligned}
$$

Unfolding the above inequality, we arrive at

$$
\begin{aligned}
e_k^2 \leq &\left(1 - \frac{\lambda'}{2}h\right)^k e_0^2 + \left(1 + (1 - \frac{\lambda'}{2}h) + \cdots + (1 - \frac{\lambda'}{2}h)^{k-1}\right)Kh^3 \\
\overset{(i)}{\leq} &Kh^3 \sum_{i=0}^{\infty}\left(1 - \frac{\lambda'}{2}h\right)^i \\
= &\frac{2K}{\lambda'}h^2
\end{aligned}
$$

where $(i)$ is due to $e_k = 0$. Therefore

$$
\left(\mathbb{E}\|\boldsymbol{x}_k - \bar{\boldsymbol{x}}_k\|^2\right)^{\frac{1}{2}} = \left(\mathbb{E}\left\|P^{-1}(\boldsymbol{y}_k - \bar{\boldsymbol{y}}_k)\right\|^2\right)^{\frac{1}{2}} \leq \frac{1}{\sigma_{\min}}e_k \leq \frac{1}{\sigma_{\min}}\sqrt{\frac{2K}{\lambda'}}h
$$

Collecting all the constants and we have

$$
\begin{aligned}
\frac{1}{\sigma_{\min}}\sqrt{\frac{2K}{\lambda'}} \leq & \frac{8\kappa'}{\lambda'}(L+G)\max\{\alpha+1.25,\gamma+1\}(1.74+0.71\alpha)\left(\sqrt{\int_{\mathbb{R}^d}\|q\|^2\,d\mu + d} + \kappa'W_2(\pi_0,\pi)\right)\\
& + \frac{4\kappa'}{\lambda'}(L+G)\max\{\alpha+1.25,\gamma+1\}\left(0.5\alpha+(1.26\sqrt{\alpha}+1.14\alpha\sqrt{\alpha}+2.32\sqrt{\gamma})\sqrt{d}\right)\\
& + \frac{8\kappa'}{\sqrt{\lambda'}}\left(\frac{\sqrt{\kappa'L'}}{\sqrt{\lambda'}}+1\right)L\max\{\alpha+1.25,\gamma+1\}(1.92+2.30\alpha L)\left(\sqrt{\int_{\mathbb{R}^d}\|q\|^2\,d\mu + d}+\kappa'W_2(\pi_0,\pi)\right)\\
& + \frac{4\kappa'}{\sqrt{\lambda'}}\left(\frac{\sqrt{\kappa'L'}}{\sqrt{\lambda'}}+1\right)L\max\{\alpha+1.25,\gamma+1\}(2.60\sqrt{\alpha}+3.34\sqrt{\gamma})\sqrt{d}\\
\triangleq & \; C
\end{aligned}
$$

It is clear that in terms of the dependence on dimension $d$, we have $C = \mathcal{O}(\sqrt{d})$. In the regime where $\frac{\gamma^2-L}{\gamma} \geq \frac{m}{\gamma}+m\alpha$, then $\lambda' = \frac{m}{\gamma}+m\alpha$. Recall the definition of $\kappa'$ and there exist $A', B' > 0$ such that $\kappa' \leq A'\sqrt{\alpha}+B'$. It follows that

$$
C \leq \frac{a_1\alpha^3 + a_2\alpha^{\frac{5}{2}} + a_3\alpha^2 + a_4\alpha^{\frac{3}{2}} + a_5\alpha + a_6\alpha^{\frac{1}{2}} + a_7}{\lambda'} \leq b\frac{\alpha^3+\frac{1}{\gamma^3}}{\lambda'} = b\frac{\alpha^3+\frac{1}{\gamma^3}}{\frac{m}{\gamma}+m\alpha} = \frac{b}{m}(\alpha^2 - \frac{\alpha}{\gamma} + \frac{1}{\gamma^2})
$$

for some positive constants $a_1, a_2, a_3, a_4, a_5, a_6, a_7, b > 0$ and independent of $\alpha$, in particular, we have $b = \mathcal{O}(\sqrt{d})$. $\qquad\square$

## C.1 PROOF OF THEOREM 5.2

*Proof.* Denote the $k$-th iterate of the Strang's splitting method of HFHR by $\bar{x}_k$ with time step $h$, the solution of HFHR dynamics at time $hk$ by $x_k$. Both $\bar{x}_k$ and $x_k$ start from $x_0 = \begin{bmatrix} q_0 \\ p_0 \end{bmatrix}$. Also denote the solution of HFHR dynamics starting from $\tilde{x}_0$ at time $kh$ by $\tilde{x}_k$ where $\tilde{x}_0 = \begin{bmatrix} \tilde{q}_0 \\ p_0 \end{bmatrix}$, $(\tilde{q}_0, \tilde{p}_0) \sim \pi$ and $\mathbb{E}\left\|\begin{bmatrix} q_0 - \tilde{q}_0 \\ p_0 - \tilde{p}_0 \end{bmatrix}\right\|^2 = W_2^2(\pi_0, \pi)$. Since $\pi$ is the invariant distribution of HFHR dynamics, it follows that $\tilde{x}_k \sim \pi$.

By Lemma D.5 and Theorem C.1, we have

$$
\begin{aligned}
W_2^2(\mu_k, \mu) &= \inf_{\xi \in \Pi(\mu_k, \mu)} \mathbb{E}_{(q_1, q_2) \sim \xi}\|q_1 - q_2\|^2\\
&\leq \inf_{\xi \in \Pi(\pi_k, \pi)} \mathbb{E}_{(x_1, x_2) \sim \xi}\|x_1 - x_2\|^2\\
&\leq \mathbb{E}\|\bar{x}_k - \tilde{x}_k\|^2\\
&\leq 2C^2h^2 + 2\mathbb{E}\left\|P^{-1}P(x_k - \tilde{x}_k)\right\|^2\\
&\leq 2C^2h^2 + 2\|P^{-1}\|_2^2\mathbb{E}\left\|P(x_k - \tilde{x}_k)\right\|^2\\
&\leq 2C^2h^2 + 2\|P^{-1}\|_2^2e^{-2\lambda'kh}\mathbb{E}\left\|P(x_0 - \tilde{x}_0)\right\|^2\\
&\leq 2C^2h^2 + 2(\kappa')^2e^{-2\lambda'kh}W_2^2(\pi_0, \pi)
\end{aligned}
$$

Take square root on both sides and apply $\sqrt{a^2+b^2} \leq a+b$, we obtain

$$
W_2(\mu_k, \mu) \leq \sqrt{2}Ch + \sqrt{2}\kappa'e^{-\lambda'kh}W_2(\pi_0, \pi).
$$

$\qquad\square$

## C.2  PROOF OF COROLLARY 5.4

*Proof.* By Theorem 5.2, we have

$$W_2(\mu_k, \mu) \leq \sqrt{2}Ch + \sqrt{2}\kappa' e^{-\lambda' kh} W_2(\pi_0, \pi).$$

Given any target accuracy $\epsilon > 0$, if we run the Strang's splitting method of HFHR with $h^\star = \min\{h_0, \frac{\epsilon}{2\sqrt{2}C}\}$, then after $k^\star = \frac{1}{\lambda'} \max\{\frac{1}{h_0}, \frac{2\sqrt{2}C}{\epsilon}\} \log \frac{2\sqrt{2}\kappa' W_2(\pi_0, \pi)}{\epsilon}$, we have

$$W_2(\mu_{k^\star}, \mu) \leq \sqrt{2}Ch + \sqrt{2}\kappa' e^{-\lambda' kh} W_2(\mu_0, \mu) \leq \frac{\epsilon}{2} + \frac{\epsilon}{2} = \epsilon.$$

Recall $C = \mathcal{O}(\sqrt{d})$, when high accuracy is needed, e.g. $\epsilon < 2\sqrt{2}Ch_0$, the iteration complexity to reach $\epsilon$-accuracy under 2-Wasserstein distance is $k^\star = \mathcal{O}(\frac{\sqrt{d}}{\epsilon} \log \frac{1}{\epsilon}) = 2\sqrt{2}\frac{C}{\lambda'}\frac{1}{\epsilon} \log \frac{2\sqrt{2}\kappa' W_2(\pi_0, \pi)}{\epsilon} = \tilde{\mathcal{O}}(\frac{\sqrt{d}}{\epsilon})$. Recall from Theorem C.1, $C \leq \frac{b}{m}(\alpha^2 - \frac{\alpha}{\gamma} + \frac{1}{\gamma^2})$, we have

$$\frac{C}{\lambda'} \leq \frac{b}{m^2} \frac{\alpha^2 - \frac{\alpha}{\gamma} + \frac{1}{\gamma^2}}{\frac{1}{\gamma} + \alpha}$$

Denote $g(\alpha) = \frac{b}{m^2} \frac{\alpha^2 - \frac{\alpha}{\gamma} + \frac{1}{\gamma^2}}{\frac{1}{\gamma} + \alpha}$, simple calculation shows that $\alpha^\star = \text{argmin}_{\alpha \geq 0} g(\alpha) = \frac{\sqrt{3}-1}{\gamma} = \mathcal{O}(\frac{1}{\gamma})$. $\qquad\square$

# D  TECHNICAL/AUXILIARY LEMMAS AND THEIR PROOFS

## D.1  DEPENDENCE OF ERROR OF SDE ON INITIAL VALUES

**Lemma D.1.** *Consider the following two SDE with different initial condition*

$$\begin{cases} d\boldsymbol{x}_t = \boldsymbol{a}(\boldsymbol{x}_t)dt + \boldsymbol{\sigma}d\boldsymbol{W}_t, \\ \boldsymbol{x}(0) = \boldsymbol{x}_0 \end{cases} \qquad \begin{cases} d\boldsymbol{y}_t = \boldsymbol{a}(\boldsymbol{y}_t)dt + \boldsymbol{\sigma}d\boldsymbol{W}_t, \\ \boldsymbol{y}(0) = \boldsymbol{y}_0 \end{cases}$$

*where $\boldsymbol{a}(\boldsymbol{u}) \in \mathbb{R}^d$ is L-Lipschitz, and $\boldsymbol{\sigma} \in \mathbb{R}^{n \times n}$ is a constant matrix. For $0 < h < \frac{1}{4L}$, we have the following representation*

$$\boldsymbol{x}_h - \boldsymbol{y}_h = \boldsymbol{x}_0 - \boldsymbol{y}_0 + \boldsymbol{z}$$

*with*

$$E\|\boldsymbol{z}\|^2 \leq 2L^2 \|\boldsymbol{x}_0 - \boldsymbol{y}_0\|^2 h^2$$

*Proof.* Let $\boldsymbol{z} = (\boldsymbol{x}_h - \boldsymbol{y}_h) - (\boldsymbol{x}_0 - \boldsymbol{y}_0) = \int_0^h \boldsymbol{a}(\boldsymbol{x}_s) - \boldsymbol{a}(\boldsymbol{y}_s)ds$. Ito's lemma readily implies that

$$\mathbb{E}\|\boldsymbol{x}_h - \boldsymbol{y}_h\|^2 = \|\boldsymbol{x}_0 - \boldsymbol{y}_0\|^2 + 2\mathbb{E}\int_0^h \langle \boldsymbol{x}_s - \boldsymbol{y}_s, \boldsymbol{a}(\boldsymbol{x}_s) - \boldsymbol{a}(\boldsymbol{y}_s)\rangle ds$$

$$\leq \|\boldsymbol{x}_0 - \boldsymbol{y}_0\|^2 + 2L\int_0^h \mathbb{E}\|\boldsymbol{x}_s - \boldsymbol{y}_s\|^2 ds$$

By Gronwall's inequality, it follows that

$$\mathbb{E}\|\boldsymbol{x}_h - \boldsymbol{y}_h\|^2 \leq \|\boldsymbol{x}_0 - \boldsymbol{y}_0\|^2 e^{2Lh} \leq 2\|\boldsymbol{x}_0 - \boldsymbol{y}_0\|^2, \text{ for } 0 < h < \frac{1}{4L}$$

and

$$\mathbb{E}\|\boldsymbol{z}\|^2 = \left\|\mathbb{E}\left[\int_0^h \boldsymbol{a}(\boldsymbol{x}_s) - \boldsymbol{a}(\boldsymbol{y}_s)ds\right]\right\|^2 \leq \left(\int_0^h \left\|\mathbb{E}\left[\boldsymbol{a}(\boldsymbol{x}_s) - \boldsymbol{a}(\boldsymbol{y}_s)\right]\right\| ds\right)^2$$

$$\leq \int_0^h 1^2 ds \int_0^h \left\|\mathbb{E}\left[\boldsymbol{a}(\boldsymbol{x}_s) - \boldsymbol{a}(\boldsymbol{y}_s)\right]\right\|^2 ds$$

$$\leq h \int_0^h \mathbb{E}\|\boldsymbol{a}(\boldsymbol{x}_s) - \boldsymbol{a}(\boldsymbol{y}_s)\|^2 ds$$

$$\leq L^2 h \int_0^h \mathbb{E}\|\boldsymbol{x}_s - \boldsymbol{y}_s\|^2 ds$$

$$\leq 2L^2 \|\boldsymbol{x}_0 - \boldsymbol{y}_0\|^2 h^2$$

$\square$

## D.2 Growth bound of SDE with additive noise

**Lemma D.2.** *Consider the following SDE with constant diffusion*

$$\begin{cases} d\boldsymbol{x}_t = \boldsymbol{a}(\boldsymbol{x}_t)dt + \boldsymbol{\sigma} d\boldsymbol{W}_t, \\ \boldsymbol{x}(0) = \boldsymbol{x}_0 \end{cases}$$

*where $\boldsymbol{a}(\boldsymbol{x}) \in \mathbb{R}^d$ is $L$-smooth, i.e., $|\boldsymbol{a}(\boldsymbol{y}) - \boldsymbol{a}(\boldsymbol{x})| \leq L|\boldsymbol{y} - \boldsymbol{x}|$, $\boldsymbol{a}(\boldsymbol{0}) = \boldsymbol{0}$ and $\boldsymbol{\sigma} \in \mathbb{R}^{d \times d}$ is a constant matrix independent of time $t$ and $\boldsymbol{x}_t$. Then for $0 < h < \frac{1}{4L}$, we have*

$$\mathbb{E}\|\boldsymbol{x}_h - \boldsymbol{x}_0\|^2 \leq 2.57 \left(\|\boldsymbol{\sigma}\|_F^2 + 2hL^2\|\boldsymbol{x}_0\|^2\right) h.$$

*Proof.* We have

$$\mathbb{E}\|\boldsymbol{x}_h - \boldsymbol{x}_0\|^2 = \mathbb{E}\left\|\int_0^h \boldsymbol{a}(\boldsymbol{x}_t)dt + \int_0^h \boldsymbol{\sigma} d\boldsymbol{W}_t\right\|^2$$

$$\leq 2\mathbb{E}\left\|\int_0^h \boldsymbol{a}(\boldsymbol{x}_t)dt\right\|^2 + 2\mathbb{E}\left\|\int_0^h \boldsymbol{\sigma} d\boldsymbol{W}_t\right\|^2$$

$$\stackrel{(i)}{=} 2\mathbb{E}\left\|\int_0^h \boldsymbol{a}(\boldsymbol{x}_t)dt\right\|^2 + 2\int_0^h \|\boldsymbol{\sigma}\|_F^2 dt$$

$$\leq 2\mathbb{E}\left[\left(\int_0^h \|\boldsymbol{a}(\boldsymbol{x}_t)\| dt\right)^2\right] + 2h\|\boldsymbol{\sigma}\|_F^2$$

$$\leq 2\mathbb{E}\left[\left(\int_0^h \|\boldsymbol{a}(\boldsymbol{x}_t) - \boldsymbol{a}(\boldsymbol{x}_0)\| dt + \int_0^h \|\boldsymbol{a}(\boldsymbol{x}_0)\| dt\right)^2\right] + 2h\|\boldsymbol{\sigma}\|_F^2$$

$$\leq 2\mathbb{E}\left[\left(L\int_0^h \|\boldsymbol{x}_t - \boldsymbol{x}_0\| dt + h\|\boldsymbol{a}(\boldsymbol{x}_0)\|\right)^2\right] + 2h\|\boldsymbol{\sigma}\|_F^2$$

$$\leq 4\mathbb{E}\left[L^2\left(\int_0^h \|\boldsymbol{x}_t - \boldsymbol{x}_0\| dt\right)^2 + h^2\|\boldsymbol{a}(\boldsymbol{x}_0)\|^2\right] + 2h\|\boldsymbol{\sigma}\|_F^2$$

$$\stackrel{(ii)}{\leq} 2h\|\boldsymbol{\sigma}\|_F^2 + 4h^2\|\boldsymbol{a}(\boldsymbol{x}_0)\|^2 + 4L^2 h\int_0^h \mathbb{E}\|\boldsymbol{x}_t - \boldsymbol{x}_0\|^2 dt$$

where $(i)$ is due to Ito's isometry, $(ii)$ is due to Cauchy-Schwarz inequality and $\|\boldsymbol{\sigma}\|_F$ is the Frobenius norm of $\boldsymbol{\sigma}$. By Gronwall's inequality, we obtain

$$\mathbb{E}\|\boldsymbol{x}_h - \boldsymbol{x}_0\|^2 \leq \left(2h\|\boldsymbol{\sigma}\|_F^2 + 4h^2\|\boldsymbol{a}(\boldsymbol{x}_0)\|^2\right)\exp\left\{4L^2h^2\right\}.$$

Since $\|\boldsymbol{a}(\boldsymbol{x}_0)\| = \|\boldsymbol{a}(\boldsymbol{x}_0) - \boldsymbol{a}(\boldsymbol{0})\| \leq L\|\boldsymbol{x}_0\|$, when $0 < h < \frac{1}{4L}$, we finally reach at

$$\mathbb{E}\|\boldsymbol{x}_h - \boldsymbol{x}_0\|^2 \leq 2\left(\|\boldsymbol{\sigma}\|_F^2 + 2hL^2\|\boldsymbol{x}_0\|^2\right)e^{\frac{1}{4}}h \leq 2.57\left(\|\boldsymbol{\sigma}\|_F^2 + 2hL^2\|\boldsymbol{x}_0\|^2\right)h.$$

$\square$

## D.3 LIPSCHITZ CONTINUITY OF THE DRIFT OF HFHR DYNAMICS

**Lemma D.3.** *Assume $\nabla f$ is $L$-Lipschitz, i.e. $\|\nabla f(\boldsymbol{x}) - \nabla f(\boldsymbol{y})\| \leq L\|\boldsymbol{x} - \boldsymbol{y}\|$, then the drift term of HFHR dynamics*

$$\begin{bmatrix} \boldsymbol{p} - \alpha\nabla f(\boldsymbol{q}) \\ -\gamma\boldsymbol{p} - \nabla f(\boldsymbol{q}) \end{bmatrix}$$

*is $L'$-Lipschitz, where $L' \triangleq \sqrt{2}\max\{\sqrt{1+\alpha^2}\max\{\frac{1}{\sqrt{2}}, L\}, \sqrt{1+\gamma^2}\}$. Let $P$ be defined in Appendix A and $\begin{bmatrix} \boldsymbol{\phi} \\ \boldsymbol{\psi} \end{bmatrix} = P\begin{bmatrix} \boldsymbol{q} \\ \boldsymbol{p} \end{bmatrix}$, then $\begin{bmatrix} \boldsymbol{\phi} \\ \boldsymbol{\psi} \end{bmatrix}$ satisfies the following SDE*

$$\begin{bmatrix} d\boldsymbol{\phi} \\ d\boldsymbol{\psi} \end{bmatrix} = P\begin{bmatrix} \boldsymbol{p}(\boldsymbol{\phi}, \boldsymbol{\psi}) - \alpha\nabla f(\boldsymbol{q}(\boldsymbol{\phi}, \boldsymbol{\psi})) \\ -\gamma\boldsymbol{p}(\boldsymbol{\phi}, \boldsymbol{\psi}) - \nabla f(\boldsymbol{q}(\boldsymbol{\phi}, \boldsymbol{\psi})) \end{bmatrix} dt + P\begin{bmatrix} \sqrt{2\alpha}I & 0 \\ 0 & \sqrt{2\gamma}I \end{bmatrix}\begin{bmatrix} d\boldsymbol{W} \\ d\boldsymbol{B} \end{bmatrix}$$

*and the drift term*

$$P\begin{bmatrix} \boldsymbol{p}(\boldsymbol{\phi}, \boldsymbol{\psi}) - \alpha\nabla f(\boldsymbol{q}(\boldsymbol{\phi}, \boldsymbol{\psi})) \\ -\gamma\boldsymbol{p}(\boldsymbol{\phi}, \boldsymbol{\psi}) - \nabla f(\boldsymbol{q}(\boldsymbol{\phi}, \boldsymbol{\psi})) \end{bmatrix}$$

*is $L''$-Lipschitz, where $L'' = \kappa'L'$ and $\kappa'$ is the condition number of $P$.*

*Proof.* By direct computation and Cauchy-Schwarz inequality, we have

$$\left\|\begin{bmatrix} \boldsymbol{p}_1 - \alpha\nabla f(\boldsymbol{q}_1) \\ -\gamma\boldsymbol{p}_1 - \nabla f(\boldsymbol{q}_1) \end{bmatrix} - \begin{bmatrix} \boldsymbol{p}_2 - \alpha\nabla f(\boldsymbol{q}_2) \\ -\gamma\boldsymbol{p}_2 - \nabla f(\boldsymbol{q}_2) \end{bmatrix}\right\|$$

$$= \sqrt{\left\|-\alpha\left(\nabla f(\boldsymbol{q}_1) - \nabla f(\boldsymbol{q}_2)\right) + (\boldsymbol{p}_1 - \boldsymbol{p}_2)\right\|^2 + \left\|-\left(\nabla f(\boldsymbol{q}_1) - \nabla f(\boldsymbol{q}_2)\right) - \gamma(\boldsymbol{p}_1 - \boldsymbol{p}_2)\right\|^2}$$

$$\leq \sqrt{2\alpha^2\|\nabla f(\boldsymbol{q}_1) - \nabla f(\boldsymbol{q}_2)\| + 2\|\boldsymbol{p}_1 - \boldsymbol{p}_2\|^2 + 2\|\nabla f(\boldsymbol{q}_1) - \nabla f(\boldsymbol{q}_2)\| + 2\gamma^2\|\boldsymbol{p}_1 - \boldsymbol{p}_2\|^2}$$

$$\leq \sqrt{(2\alpha^2 L^2 + 2L^2)\|\boldsymbol{q}_1 - \boldsymbol{q}_2\| + (2 + 2\gamma^2)\|\boldsymbol{p}_1 - \boldsymbol{p}_2\|^2}$$

$$\leq \sqrt{2}\max\{L\sqrt{1+\alpha^2}, \sqrt{1+\gamma^2}\}\left\|\begin{bmatrix} \boldsymbol{q}_1 - \boldsymbol{q}_2 \\ \boldsymbol{p}_1 - \boldsymbol{p}_2 \end{bmatrix}\right\|$$

$$\leq \sqrt{2}\max\{\sqrt{1+\alpha^2}\max\{\frac{1}{\sqrt{2}}, L\}, \sqrt{1+\gamma^2}\}\left\|\begin{bmatrix} \boldsymbol{q}_1 - \boldsymbol{q}_2 \\ \boldsymbol{p}_1 - \boldsymbol{p}_2 \end{bmatrix}\right\|$$

$$\triangleq L'\left\|\begin{bmatrix} \boldsymbol{q}_1 - \boldsymbol{q}_2 \\ \boldsymbol{p}_1 - \boldsymbol{p}_2 \end{bmatrix}\right\|$$

By Ito's lemma, we have

$$\begin{bmatrix} d\boldsymbol{\phi} \\ d\boldsymbol{\psi} \end{bmatrix} = P\begin{bmatrix} \boldsymbol{p}(\boldsymbol{\phi}, \boldsymbol{\psi}) - \alpha\nabla f(\boldsymbol{q}(\boldsymbol{\phi}, \boldsymbol{\psi})) \\ -\gamma\boldsymbol{p}(\boldsymbol{\phi}, \boldsymbol{\psi}) - \nabla f(\boldsymbol{q}(\boldsymbol{\phi}, \boldsymbol{\psi})) \end{bmatrix} dt + P\begin{bmatrix} \sqrt{2\alpha}I & 0 \\ 0 & \sqrt{2\gamma}I \end{bmatrix}\begin{bmatrix} d\boldsymbol{W} \\ d\boldsymbol{B} \end{bmatrix}$$

Using the Lipschitz constant obtained for the drift of HFHR, we further have

$$
\left\| P \begin{bmatrix} \boldsymbol{p}(\boldsymbol{\phi}_1, \boldsymbol{\psi}_1) - \alpha \nabla f(\boldsymbol{q}(\boldsymbol{\phi}_1, \boldsymbol{\psi}_1)) \\ -\gamma \boldsymbol{p}(\boldsymbol{\phi}_1, \boldsymbol{\psi}_1) - \nabla f(\boldsymbol{q}(\boldsymbol{\phi}_1, \boldsymbol{\psi}_1)) \end{bmatrix} - P \begin{bmatrix} \boldsymbol{p}(\boldsymbol{\phi}_2, \boldsymbol{\psi}_2) - \alpha \nabla f(\boldsymbol{q}(\boldsymbol{\phi}_2, \boldsymbol{\psi}_2)) \\ -\gamma \boldsymbol{p}(\boldsymbol{\phi}_2, \boldsymbol{\psi}_2) - \nabla f(\boldsymbol{q}(\boldsymbol{\phi}_2, \boldsymbol{\psi}_2)) \end{bmatrix} \right\|
$$

$$
\leq \sigma_{\max} \left\| \begin{bmatrix} \boldsymbol{p}_1 - \alpha \nabla f(\boldsymbol{q}_1) \\ -\gamma \boldsymbol{p}_1 - \nabla f(\boldsymbol{q}_1) \end{bmatrix} - \begin{bmatrix} \boldsymbol{p}_2 - \alpha \nabla f(\boldsymbol{q}_2) \\ -\gamma \boldsymbol{p}_2 - \nabla f(\boldsymbol{q}_2) \end{bmatrix} \right\|
$$

$$
\leq \sigma_{\max} L' \left\| \begin{bmatrix} \boldsymbol{q}_1 - \boldsymbol{q}_2 \\ \boldsymbol{p}_1 - \boldsymbol{p}_2 \end{bmatrix} \right\|
$$

$$
\leq \sigma_{\max} L' \left\| P^{-1} \begin{bmatrix} \boldsymbol{\phi}_1 - \boldsymbol{\phi}_2 \\ \boldsymbol{\psi}_1 - \boldsymbol{\psi}_2 \end{bmatrix} \right\|
$$

$$
\leq \sigma_{\max} L' \frac{1}{\sigma_{\min}} \left\| \begin{bmatrix} \boldsymbol{\phi}_1 - \boldsymbol{\phi}_2 \\ \boldsymbol{\psi}_1 - \boldsymbol{\psi}_2 \end{bmatrix} \right\|
$$

$$
= \kappa' L' \left\| \begin{bmatrix} \boldsymbol{\phi}_1 - \boldsymbol{\phi}_2 \\ \boldsymbol{\psi}_1 - \boldsymbol{\psi}_2 \end{bmatrix} \right\|
$$

where $\sigma_{\max}, \sigma_{\min}$ and $\kappa'$ are the largest, smallest singular values and the condition number (w.r.t. 2-norm) of matrix $P$. $\qquad \square$

**Remark D.4.** *The following inequalities associated with $L'$ will turn out to be useful in many proofs*

$$
L' \geq 1, \ L' \geq \sqrt{2}\gamma, \ L' \geq \sqrt{2}\alpha, \ L \geq \sqrt{2}L \text{ and } L' \geq \sqrt{2}\alpha L.
$$

### D.4 CONTRACTION OF (TRANSFORMED) HFHR DYNAMICS

**Lemma D.5.** *Suppose $f$ is $L$-smooth, $m$-strongly convex and $\gamma^2 > L$. Consider two copies of HFHR dynamics $\begin{bmatrix} \boldsymbol{q}_t \\ \boldsymbol{p}_t \end{bmatrix}$, $\begin{bmatrix} \tilde{\boldsymbol{q}}_t \\ \tilde{\boldsymbol{p}}_t \end{bmatrix}$ (driven by the same Brownian motion) with initialization $\begin{bmatrix} \boldsymbol{q}_0 \\ \boldsymbol{p}_0 \end{bmatrix}$, $\begin{bmatrix} \tilde{\boldsymbol{q}}_0 \\ \tilde{\boldsymbol{p}}_0 \end{bmatrix}$ respectively, then we have*

$$
\left\| P \begin{bmatrix} \boldsymbol{q}_t - \tilde{\boldsymbol{q}}_t \\ \boldsymbol{p}_t - \tilde{\boldsymbol{p}}_t \end{bmatrix} \right\| \leq e^{-\lambda' t} \left\| P \begin{bmatrix} \boldsymbol{q}_0 - \tilde{\boldsymbol{q}}_0 \\ \boldsymbol{p}_0 - \tilde{\boldsymbol{p}}_0 \end{bmatrix} \right\|
$$

*where $P = \begin{bmatrix} \gamma I & \frac{I}{\sqrt{1+\alpha\gamma}}I \\ 0 & \sqrt{1+\alpha\gamma}I \end{bmatrix}$ and $\lambda' = \min\{\frac{m}{\gamma} + \alpha m, \frac{\gamma^2 - L}{\gamma}\}$.*

*Proof.* Consider two copies of HFHR that are driven by the same Brownian motion

$$
\begin{cases} d\boldsymbol{q}_t = (\boldsymbol{p}_t - \alpha \nabla f(\boldsymbol{q}_t))dt + \sqrt{2\alpha}d\boldsymbol{B}_t^1 \\ d\boldsymbol{p}_t = (-\gamma \boldsymbol{p}_t - \nabla f(\boldsymbol{q}_t))dt + \sqrt{2\gamma}d\boldsymbol{B}_t^2 \end{cases}, \quad \begin{cases} d\tilde{\boldsymbol{q}}_t = (\tilde{\boldsymbol{p}}_t - \alpha \nabla f(\tilde{\boldsymbol{q}}_t))dt + \sqrt{2\alpha}d\boldsymbol{B}_t^1 \\ d\tilde{\boldsymbol{p}}_t = (-\gamma \tilde{\boldsymbol{p}}_t - \nabla f(\tilde{\boldsymbol{q}}_t))dt + \sqrt{2\gamma}d\boldsymbol{B}_t^2 \end{cases}.
$$

Based on Taylor's expansion, the difference of the two copies is expressed as

$$
\frac{d}{dt} \begin{bmatrix} \boldsymbol{q}_t - \tilde{\boldsymbol{q}}_t \\ \boldsymbol{p}_t - \tilde{\boldsymbol{p}}_t \end{bmatrix} = - \begin{bmatrix} \alpha H_t & -I \\ H_t & \gamma I \end{bmatrix} \begin{bmatrix} \boldsymbol{q}_t - \tilde{\boldsymbol{q}}_t \\ \boldsymbol{p}_t - \tilde{\boldsymbol{p}}_t \end{bmatrix} \triangleq -A \begin{bmatrix} \boldsymbol{q}_t - \tilde{\boldsymbol{q}}_t \\ \boldsymbol{p}_t - \tilde{\boldsymbol{p}}_t \end{bmatrix}
$$

where $H_t = \int_0^1 \nabla^2 f(\tilde{\boldsymbol{q}}_t + s(\boldsymbol{q} - \tilde{\boldsymbol{q}}_t))ds$. Denote the eigenvalues of $H_t$ by $\eta_i, 1 \leq i \leq d$, by strong convexity and smoothness assumption on $f$, we have $m \leq \eta_i \leq L, 1 \leq i \leq d$.

Denote $\begin{bmatrix} \boldsymbol{\phi}_t \\ \boldsymbol{\psi}_t \end{bmatrix} = P \begin{bmatrix} \boldsymbol{q}_t - \tilde{\boldsymbol{q}}_t \\ \boldsymbol{p}_t - \tilde{\boldsymbol{p}}_t \end{bmatrix}$ and consider $\mathcal{L}_t = \frac{1}{2} \left\| \begin{bmatrix} \boldsymbol{\phi}_t \\ \boldsymbol{\psi}_t \end{bmatrix} \right\|^2$, we have

$$
\begin{aligned}
\frac{d}{dt}\mathcal{L}_t &= -\begin{bmatrix} \boldsymbol{\phi}_t \\ \boldsymbol{\psi}_t \end{bmatrix}^T P A P^{-1} \begin{bmatrix} \boldsymbol{\phi}_t \\ \boldsymbol{\psi}_t \end{bmatrix} \\
&= -\begin{bmatrix} \boldsymbol{\phi}_t \\ \boldsymbol{\psi}_t \end{bmatrix}^T \frac{1}{2}(PAP^{-1} + (P^{-1})^T A^T P^T) \begin{bmatrix} \boldsymbol{\phi}_t \\ \boldsymbol{\psi}_t \end{bmatrix} \\
&= -\begin{bmatrix} \boldsymbol{\phi}_t \\ \boldsymbol{\psi}_t \end{bmatrix}^T \frac{1}{\gamma} \begin{bmatrix} (1+\alpha\gamma)H_t & 0_{d\times d} \\ 0_{d\times d} & \gamma^2 I - H_t \end{bmatrix} \begin{bmatrix} \boldsymbol{\phi}_t \\ \boldsymbol{\psi}_t \end{bmatrix} \\
&\triangleq -\begin{bmatrix} \boldsymbol{\phi}_t \\ \boldsymbol{\psi}_t \end{bmatrix}^T B(\alpha) \begin{bmatrix} \boldsymbol{\phi}_t \\ \boldsymbol{\psi}_t \end{bmatrix}
\end{aligned}
$$

It is easy to see that

$$
\lambda_{\min}(B(\alpha)) = \min_{i=1,2,\cdots,d}\{\min\{\frac{\eta_i}{\gamma} + \alpha\eta_i, \gamma - \frac{\eta_i}{\gamma}\}\} \geq \min\{\frac{m}{\gamma} + \alpha m, \frac{\gamma^2 - L}{\gamma}\} \triangleq \lambda'.
$$

Therefore we have $\frac{d}{dt}\mathcal{L}_t \leq -2\lambda_{\min} B(\alpha)\mathcal{L}_t \leq -2\lambda'\mathcal{L}_t$. By Gronwall's inequality, we obtain

$$
\left\| \begin{bmatrix} \boldsymbol{\phi}_t \\ \boldsymbol{\psi}_t \end{bmatrix} \right\|^2 \leq e^{-2\lambda' t} \left\| \begin{bmatrix} \boldsymbol{\phi}_0 \\ \boldsymbol{\psi}_0 \end{bmatrix} \right\|^2.
$$

and the desired inequality follows by taking square root. □

## D.5 LOCAL ERROR BETWEEN THE EXACT STRANG'S SPLITTING METHOD AND HFHR DYNAMICS

**Lemma D.6.** *Assume $f$ is $L$-smooth and $\boldsymbol{0} \in \arg\min_{\boldsymbol{x}\in\mathbb{R}^d} f(\boldsymbol{x})$, i.e. $\nabla f(\boldsymbol{0}) = \boldsymbol{0}$. If $0 < h \leq \frac{1}{4L'}$, then compared with the HFHR dynamics, the exact Strang's splitting method has local mathematical expectation of deviation of order $p_1 = 2$ and local mean-squared error of order $p_2 = 2$, i.e. there exist constants $\widehat{C}_1, \widehat{C}_2 > 0$ such that*

$$
\left\| \mathbb{E}\boldsymbol{x}(h) - \mathbb{E}\hat{\boldsymbol{x}}(h) \right\| \leq \widehat{C}_1 h^{p_1}
$$

$$
\left( \mathbb{E}\left[ \left\| \boldsymbol{x}(h) - \hat{\boldsymbol{x}}(h) \right\|^2 \right] \right)^{\frac{1}{2}} \leq \widehat{C}_2 h^{p_2}
$$

*where $\boldsymbol{x}(h) = \begin{bmatrix} \boldsymbol{q}(h) \\ \boldsymbol{p}(h) \end{bmatrix}$ is the solution of the HFHR dynamics with initial value $\boldsymbol{x}_0 = \begin{bmatrix} \boldsymbol{q}_0 \\ \boldsymbol{p}_0 \end{bmatrix}$ and $\hat{\boldsymbol{x}}(h) = \begin{bmatrix} \hat{\boldsymbol{q}}(h) \\ \hat{\boldsymbol{p}}(h) \end{bmatrix}$ is the solution of the implementable Strang's splitting with initial value $\boldsymbol{x}_0 = \begin{bmatrix} \boldsymbol{q}_0 \\ \boldsymbol{p}_0 \end{bmatrix}$, $p_1 = 2$ and $p_2 = 2$. More concretely, we have*

$$
\widehat{C}_1 = L\max\{\alpha + 1.25, \gamma + 1\}\left(1.74\|\boldsymbol{x}_0\| + (1.26\sqrt{\alpha} + 2.84\sqrt{\gamma})\sqrt{hd}\right),
$$

$$
\widehat{C}_2 = L\max\{\alpha + 1.25, \gamma + 1\}\left(1.92\|\boldsymbol{x}_0\| + (1.30\sqrt{\alpha} + 3.22\sqrt{\gamma})\sqrt{hd}\right).
$$

*Proof.* The exact Strang's splitting integrator with step size $h$ reads as $\phi^{\frac{h}{2}} \circ \psi^h \circ \phi^{\frac{h}{2}}$ where

$$
\phi : \begin{cases} d\boldsymbol{q} = \boldsymbol{p}dt \\ d\boldsymbol{p} = -\gamma\boldsymbol{p}dt + \sqrt{2\gamma}d\boldsymbol{B} \end{cases} \qquad \psi : \begin{cases} d\boldsymbol{q} = -\alpha\nabla f(\boldsymbol{q})dt + \sqrt{2\alpha}d\boldsymbol{W} \\ d\boldsymbol{p} = -\nabla f(\boldsymbol{q})dt \end{cases} .
$$

The $\phi$ flow can be explicitly solved and the solution is

$$
\begin{cases} \boldsymbol{q}(t) = \boldsymbol{q}_0 + \frac{1-e^{-\gamma t}}{\gamma}\boldsymbol{p}_0 + \sqrt{2\gamma}\int_0^t \frac{1-e^{-\gamma(t-s)}}{\gamma}d\boldsymbol{B}(s) \\ \boldsymbol{p}(t) = e^{-\gamma t}\boldsymbol{p}_0 + \sqrt{2\gamma}\int_0^t e^{-\gamma(t-s)}d\boldsymbol{B}(s) \end{cases} .
$$

The $\psi$ flow can be written as

$$\begin{cases} \boldsymbol{q}(t) = \boldsymbol{q}_0 - \int_0^t \alpha \nabla f(\boldsymbol{q}(s))ds + \sqrt{2\alpha} \int_0^t d\boldsymbol{W}(s) \\ \boldsymbol{p}(t) = \boldsymbol{p}_0 - \int_0^t \nabla f(\boldsymbol{q}(s))ds \end{cases}.$$

The solution of one-step exact Strang's splitting integrator with step size $h$ can be written as

$$\begin{cases} \boldsymbol{q}_3 = \boldsymbol{q}_2(h) + \frac{1-e^{-\gamma\frac{h}{2}}}{\gamma}\boldsymbol{p}_2(h) + \sqrt{2\gamma}\int_{\frac{h}{2}}^h \frac{1-e^{-\gamma(h-s)}}{\gamma}d\boldsymbol{B}(s) \\ \boldsymbol{p}_3 = e^{-\gamma\frac{h}{2}}\boldsymbol{p}_2(h) + \sqrt{2\gamma}\int_{\frac{h}{2}}^h e^{-\gamma(h-s)}d\boldsymbol{B}(s) \\ \boldsymbol{q}_2(r) = \boldsymbol{q}_1 - \int_0^r \alpha\nabla f(\boldsymbol{q}_2(s))ds + \sqrt{2\alpha}\int_0^r d\boldsymbol{W}(s) \qquad (0 \le r \le h) \\ \boldsymbol{p}_2(r) = \boldsymbol{p}_1 - \int_0^r \nabla f(\boldsymbol{q}_2(s))ds \\ \boldsymbol{q}_1 = \boldsymbol{q}_0 + \frac{1-e^{-\gamma\frac{h}{2}}}{\gamma}\boldsymbol{p}_0 + \sqrt{2\gamma}\int_0^{\frac{h}{2}} \frac{1-e^{-\gamma(\frac{h}{2}-s)}}{\gamma}d\boldsymbol{B}(s) \\ \boldsymbol{p}_1 = e^{-\gamma\frac{h}{2}}\boldsymbol{p}_0 + \sqrt{2\gamma}\int_0^{\frac{h}{2}} e^{-\gamma(\frac{h}{2}-s)}d\boldsymbol{B}(s) \end{cases}$$

Therefore, we have $\hat{\boldsymbol{q}}(h) = \boldsymbol{q}_3, \hat{\boldsymbol{p}}(h) = \boldsymbol{p}_3$ and

$$\hat{\boldsymbol{q}}(h) = \sqrt{2\gamma}\int_{\frac{h}{2}}^h \frac{1-e^{-\gamma(h-s)}}{\gamma}d\boldsymbol{B}(s) + \underbrace{\boldsymbol{q}_1 - \int_0^h \alpha\nabla f(\boldsymbol{q}_2(s))ds + \sqrt{2\alpha}\int_0^h d\boldsymbol{W}(s)}_{\boldsymbol{q}_2(h)}$$

$$+ \frac{1-e^{-\gamma\frac{h}{2}}}{\gamma}\left[\underbrace{\boldsymbol{p}_1 - \int_0^h \nabla f(\boldsymbol{q}_2(s))ds}_{\boldsymbol{p}_2(h)}\right]$$

$$= \sqrt{2\gamma}\int_{\frac{h}{2}}^h \frac{1-e^{-\gamma(h-s)}}{\gamma}d\boldsymbol{B}(s) - \int_0^h \alpha\nabla f(\boldsymbol{q}_2(s))ds + \sqrt{2\alpha}\int_0^h d\boldsymbol{W}(s) - \frac{1-e^{-\gamma\frac{h}{2}}}{\gamma}\int_0^h \nabla f(\boldsymbol{q}_2(s))ds$$

$$+ \underbrace{\boldsymbol{q}_0 + \frac{1-e^{-\gamma\frac{h}{2}}}{\gamma}\boldsymbol{p}_0 + \sqrt{2\gamma}\int_0^{\frac{h}{2}} \frac{1-e^{-\gamma(\frac{h}{2}-s)}}{\gamma}d\boldsymbol{B}(s)}_{\boldsymbol{q}_1} + \frac{1-e^{-\gamma\frac{h}{2}}}{\gamma}\left[\underbrace{e^{-\gamma\frac{h}{2}}\boldsymbol{p}_0 + \sqrt{2\gamma}\int_0^{\frac{h}{2}} e^{-\gamma(\frac{h}{2}-s)}d\boldsymbol{B}(s)}_{\boldsymbol{p}_1}\right]$$

$$= \boldsymbol{q}_0 + \frac{1-e^{-\gamma h}}{\gamma}\boldsymbol{p}_0 - \left(\alpha + \frac{1-e^{-\gamma\frac{h}{2}}}{\gamma}\right)\int_0^h \nabla f(\boldsymbol{q}_2(s))ds$$

$$+ \sqrt{2\alpha}\int_0^h d\boldsymbol{W}(s) + \sqrt{2\gamma}\int_{\frac{h}{2}}^h \frac{1-e^{-\gamma(h-s)}}{\gamma}d\boldsymbol{B}(s) + \sqrt{2\gamma}\int_0^{\frac{h}{2}} \frac{1-e^{-\gamma(\frac{h}{2}-s)}}{\gamma}d\boldsymbol{B}(s)$$

$$+ \frac{1-e^{-\gamma\frac{h}{2}}}{\gamma}\sqrt{2\gamma}\int_0^{\frac{h}{2}} e^{-\gamma(\frac{h}{2}-s)}d\boldsymbol{B}(s)$$

$$\hat{\boldsymbol{p}}(h) = e^{-\gamma\frac{h}{2}}\left[\underbrace{\boldsymbol{p}_1 - \int_0^h \nabla f(\boldsymbol{q}_2(s))ds}_{\boldsymbol{p}_2(h)}\right] + \sqrt{2\gamma}\int_{\frac{h}{2}}^h e^{-\gamma(h-s)}d\boldsymbol{B}(s)$$

$$= e^{-\gamma\frac{h}{2}}\left[\underbrace{e^{-\gamma\frac{h}{2}}\boldsymbol{p}_0 + \sqrt{2\gamma}\int_0^{\frac{h}{2}} e^{-\gamma(\frac{h}{2}-s)}d\boldsymbol{B}(s)}_{\boldsymbol{p}_1}\right] - e^{-\gamma\frac{h}{2}}\int_0^h \nabla f(\boldsymbol{q}_2(s))ds + \sqrt{2\gamma}\int_{\frac{h}{2}}^h e^{-\gamma(h-s)}d\boldsymbol{B}(s)$$

$$= e^{-\gamma h}\boldsymbol{p}_0 - e^{-\gamma\frac{h}{2}}\int_0^h \nabla f(\boldsymbol{q}_2(s))ds + e^{-\gamma\frac{h}{2}}\sqrt{2\gamma}\int_0^{\frac{h}{2}} e^{-\gamma(\frac{h}{2}-s)}d\boldsymbol{B}(s) + \sqrt{2\gamma}\int_{\frac{h}{2}}^h e^{-\gamma(h-s)}d\boldsymbol{B}(s)$$

It is clear that $\hat{\boldsymbol{q}}(h), \hat{\boldsymbol{p}}(h)$ should be compared with the exact solution of HFHR at time $h$, which can be written as

$$\boldsymbol{q}(h) = \boldsymbol{q}_0 + \frac{1-e^{-\gamma h}}{\gamma}\boldsymbol{p}_0 - \int_0^h \left(\frac{1-e^{-\gamma(h-s)}}{\gamma} + \alpha\right)\nabla f(\boldsymbol{q}(s))ds + \sqrt{2\alpha}\int_0^h d\boldsymbol{W}_s + \sqrt{2\gamma}\int_0^h \frac{1-e^{-\gamma(h-s)}}{\gamma}d\boldsymbol{B}_s$$

$$\boldsymbol{p}(h) = e^{-\gamma h}\boldsymbol{p}_0 - \int_0^h e^{-\gamma(h-s)}\nabla f(\boldsymbol{q}(s))ds + \sqrt{2\gamma}\int_0^h e^{-\gamma(h-s)}d\boldsymbol{B}(s)$$

Subtracting $\boldsymbol{q}(h), \boldsymbol{p}(h)$ from $\hat{\boldsymbol{q}}(h), \hat{\boldsymbol{p}}(h)$ respectively, we obtain

$$\hat{\boldsymbol{q}}(h) - \boldsymbol{q}(h) = -\left(\alpha + \frac{1-e^{-\gamma\frac{h}{2}}}{\gamma}\right)\int_0^h \nabla f(\boldsymbol{q}_2(s)) - \nabla f(\boldsymbol{q}(s))ds$$
$$+ \int_0^h \left(\frac{1-e^{-\gamma(h-s)}}{\gamma} - \frac{1-e^{-\gamma\frac{h}{2}}}{\gamma}\right)\nabla f(\boldsymbol{q}(s))ds$$

$$\hat{\boldsymbol{p}}(h) - \boldsymbol{p}(h) = -e^{-\gamma\frac{h}{2}}\int_0^h \nabla f(\boldsymbol{q}_2(s)) - \nabla f(\boldsymbol{q}(s))ds + \int_0^h \left(e^{-\gamma(h-s)} - e^{-\gamma\frac{h}{2}}\right)\nabla f(\boldsymbol{q}(s))ds$$

It should be clear now that we will need to bound the term $\nabla f(\boldsymbol{q}_2) - \nabla f(\boldsymbol{q})$ and $\nabla f(\boldsymbol{q})$. Since

$$\boldsymbol{q}_2(r) = \boldsymbol{q}_0 + \frac{1-e^{-\gamma\frac{h}{2}}}{\gamma}\boldsymbol{p}_0 + \sqrt{2\gamma}\int_0^{\frac{h}{2}} \frac{1-e^{-\gamma(\frac{h}{2}-s)}}{\gamma}d\boldsymbol{B}(s) - \alpha\int_0^r \nabla f(\boldsymbol{q}_2(s))ds + \sqrt{2\alpha}\int_0^r d\boldsymbol{W}(s)$$

$$\boldsymbol{q}(r) = \boldsymbol{q}_0 + \frac{1-e^{-\gamma r}}{\gamma}\boldsymbol{p}_0 - \int_0^r \left(\frac{1-e^{-\gamma(r-s)}}{\gamma} + \alpha\right)\nabla f(\boldsymbol{q}(s))ds + \sqrt{2\alpha}\int_0^r d\boldsymbol{W}(s)$$
$$+ \sqrt{2\gamma}\int_0^r \frac{1-e^{-\gamma(r-s)}}{\gamma}d\boldsymbol{B}(s),$$

we then have

$$\boldsymbol{q}_2(r) - \boldsymbol{q}(r) = \frac{e^{-\gamma r} - e^{-\gamma\frac{h}{2}}}{\gamma}\boldsymbol{p}_0 - \alpha\int_0^r \nabla f(\boldsymbol{q}_2(s)) - \nabla f(\boldsymbol{q}(s))ds + \int_0^r \frac{1-e^{-\gamma(r-s)}}{\gamma}\nabla f(\boldsymbol{q}(s))ds$$
$$+ \sqrt{2\gamma}\int_0^{\frac{h}{2}} \frac{1-e^{-\gamma(\frac{h}{2}-s)}}{\gamma}d\boldsymbol{B}(s) - \sqrt{2\gamma}\int_0^r \frac{1-e^{-\gamma(r-s)}}{\gamma}d\boldsymbol{B}(s)$$

By Lemma D.3 and D.2, when $0 < h < \frac{1}{4L'}$, we have the following for the solution of HFHR dynamics

$$\mathbb{E}[\|\boldsymbol{x}_{0,\boldsymbol{x}_0}(h) - \boldsymbol{x}_0\|^2] \le \widehat{C}_0 h$$

where $\widehat{C}_0 = 5.14\left\{(\alpha+\gamma)d + h\left(L'\right)^2\|\boldsymbol{x}_0\|^2\right\}$ and hence

$$\mathbb{E}\left[\int_0^r \|\nabla f(\boldsymbol{q}(s))\|^2 ds\right] \le \mathbb{E}\left[2\int_0^r \|\nabla f(\boldsymbol{q}(0))\|^2 ds + 2\int_0^r \|\nabla f(\boldsymbol{q}(s)) - \nabla f(\boldsymbol{q}(0))\|^2 ds\right]$$
$$\le \mathbb{E}\left[2L^2 r\|\boldsymbol{q}(0)\|^2 + 2L^2\int_0^r \|\boldsymbol{q}(s) - \boldsymbol{q}(0)\|^2 ds\right]$$
$$\le 2L^2 r\|\boldsymbol{x}_0\|^2 + 2L^2\mathbb{E}\left[\int_0^r \|\boldsymbol{q}(s) - \boldsymbol{q}(0)\|^2 ds\right]$$
$$\le 2L^2 r\|\boldsymbol{x}_0\|^2 + 2L^2\widehat{C}_0\int_0^r s\,ds$$
$$\le L^2 r\left(2\|\boldsymbol{x}_0\|^2 + h\widehat{C}_0\right)$$
$$\le L^2 r\left(2.33\|\boldsymbol{x}_0\|^2 + 5.14(\alpha+\gamma)dh\right) \tag{12}$$

Now $\mathbb{E}\left[\|\boldsymbol{q}_2 - \boldsymbol{q}\|^2\right]$ can be bounded as follow

$$\mathbb{E}\left[\left\|\boldsymbol{q}_2(r) - \boldsymbol{q}(r)\right\|^2\right]$$

$$\leq 5\left\{\left(\frac{e^{-\gamma r} - e^{-\gamma\frac{h}{2}}}{\gamma}\right)^2 \|\boldsymbol{p}_0\|^2 + \alpha^2\mathbb{E}\left\|\int_0^r \nabla f(\boldsymbol{q}_2(s)) - \nabla f(\boldsymbol{q}(s))ds\right\|^2 + \mathbb{E}\left\|\int_0^r \frac{1 - e^{-\gamma(r-s)}}{\gamma}\nabla f(\boldsymbol{q}(s))ds\right\|^2\right\}$$

$$+ 5\left\{2\gamma\mathbb{E}\left\|\int_0^{\frac{h}{2}} \frac{1 - e^{-\gamma(\frac{h}{2}-s)}}{\gamma}d\boldsymbol{B}(s)\right\|^2 + 2\gamma\mathbb{E}\left\|\int_0^r \frac{1 - e^{-\gamma(r-s)}}{\gamma}d\boldsymbol{B}(s)\right\|^2\right\} \qquad \text{(Cauchy-Schwartz Inequality)}$$

$$\leq 5\left\{\frac{h^2}{4}\|\boldsymbol{x}_0\|^2 + \alpha^2 L^2 r \int_0^r \mathbb{E}\left\|\boldsymbol{q}_2(s) - \boldsymbol{q}(s)\right\|^2 ds + \int_0^r \left(\frac{1 - e^{-\gamma(r-s)}}{\gamma}\right)^2 ds \int_0^r \mathbb{E}\left\|\nabla f(\boldsymbol{q}(s))\right\|^2 ds\right\}$$

$$+ 5\left\{\frac{\gamma dh^3}{12} + \frac{2\gamma d}{3}r^3\right\}$$

$$\leq 5\left\{\frac{h^2}{4}\|\boldsymbol{x}_0\|^2 + \alpha^2 L^2 r \int_0^r \mathbb{E}\left\|\boldsymbol{q}_2(s) - \boldsymbol{q}(s)\right\|^2 ds + \frac{h^3}{3}\mathbb{E}\left[\int_0^r \left\|\nabla f(\boldsymbol{q}(s))\right\|^2\right] + \frac{3\gamma d}{4}h^3\right\}$$

$$\leq 5\left\{\frac{h^2}{4}\|\boldsymbol{x}_0\|^2 + \frac{3\gamma d}{4}h^3 + \frac{h^3}{3}L^2\left(2.33\|\boldsymbol{x}_0\|^2 + 5.14(\alpha + \gamma)dh\right)r + \alpha^2 L^2 r \int_0^r \mathbb{E}\left\|\boldsymbol{q}_2(s) - \boldsymbol{q}(s)\right\|^2 ds\right\}$$

$$\leq 5h^2\left\{\frac{1}{4}\|\boldsymbol{x}_0\|^2 + \frac{3\gamma d}{4}h + \frac{h^2}{3}L^2\left(2.33\|\boldsymbol{x}_0\|^2 + 5.14(\alpha + \gamma)dh\right)\right\} + 5\alpha^2 L^2 h \int_0^r \mathbb{E}\left\|\boldsymbol{q}_2(s) - \boldsymbol{q}(s)\right\|^2 ds$$

By Gronwall's inequality and $0 < h \leq \frac{1}{4L'}$, we have

$$\mathbb{E}\left[\left\|\boldsymbol{q}_2(r) - \boldsymbol{q}(r)\right\|^2\right] \leq 5h^2\left\{\frac{1}{4}\|\boldsymbol{x}_0\|^2 + \frac{3\gamma d}{4}h + \frac{h^2}{3}L^2\left(2.33\|\boldsymbol{x}_0\|^2 + 5.14(\alpha + \gamma)dh\right)\right\}\exp\{5\alpha^2 L^2 h^2\}$$

$$\leq 5h^2\left\{\frac{1}{4}\|\boldsymbol{x}_0\|^2 + \frac{3\gamma d}{4}h + \frac{h^2}{3}L^2\left(2.33\|\boldsymbol{x}_0\|^2 + 5.14(\alpha + \gamma)dh\right)\right\}e^{\frac{5}{32}}$$

$$\leq 5.85h^2\left\{0.28\|\boldsymbol{x}_0\|^2 + (0.06\alpha + 0.81\gamma)hd\right\}$$

$$\leq h^2\left\{1.64\|\boldsymbol{x}_0\|^2 + (0.36\alpha + 4.74\gamma)hd\right\}. \tag{13}$$

With bounds in Equation (12) and (13), we are now ready to show $p_1$ and $p_2$. For $p_1$, i.e. the order of the mathematical expectation of deviation, we have

$$
\left\| \mathbb{E} \left[ \begin{bmatrix} \hat{\boldsymbol{q}}(h) \\ \hat{\boldsymbol{p}}(h) \end{bmatrix} - \begin{bmatrix} \boldsymbol{q}(h) \\ \boldsymbol{p}(h) \end{bmatrix} \right] \right\|
$$

$$
\leq \left\| \mathbb{E} \left[ \hat{\boldsymbol{q}}(h) - \boldsymbol{q}(h) \right] \right\| + \left\| \mathbb{E} \left[ \hat{\boldsymbol{p}}(h) - \boldsymbol{p}(h) \right] \right\|
$$

$$
\leq \left( \alpha + \frac{1 - e^{-\gamma \frac{h}{2}}}{\gamma} \right) \left\| \int_0^h \mathbb{E} \left[ \nabla f(\boldsymbol{q}_2(s)) - \nabla f(\boldsymbol{q}(s)) \right] ds \right\| + \left\| \int_0^h \left( \frac{1 - e^{-\gamma(h-s)}}{\gamma} - \frac{1 - e^{-\gamma \frac{h}{2}}}{\gamma} \right) \mathbb{E} \left[ \nabla f(\boldsymbol{q}(s)) \right] ds \right\|
$$

$$
+ e^{-\gamma \frac{h}{2}} \left\| \int_0^{\frac{h}{2}} \mathbb{E} \left[ \nabla f(\boldsymbol{q}_2(s)) - \nabla f(\boldsymbol{q}(s)) \right] ds \right\| + \left\| \int_0^h \left( e^{-\gamma(h-s)} - e^{-\gamma \frac{h}{2}} \right) \mathbb{E} \left[ \nabla f(\boldsymbol{q}(s)) \right] ds \right\|
$$

$$
\leq \left( \alpha + 1 + \frac{h}{2} \right) L \int_0^h \mathbb{E} \left\| \boldsymbol{q}_2(s) - \boldsymbol{q}(s) \right\| ds
$$

$$
+ \int_0^h \left( \left| \frac{1 - e^{-\gamma(h-s)}}{\gamma} - \frac{1 - e^{-\gamma \frac{h}{2}}}{\gamma} \right| + \left| e^{-\gamma(h-s)} - e^{-\gamma \frac{h}{2}} \right| \right) \left\| \mathbb{E} \left[ \nabla f(\boldsymbol{q}(s)) \right] \right\| ds
$$

$$
\leq L \left( \alpha + 1 + \frac{h}{2} \right) \int_0^h \mathbb{E} \left\| \boldsymbol{q}_2(s) - \boldsymbol{q}(s) \right\| ds
$$

$$
+ \left\{ \left( \int_0^h \left| \frac{1 - e^{-\gamma(h-s)}}{\gamma} - \frac{1 - e^{-\gamma \frac{h}{2}}}{\gamma} \right|^2 ds \right)^{\frac{1}{2}} + \left( \int_0^h \left| e^{-\gamma(h-s)} - e^{-\gamma \frac{h}{2}} \right|^2 ds \right)^{\frac{1}{2}} \right\} \left( \int_0^h \left\| \mathbb{E} \left[ \nabla f(\boldsymbol{q}(s)) \right] \right\|^2 ds \right)^{\frac{1}{2}}
$$

$$
\leq L \left( \alpha + 1 + \frac{h}{2} \right) \int_0^h \left( \mathbb{E} \left\| \boldsymbol{q}_2(s) - \boldsymbol{q}(s) \right\|^2 \right)^{\frac{1}{2}} ds + \frac{1 + \gamma}{2\sqrt{3}} h^{\frac{3}{2}} \left( \mathbb{E} \int_0^h \left\| \left[ \nabla f(\boldsymbol{q}(s)) \right] \right\|^2 ds \right)^{\frac{1}{2}}
$$

$$
\leq L \left( \alpha + 1 + \frac{h}{2} \right) h^2 \left\{ 1.64 \|\boldsymbol{x}_0\|^2 + (0.36\alpha + 4.74\gamma)hd \right\}^{\frac{1}{2}} + \frac{1 + \gamma}{2\sqrt{3}} h^2 L \left( 2.33 \|\boldsymbol{x}_0\|^2 + 5.14(\alpha + \gamma)dh \right)^{\frac{1}{2}}
$$

$$
\leq L \left( \alpha + 1.25 \right) h^2 \left( 1.29 \|\boldsymbol{x}_0\| + \sqrt{0.36\alpha + 4.74\gamma}\sqrt{hd} \right) + (1 + \gamma)h^2 L \left( 0.45 \|\boldsymbol{x}_0\| + \sqrt{0.43\alpha + 0.43\gamma}\sqrt{dh} \right)
$$

$$
\leq L h^2 \max\{\alpha + 1.25, \gamma + 1\} \left( 1.74 \|\boldsymbol{x}_0\| + (1.26\sqrt{\alpha} + 2.84\sqrt{\gamma})\sqrt{hd} \right)
$$

The above derivation proves $p_1 = 2$ with

$$
\widehat{C}_1 = L \max\{\alpha + 1.25, \gamma + 1\} \left( 1.74 \|\boldsymbol{x}_0\| + (1.26\sqrt{\alpha} + 2.84\sqrt{\gamma})\sqrt{hd} \right).
$$

We now proceed with $p_2$, i.e. mean-square error

$$\mathbb{E}\left\|\begin{bmatrix}\hat{\boldsymbol{q}}(h)\\\hat{\boldsymbol{p}}(h)\end{bmatrix}-\begin{bmatrix}\boldsymbol{q}(h)\\\boldsymbol{p}(h)\end{bmatrix}\right\|^2$$

$$\leq 2\left(\alpha+\frac{h}{2}\right)^2\mathbb{E}\left\|\int_0^h\nabla f(\boldsymbol{q}_2(s))-\nabla f(\boldsymbol{q}(s))ds\right\|^2+2\mathbb{E}\left\|\int_0^h\left(\frac{1-e^{-\gamma(h-s)}}{\gamma}-\frac{1-e^{-\gamma\frac{h}{2}}}{\gamma}\right)\nabla f(\boldsymbol{q}(s))ds\right\|^2$$

$$+2\mathbb{E}\left\|\int_0^h\nabla f(\boldsymbol{q}_2(s))-\nabla f(\boldsymbol{q}(s))ds\right\|^2+2\mathbb{E}\left\|\int_0^h\left(e^{-\gamma(h-s)}-e^{-\gamma\frac{h}{2}}\right)\nabla f(\boldsymbol{q}(s))ds\right\|^2$$

$$\leq 2\left((\alpha+\frac{h}{2})^2+1\right)L^2\mathbb{E}\left(\int_0^h|\boldsymbol{q}_2(s)-\boldsymbol{q}(s)|ds\right)^2+2\int_0^h\left|\frac{1-e^{-\gamma(h-s)}}{\gamma}-\frac{1-e^{-\gamma\frac{h}{2}}}{\gamma}\right|^2ds\int_0^h\mathbb{E}\|\nabla f(\boldsymbol{q}(s))\|^2ds$$

$$+2\int_0^h\left|e^{-\gamma(h-s)}-e^{-\gamma\frac{h}{2}}\right|^2ds\int_0^h\mathbb{E}\|\nabla f(\boldsymbol{q}(s))\|^2ds$$

$$\leq 2\left((\alpha+\frac{h}{2})^2+1\right)L^2h\int_0^h\mathbb{E}|\boldsymbol{q}_2(s)-\boldsymbol{q}(s)|^2ds+\frac{1+\gamma^2}{6}h^3\int_0^h\mathbb{E}|\nabla f(\boldsymbol{q}(s))|^2ds$$

$$\leq 2\left((\alpha+\frac{h}{2})^2+1\right)L^2\left\{1.64\|\boldsymbol{x}_0\|^2+(0.36\alpha+4.74\gamma)hd\right\}h^4+\frac{1+\gamma^2}{6}L^2\left\{2.33\|\boldsymbol{x}_0\|^2+5.14(\alpha+\gamma)hd\right\}h^4$$

$$\leq L^2\max\{(\alpha+1.25)^2,1+\gamma^2\}\left(3.67\|\boldsymbol{x}_0\|^2+(1.68\alpha+10.34\gamma)hd\right)h^4$$

The above derivation implies $p_2=2$ with

$$\widehat{C}_2=L\max\{\alpha+1.25,1+\gamma\}\left(1.92\|\boldsymbol{x}_0\|+(1.30\sqrt{\alpha}+3.22\sqrt{\gamma})\sqrt{hd}\right).$$

$\square$

### D.6 LOCAL ERROR BETWEEN ALGORITHM 1 AND THE EXACT STRANG'S SPLITTING METHOD

**Lemma D.7.** *Assume $f$ is $L$-smooth, $\boldsymbol{0}\in\arg\min_{\boldsymbol{x}\in\mathbb{R}^d}f(\boldsymbol{x})$, i.e. $\nabla f(\boldsymbol{0})=\boldsymbol{0}$ and the operator $\nabla\Delta f$ grows at most linearly, i.e. $\|\nabla\Delta f(\boldsymbol{q})\|\leq G\sqrt{1+\|\boldsymbol{q}\|^2}$. If $0<h\leq\frac{1}{4L'}$, then compared with the exact Strang's splitting method of HFHR dynamics, the implementable Strang's splitting method has local mathematical expectation of deviation of order $p_1=2$ and local mean-squared error of order $p_2=1.5$, i.e. there exist constants $\bar{C}_1,\bar{C}_2>0$ such that*

$$\|\mathbb{E}\hat{\boldsymbol{x}}(h)-\mathbb{E}\bar{\boldsymbol{x}}(h)\|\leq\bar{C}_1h^{p_1}$$

$$\left(\mathbb{E}\left[\|\hat{\boldsymbol{x}}(h)-\bar{\boldsymbol{x}}(h)\|^2\right]\right)^{\frac{1}{2}}\leq\bar{C}_2h^{p_2}$$

*where $\hat{\boldsymbol{x}}(h)=\begin{bmatrix}\hat{\boldsymbol{q}}(h)\\\hat{\boldsymbol{p}}(h)\end{bmatrix}$ is the solution of the exact Strang's splitting method for HFHR with initial value $\boldsymbol{x}_0=\begin{bmatrix}\boldsymbol{q}_0\\\boldsymbol{p}_0\end{bmatrix}$ and $\bar{\boldsymbol{x}}(h)=\begin{bmatrix}\bar{\boldsymbol{q}}(h)\\\bar{\boldsymbol{p}}(h)\end{bmatrix}$ is the one-step result of Algorithm 1 with initial value $\boldsymbol{x}_0=\begin{bmatrix}\boldsymbol{q}_0\\\boldsymbol{p}_0\end{bmatrix}$, $p_1=2$ and $p_2=1.5$. More concretely, we have*

$$\bar{C}_1=\alpha(\alpha+1.125)(L+G)\left[0.5+0.71\|\boldsymbol{x}_0\|+(1.14\sqrt{\alpha}+0.21\sqrt{\gamma}h)\sqrt{hd}\right]$$

*and*

$$\bar{C}_2=L(\alpha+0.73)\left(2.30\sqrt{h}\alpha L\|\boldsymbol{x}_0\|+(2.27\sqrt{\alpha}+0.12\sqrt{\gamma}h)\sqrt{d}\right).$$

*Proof.* The solution of one-step exact Strang's splitting integrator with step size $h$ can be written as

$$\begin{cases} \boldsymbol{q}_3 = \boldsymbol{q}_2(h) + \frac{1-e^{-\gamma\frac{h}{2}}}{\gamma}\boldsymbol{p}_2(h) + \sqrt{2\gamma}\int_{\frac{h}{2}}^h \frac{1-e^{-\gamma(h-s)}}{\gamma}d\boldsymbol{B}(s) \\ \boldsymbol{p}_3 = e^{-\gamma\frac{h}{2}}\boldsymbol{p}_2(h) + \sqrt{2\gamma}\int_{\frac{h}{2}}^h e^{-\gamma(h-s)}d\boldsymbol{B}(s) \\ \boldsymbol{q}_2(r) = \boldsymbol{q}_1 - \int_0^r \alpha\nabla f(\boldsymbol{q}_2(s))ds + \sqrt{2\alpha}\int_0^r d\boldsymbol{W}(s) \qquad (0 \le r \le h) \\ \boldsymbol{p}_2(r) = \boldsymbol{p}_1 - \int_0^r \nabla f(\boldsymbol{q}_2(s))ds \\ \boldsymbol{q}_1 = \boldsymbol{q}_0 + \frac{1-e^{-\gamma\frac{h}{2}}}{\gamma}\boldsymbol{p}_0 + \sqrt{2\gamma}\int_0^{\frac{h}{2}} \frac{1-e^{-\gamma(\frac{h}{2}-s)}}{\gamma}d\boldsymbol{B}(s) \\ \boldsymbol{p}_1 = e^{-\gamma\frac{h}{2}}\boldsymbol{p}_0 + \sqrt{2\gamma}\int_0^{\frac{h}{2}} e^{-\gamma(\frac{h}{2}-s)}d\boldsymbol{B}(s) \end{cases}$$

and the solution of one-step implementable Strang's splitting integrator with step size $h$ can be written as

$$\begin{cases} \bar{\boldsymbol{q}}_3 = \bar{\boldsymbol{q}}_2(h) + \frac{1-e^{-\gamma\frac{h}{2}}}{\gamma}\bar{\boldsymbol{p}}_2(h) + \sqrt{2\gamma}\int_0^{\frac{h}{2}} \frac{1-e^{-\gamma(\frac{h}{2}-s)}}{\gamma}d\boldsymbol{B}(\frac{h}{2}+s) \\ \bar{\boldsymbol{p}}_3 = e^{-\gamma\frac{h}{2}}\bar{\boldsymbol{p}}_2(h) + \sqrt{2\gamma}\int_0^{\frac{h}{2}} e^{-\gamma(\frac{h}{2}-s)}d\boldsymbol{B}(\frac{h}{2}+s) \\ \bar{\boldsymbol{q}}_2(r) = \boldsymbol{q}_1 - \int_0^r \alpha\nabla f(\boldsymbol{q}_1)ds + \sqrt{2\alpha}\int_0^r d\boldsymbol{W}(s) \qquad (0 \le r \le h) \\ \bar{\boldsymbol{p}}_2(r) = \boldsymbol{p}_1 - \int_0^r \nabla f(\boldsymbol{q}_1)ds \\ \boldsymbol{q}_1 = \boldsymbol{q}_0 + \frac{1-e^{-\gamma\frac{h}{2}}}{\gamma}\boldsymbol{p}_0 + \sqrt{2\gamma}\int_0^{\frac{h}{2}} \frac{1-e^{-\gamma(\frac{h}{2}-s)}}{\gamma}d\boldsymbol{B}(s) \\ \boldsymbol{p}_1 = e^{-\gamma\frac{h}{2}}\boldsymbol{p}_0 + \sqrt{2\gamma}\int_0^{\frac{h}{2}} e^{-\gamma(\frac{h}{2}-s)}d\boldsymbol{B}(s) \end{cases}$$

Note that in the implementable Strang's splitting method, $\phi$ flow can be explicitly integrated and hence $\boldsymbol{q}_1, \boldsymbol{p}_1$ are the same as that in the exact Strang's splitting method.

First, we will bound the deviation of mathematical expectation and mean squared error of $\boldsymbol{q}_2(h) - \bar{\boldsymbol{q}}_2(h)$ and $\boldsymbol{p}_2(h) - \bar{\boldsymbol{p}}_2(h)$. We have

$$\begin{cases} \boldsymbol{q}_2(h) - \bar{\boldsymbol{q}}_2(h) = & -\alpha\int_0^h \nabla f(\boldsymbol{q}_2(s)) - \nabla f(\boldsymbol{q}_1)ds \\ \boldsymbol{p}_2(h) - \bar{\boldsymbol{p}}_2(h) = & -\int_0^h \nabla f(\boldsymbol{q}_2(s)) - \nabla f(\boldsymbol{q}_1)ds \end{cases} \tag{14}$$

Square both sides of the first equation in (14) and take expectation, we obtain

$$\mathbb{E}\big\|\boldsymbol{q}_2(h) - \bar{\boldsymbol{q}}_2(h)\big\|^2 = \alpha^2 \mathbb{E}\left\|\int_0^h \nabla f(\boldsymbol{q}_2(s)) - \nabla f(\boldsymbol{q}_1)ds\right\|^2$$

$$\le \alpha^2 \mathbb{E}\left(\int_0^h \big\|\nabla f(\boldsymbol{q}_2(s)) - \nabla f(\boldsymbol{q}_1)\big\|\, ds\right)^2$$

$$\le \alpha^2 L^2 \mathbb{E}\left(\int_0^h \big\|\boldsymbol{q}_2(s) - \boldsymbol{q}_1\big\|\, ds\right)^2$$

$$\le \alpha^2 L^2 h \int_0^h \mathbb{E}\big\|\boldsymbol{q}_2(s) - \boldsymbol{q}_1\big\|^2 ds$$

Note that $\boldsymbol{q}_2$ is the solution of a rescaled overdamped Langevin dynamics whose drift vector field is $\alpha L$-Lipschitz, by conditional expectation version of Lemma D.2, for $0 < h < \frac{1}{4L'} < \frac{1}{4\alpha L}$, we have $\mathbb{E}\big\|\boldsymbol{q}_2(h) - \boldsymbol{q}_1\big\|^2 \le \bar{C}_0 h$ with $\bar{C}_0 = 5.14\left\{\alpha d + h(\alpha L)^2 \mathbb{E}\|\boldsymbol{q}_1\|^2\right\}$ and it follows that

$$\begin{cases} \mathbb{E}\big\|\boldsymbol{q}_2(h) - \bar{\boldsymbol{q}}_2(h)\big\|^2 \le & \alpha^2 L^2 \bar{C}_0 h^3 \\ \mathbb{E}\big\|\boldsymbol{p}_2(h) - \bar{\boldsymbol{p}}_2(h)\big\|^2 \le & L^2 \bar{C}_0 h^3. \end{cases}$$

Now consider $p_1$, i.e., the deviation of mathematical expectation. By Ito's lemma, we have

$$\boldsymbol{q}_2(h) - \bar{\boldsymbol{q}}_2(h)$$

$$= -\alpha\int_0^h \nabla f(\boldsymbol{q}_2(s)) - \nabla f(\boldsymbol{q}_1)ds$$

$$= -\alpha\int_0^h \left[\int_0^s -\alpha\nabla^2 f(\boldsymbol{q}_2(r))\nabla f(\boldsymbol{q}_2(r))dr + \alpha\int_0^s \nabla\Delta f(\boldsymbol{q}_2(r))dr + \rho\right]ds \tag{15}$$

where $\rho$ is a stochastic integral term. Take expectation and norm for Equation (15), we have

$$\left\| \mathbb{E}\left[ \boldsymbol{q}_2(h) - \bar{\boldsymbol{q}}_2(h) \right] \right\|$$

$$= \alpha^2 \left\| \int_0^h \mathbb{E}\left[ \int_0^s \nabla^2 f(\boldsymbol{q}_2(r)) \nabla f(\boldsymbol{q}_2(r)) dr - \int_0^s \nabla \Delta f(\boldsymbol{q}_2(r)) dr \right] ds \right\|$$

$$\leq \alpha^2 \int_0^h \mathbb{E}\left[ \int_0^s \left\| \nabla^2 f(\boldsymbol{q}_2(r)) \right\|_2 \left\| \nabla f(\boldsymbol{q}_2(r)) \right\| dr + \int_0^s \left\| \nabla \Delta f(\boldsymbol{q}_2(r)) \right\| dr \right] ds$$

$$\leq \alpha^2 \int_0^h \mathbb{E}\left[ L \int_0^s \left\| \boldsymbol{q}_2(r) \right\| dr + \int_0^s G(1 + \left\| \boldsymbol{q}_2(r) \right\|) dr \right] ds$$

$$= \alpha^2 (L + G) \int_0^h \int_0^s \mathbb{E}\left\| \boldsymbol{q}_2(r) \right\| dr + \alpha^2 G \frac{h^2}{2}$$

$$\leq \alpha^2 (L + G) \int_0^h \int_0^s \mathbb{E}\left\| \boldsymbol{q}_2(r) - \boldsymbol{q}_1 \right\| + \mathbb{E}\left\| \boldsymbol{q}_1 \right\| dr + \alpha^2 G \frac{h^2}{2}$$

$$\leq \alpha^2 (L + G) \int_0^h \int_0^s \sqrt{\mathbb{E}\left\| \boldsymbol{q}_2(r) - \boldsymbol{q}_1 \right\|^2} + \mathbb{E}\left\| \boldsymbol{q}_1 \right\| dr + \alpha^2 G \frac{h^2}{2}$$

$$\leq \alpha^2 (L + G) \sqrt{\bar{C}_0 h} \frac{h^2}{2} + \alpha^2 (L + G) \frac{h^2}{2} \mathbb{E}\left\| \boldsymbol{q}_1 \right\| + \alpha^2 G \frac{h^2}{2}$$

$$\leq \alpha^2 \left\{ \frac{\sqrt{\bar{C}_0 h} + \mathbb{E}\left\| \boldsymbol{q}_1 \right\|}{2} (L + G) + \frac{G}{2} \right\} h^2$$

$$\leq \frac{1}{2} \alpha^2 (L + G) \left\{ \sqrt{\bar{C}_0 h} + \mathbb{E}\left\| \boldsymbol{q}_1 \right\| + 1 \right\} h^2$$

Similarly, we have $\left\| \mathbb{E}\left[ \boldsymbol{p}_2(h) - \bar{\boldsymbol{p}}_2(h) \right] \right\| \leq \frac{1}{2} \alpha(L + G) \left\{ \sqrt{\bar{C}_0 h} + \mathbb{E}\left\| \boldsymbol{q}_1 \right\| + 1 \right\} h^2$.

For $p_2$, i.e., mean-square error, we have

$$\mathbb{E}\left\| \boldsymbol{q}_2(h) - \bar{\boldsymbol{q}}_2(h) \right\|^2 \leq \alpha^2 \mathbb{E}\left\{ \int_0^h \left\| \nabla f(\boldsymbol{q}_2(s)) - \nabla f(\boldsymbol{q}_1) \right\| ds \right\}^2$$

$$\leq \alpha^2 \mathbb{E}\left\{ \int_0^h 1 ds \int_0^h \left\| \nabla f(\boldsymbol{q}_2(s)) - \nabla f(\boldsymbol{q}_1) \right\|^2 ds \right\}$$

$$\leq \alpha^2 L^2 h \int_0^h \mathbb{E}\left\| \boldsymbol{q}_2(s) - \boldsymbol{q}_1 \right\|^2 ds$$

$$\leq \frac{\alpha^2 L^2 \bar{C}_0}{2} h^3$$

Similarly we obtain $\mathbb{E}\left\| \boldsymbol{p}_2(h) - \bar{\boldsymbol{p}}_2(h) \right\|^2 \leq \frac{L^2 \bar{C}_0}{2} h^3$. Recall

$$\begin{cases} \boldsymbol{q}_3 - \bar{\boldsymbol{q}}_3 = \boldsymbol{q}_2(h) - \bar{\boldsymbol{q}}_2(h) + \frac{1 - e^{-\gamma \frac{h}{2}}}{\gamma} (\boldsymbol{p}_2(h) - \bar{\boldsymbol{p}}_2(h)) \\ \boldsymbol{p}_3 - \bar{\boldsymbol{p}}_3 = e^{-\gamma \frac{h}{2}} (\boldsymbol{p}_2(h) - \bar{\boldsymbol{p}}_2(h)) \end{cases}.$$

and it follows that when $0 < h \leq \frac{1}{4L'} < 1$

$$\left\| \mathbb{E}\begin{bmatrix} \boldsymbol{q}_3 - \bar{\boldsymbol{q}}_3 \\ \boldsymbol{p}_3 - \bar{\boldsymbol{p}}_3 \end{bmatrix} \right\| \leq \alpha(\alpha + 1 + \frac{h}{2})(L + G) \frac{\sqrt{\bar{C}_0 h} + \mathbb{E}\left\| \boldsymbol{q}_1 \right\| + 1}{2} h^2 \tag{16}$$

$$\mathbb{E}\left\| \begin{bmatrix} \boldsymbol{q}_3 - \bar{\boldsymbol{q}}_3 \\ \boldsymbol{p}_3 - \bar{\boldsymbol{p}}_3 \end{bmatrix} \right\|^2 \leq L^2 \bar{C}_0 \left( \alpha^2 + \frac{1}{2} + \frac{h^2}{4} \right) h^3. \tag{17}$$

Finally we need to bound $\mathbb{E}\|\boldsymbol{q}_1\|^2$ by $\mathbb{E}\|\boldsymbol{x}_0\|^2$, to this end, we have

$$
\begin{aligned}
\mathbb{E}\|\boldsymbol{q}_1\|^2 =& \mathbb{E}\left\|\boldsymbol{q}_0 + \frac{1-e^{-\gamma\frac{h}{2}}}{\gamma}\boldsymbol{p}_0 + \sqrt{2\gamma}\int_0^{\frac{h}{2}}\frac{1-e^{-\gamma(\frac{h}{2}-s)}}{\gamma}d\boldsymbol{B}(s)\right\|^2 \\
\leq& (1+\frac{h^2}{4})\mathbb{E}\|\boldsymbol{q}_0\|^2 + (1+\frac{h^2}{4})\mathbb{E}\|\boldsymbol{p}_0\|^2 + 2\gamma d\int_0^{\frac{h}{2}}\left(\frac{1-e^{-\gamma(\frac{h}{2}-s)}}{\gamma}\right)^2 ds \\
\leq& (1+\frac{h^2}{4})\mathbb{E}\|\boldsymbol{x}_0\|^2 + \frac{\gamma d}{12}h^3 \qquad\qquad\qquad (18) \\
=& (1+\frac{h^2}{4})\|\boldsymbol{x}_0\|^2 + \frac{\gamma d}{12}h^3 \qquad\qquad\qquad (19)
\end{aligned}
$$

Collecting all pieces together, including (16), (17), (19), the definition of $\bar{C}_0$ and $0 < h < \frac{1}{4L'}$, it is not difficult to obtain the following

$$
\left\|\mathbb{E}\begin{bmatrix}\boldsymbol{q}_3 - \bar{\boldsymbol{q}}_3 \\ \boldsymbol{p}_3 - \bar{\boldsymbol{p}}_3\end{bmatrix}\right\| \leq \bar{C}_1 h^2
$$

$$
\left(\mathbb{E}\left\|\begin{bmatrix}\boldsymbol{q}_3 - \bar{\boldsymbol{q}}_3 \\ \boldsymbol{p}_3 - \bar{\boldsymbol{p}}_3\end{bmatrix}\right\|^2\right)^{\frac{1}{2}} \leq \bar{C}_2 h^{\frac{3}{2}}
$$

with

$$
\bar{C}_1 = \alpha(\alpha + 1.125)(L+G)\left[0.5 + 0.71\|\boldsymbol{x}_0\| + (1.14\sqrt{\alpha} + 0.21\sqrt{\gamma}h)\sqrt{hd}\right]
$$

and

$$
\bar{C}_2 = L(\alpha + 0.73)\left(2.30\sqrt{h}\alpha L\|\boldsymbol{x}_0\| + (2.27\sqrt{\alpha} + 0.12\sqrt{\gamma}h)\sqrt{d}\right)
$$

$\square$

### D.7 Local error between Algorithm 1 and HFHR dynamics

**Lemma D.8.** *Assume $f$ is $L$-smooth, $\boldsymbol{0} \in \mathrm{argmin}_{\boldsymbol{x}\in\mathbb{R}^d} f(\boldsymbol{x})$, i.e. $\nabla f(\boldsymbol{0}) = \boldsymbol{0}$ and the operator $\nabla\Delta f$ grows at most linearly, i.e. $\left\|\nabla\Delta f(\boldsymbol{q})\right\| \leq G\sqrt{1+\|\boldsymbol{q}\|^2}$. If $0 < h \leq \frac{1}{4L}$, then compared with the HFHR dynamics, the implementable Strang's splitting method has local weak error of order $p_1 = 2$ and local mean-squared error of order $p_2 = 1.5$, i.e. there exist constants $C_1, C_2 > 0$ such that*

$$
\left\|\mathbb{E}\boldsymbol{x}(h) - \mathbb{E}\bar{\boldsymbol{x}}(h)\right\| \leq C_1 h^{p_1}
$$

$$
\left(\mathbb{E}\left[\left\|\boldsymbol{x}(h) - \bar{\boldsymbol{x}}(h)\right\|^2\right]\right)^{\frac{1}{2}} \leq C_2 h^{p_2}
$$

*where $\boldsymbol{x}(h) = \begin{bmatrix}\boldsymbol{q}(h) \\ \boldsymbol{p}(h)\end{bmatrix}$ is the solution of HFHR with initial value $\boldsymbol{x}_0 = \begin{bmatrix}\boldsymbol{q}_0 \\ \boldsymbol{p}_0\end{bmatrix}$ and $\bar{\boldsymbol{x}}(h) = \begin{bmatrix}\bar{\boldsymbol{q}}(h) \\ \bar{\boldsymbol{p}}(h)\end{bmatrix}$ is the solution of the implementable Strang's splitting with initial value $\boldsymbol{x}_0 = \begin{bmatrix}\boldsymbol{q}_0 \\ \boldsymbol{p}_0\end{bmatrix}$, $p_1 = 2$ and $p_2 = 1.5$. More concretely, we have*

$$
C_1 = (L+G)\max\{\alpha+1.25, \gamma+1\}\left[0.5\alpha + (1.74 + 0.71\alpha)\|\boldsymbol{x}_0\| + \left(1.26\sqrt{\alpha} + 1.14\alpha\sqrt{\alpha} + 2.32\sqrt{\gamma}\right)\sqrt{hd}\right]
$$

*and*

$$
C_2 = L\max\{\alpha+1.25, \gamma+1\}\left[(1.92 + 2.30\alpha L)\sqrt{h}\|\boldsymbol{x}_0\| + (2.60\sqrt{\alpha} + 3.34\sqrt{\gamma}h)\sqrt{d}\right]
$$

*Proof.* Denote by $\hat{\boldsymbol{x}}(h) = \begin{bmatrix} \hat{\boldsymbol{q}}(h) \\ \hat{\boldsymbol{p}}(h) \end{bmatrix}$ the solution of the exact Strang's splitting method with initial

value $\boldsymbol{x}_0 = \begin{bmatrix} \boldsymbol{q}_0 \\ \boldsymbol{p}_0 \end{bmatrix}$. By triangle inequality and Minkowski's inequality, we have

$$\left\| \mathbb{E}\boldsymbol{x}(h) - \mathbb{E}\bar{\boldsymbol{x}}(h) \right\| \leq \left\| \mathbb{E}\boldsymbol{x}(h) - \mathbb{E}\hat{\boldsymbol{x}}(h) \right\| + \left\| \mathbb{E}\hat{\boldsymbol{x}}(h) - \mathbb{E}\bar{\boldsymbol{x}}(h) \right\|,$$

$$\left( \mathbb{E}\left\| \boldsymbol{x}(h) - \bar{\boldsymbol{x}}(h) \right\|^2 \right)^{\frac{1}{2}} \leq \left( \mathbb{E}\left\| \boldsymbol{x}(h) - \hat{\boldsymbol{x}}(h) \right\|^2 \right)^{\frac{1}{2}} + \left( \mathbb{E}\left\| \hat{\boldsymbol{x}}(h) - \bar{\boldsymbol{x}}(h) \right\|^2 \right)^{\frac{1}{2}}.$$

By Lemma D.6 and D.7, we have

$$\left\| \mathbb{E}\boldsymbol{x}(h) - \mathbb{E}\hat{\boldsymbol{x}}(h) \right\| \leq \widehat{C}_1 h^2, \quad \left\| \mathbb{E}\hat{\boldsymbol{x}}(h) - \mathbb{E}\bar{\boldsymbol{x}}(h) \right\| \leq \bar{C}_1 h^2$$

$$\left( \mathbb{E}\left\| \boldsymbol{x}(h) - \hat{\boldsymbol{x}}(h) \right\|^2 \right)^{\frac{1}{2}} \leq \widehat{C}_2 h^{\frac{3}{2}}, \quad \left( \mathbb{E}\left\| \hat{\boldsymbol{x}}(h) - \bar{\boldsymbol{x}}(h) \right\|^2 \right)^{\frac{1}{2}} \leq \bar{C}_2 h^{\frac{3}{2}}$$

and hence

$$\left\| \mathbb{E}\boldsymbol{x}(h) - \mathbb{E}\bar{\boldsymbol{x}}(h) \right\| \leq (\widehat{C}_1 + \bar{C}_1) h^2$$

$$\left( \mathbb{E}\left\| \boldsymbol{x}(h) - \bar{\boldsymbol{x}}(h) \right\|^2 \right)^{\frac{1}{2}} \leq (\widehat{C}_2 + \bar{C}_2) h^{\frac{3}{2}}$$

with

$$\widehat{C}_1 + \bar{C}_1 \leq C_1$$
$$\triangleq (L + G) \max\{\alpha + 1.25, \gamma + 1\} \left[ 0.5\alpha + (1.74 + 0.71\alpha)\|\boldsymbol{x}_0\| + \left( 1.26\sqrt{\alpha} + 1.14\alpha\sqrt{\alpha} + 2.32\sqrt{\gamma} \right) \sqrt{hd} \right]$$
$$\widehat{C}_2 + \bar{C}_2 \leq C_2 \triangleq L \max\{\alpha + 1.25, \gamma + 1\} \left[ (1.92 + 2.30\alpha L)\sqrt{h}\|\boldsymbol{x}_0\| + (2.60\sqrt{\alpha} + 3.34\sqrt{\gamma}h)\sqrt{d} \right]$$

$\square$

# E $\alpha$ DOES CREATE ACCELERATION EVEN AFTER DISCRETIZATION: AN ANALYTICAL DEMONSTRATION

If $\alpha \to \infty$ while $\gamma$ remains fixed, then $dq = -\alpha\nabla f(q) + \sqrt{2\alpha}dW$ is the dominant part of the dynamics, and in this case the role of $\alpha$ could be intuitively understood as to simply rescale the time of gradient flow, which does not create any algorithmic advantage, as the timestep of discretization has to scale like $1/\alpha$ in this case. However, finite $\alpha$ no longer corresponds to solely a time-scaling, but closely couples with the dynamics and creates acceleration. This is true even after the continuous dynamics is discretized by an algorithm .

We will analytically illustrate this point by considering quadratic $f$. In this case, the diffusion process remains Gaussian, and it suffices to quantify the convergence of its mean and covariance. In fact, it can be shown that both have the same speed of convergence, and therefore for simplicity we will only consider the mean process. Two demonstrations (with different focuses) will be provided.

**Demonstration 1 (1D, $\gamma$ given; infinite acceleration).** Consider $f(x) = x^2/2$, $\gamma$ fixed. The mean process is

$$\begin{cases} \dot{q} &= p - \alpha q \\ \dot{p} &= -q - \gamma p \end{cases}$$

Consider, for simplicity, an Euler-Maruyama discretization of the HFHR dynamics, which coressponds to a Forward Euler discretization of the mean process (other numerical methods can be analyzed analogously):

$$\begin{bmatrix} q_{k+1} \\ p_{k+1} \end{bmatrix} = A \begin{bmatrix} q_k \\ p_k \end{bmatrix}, \qquad A = \begin{bmatrix} 1 - \alpha h & h \\ -h & 1 - \gamma h \end{bmatrix}.$$

We will show that, unless $\gamma = 2$, an appropriately chosen $\alpha$ will converge infinitely faster than the case with $\alpha = 0$, if both cases use the optimal $h$.

To do so, let us compute $A$'s eigenvalues, which are

$$\frac{1}{2}\left(2 - (\alpha + \gamma)h \pm h\sqrt{-4 + (\alpha - \gamma)^2}\right)$$

Consider the case where $|\alpha - \gamma| \leq 2$, then the eigenvalues are a pair of complex conjugates. Their modulus determines the speed of convergence, and it can be computed to be

$$\frac{1}{2}\sqrt{(2 - (\alpha + \gamma)h)^2 + h^2(4 - (\alpha - \gamma)^2)} = \sqrt{1 - (\alpha + \gamma)h + (1 + \alpha\gamma)h^2}$$

Minimizing the quadratic function gives the optimal $h$ that ensures the fastest speed of convergence, and the optimal $h$ is

$$h = \frac{\alpha + \gamma}{2(1 + \alpha\gamma)}$$

and the optimal spectral radius is

$$\sqrt{1 - \frac{(\alpha + \gamma)^2}{4(1 + \alpha\gamma)}}.$$

When one uses low-resolution ODE, in which $\alpha = 0$, the optimal rate is $1 - \gamma^2/4$ (note it is not surprising that the critically damped case, i.e., $\gamma = 2$, will give the fastest convergence).

If $\gamma \neq 2$, the additional introduction of $\alpha$ can accelerate the convergence by reducing the spectral radius. For instance, if $\alpha = \gamma + 2$, upon choosing the optimal $h = \frac{1}{1+\gamma}$, the optimal spectral radius is 0 (note in this case $A$ actually has Jordan canonical form of $\begin{bmatrix} 0 & 1 \\ 0 & 0 \end{bmatrix}$ and thus the discretization converges in 2 steps instead of 1, irrespective of the initial condition).

**Demonstration 2 (multi-dim, $\gamma$, $\alpha$ and $h$ all to be chosen; acceleration quantified in terms of condition number).** Consider quadratic $f$ with positive definite Hessian, whose eigenvalues are $1 = \lambda_1 < \cdots < \lambda_n = \epsilon^{-1}$ for some $0 < \epsilon \ll 1$. Assume without loss of generality that $f = q_1^2/2 + \epsilon^{-1}q_2^2/2$. Similar to Demonstration 1, the forward Euler discretization of the mean process is

$$\begin{bmatrix} q_{1,k+1} \\ p_{1,k+1} \\ q_{2,k+1} \\ p_{2,k+1} \end{bmatrix} = \begin{bmatrix} A_1 & 0 \\ 0 & A_2 \end{bmatrix} \begin{bmatrix} q_{1,k} \\ p_{1,k} \\ q_{2,k} \\ p_{2,k} \end{bmatrix}, \quad A_1 = \begin{bmatrix} 1 - \alpha h & h \\ -h & 1 - \gamma h \end{bmatrix}, \quad A_2 = \begin{bmatrix} 1 - \alpha\epsilon^{-1}h & h \\ -\epsilon^{-1}h & 1 - \gamma h \end{bmatrix}$$

(20)

We will (i) find $h$ and $\gamma$ that lead to fastest convergence of the ULD discretization, i.e. the above iteration with $\alpha = 0$, and then (ii) constructively show the existence of $h$, $\gamma$ and $\alpha$ that lead to faster convergence than the optimal one in (i) — note these may not even be the optimal choices for HFHR, but they already lead to significant acceleration. More specifically,

**(i)** In a ULD setup, $\alpha = 0$. It can be computed that the eigenvalues of $A_1$ and $A_2$ are respectively

$$\frac{1}{2}\left(2 - h\gamma \pm h\sqrt{-4 + \gamma^2}\right) \qquad \text{and} \qquad \frac{1}{2}\left(2 - h\gamma \pm h\sqrt{-4\epsilon^{-1} + \gamma^2}\right)$$

We now seek $\gamma > 0, h > 0$ to minimize the maximum of their norms for obtaining the optimal convergence rate. This is done in cases.

Case (i1) When $\gamma \leq 2$, both $A_1$ and $A_2$ eigenvalues are complex conjugate pairs. To minimize the maximum of their norms, let's first see if their norms could be made equal.

$A_1$ eigenvalue's norm squared $\times 4$ is

$$(2 - h\gamma)^2 - h^2(-4 + \gamma^2) = 4(h - \gamma/2)^2 + 4 - \gamma^2$$

(21)

$A_2$ eigenvalue's norm squared $\times 4$ is

$$(2 - h\gamma)^2 - h^2(-4\epsilon^{-1} + \gamma^2) = 4\epsilon^{-1}(h - \epsilon\gamma/2)^2 + 4 - \epsilon\gamma^2$$

(22)

It can be seen that for (21) is always strictly smaller than (22) for any $h > 0$. Therefore, the max of the two is minimized when $h = \epsilon\gamma/2$, and the corresponding max value is $4 - \epsilon\gamma^2$. $\gamma$ that minimizes this max value is $\gamma = 2$. Corresponding rate of convergence is

$$\sqrt{1 - \epsilon}.$$

Case (i2) When $\gamma \geq 2\epsilon^{-1/2}$, both $A_1$ and $A_2$ eigenvalues are real. Since $\epsilon \ll 1$, we can order them$\times 2$ as

$$2 - h\gamma - h\sqrt{-4 + \gamma^2} < 2 - h\gamma - h\sqrt{-4\epsilon^{-1} + \gamma^2} < 2 - h\gamma + h\sqrt{-4\epsilon^{-1} + \gamma^2} < 2 - h\gamma + h\sqrt{-4 + \gamma^2} < 2.$$

To minimize the max of their norms, consider cases in which the smallest of four is negative, in which case at optimum one should have

$$-(2 - h\gamma - h\sqrt{-4 + \gamma^2}) = 2 - h\gamma + h\sqrt{-4 + \gamma^2}.$$

This gives $h = 2/\gamma$ (which does verify the assumption that the smallest of four is negative). Corresponding max of their norms is thus $\sqrt{1 - 4/\gamma^2}$. $\gamma$ that minimizes this max value is $\gamma = 2\epsilon^{-1/2}$, which gives rate of convergence of

$$\sqrt{1 - \epsilon}.$$

Case (i3) When $2 \leq \gamma \leq 2\epsilon^{-1/2}$, $A_1$ eigenvalues are real and $A_2$ eigenvalues are complex conjugates. Again, the max of their norms is minimized if the norms can be made all equal.

Note $A_1$ eigenvalues cannot be of the same sign, because otherwise $2 - h\gamma - h\sqrt{-4 + \gamma^2} = 2 - h\gamma + h\sqrt{-4 + \gamma^2}$, which means either $h = 0$ or $\gamma = 2$, but if $\gamma = 2$ then $2 - h\gamma + h\sqrt{-4 + \gamma^2}$ being equal to 2*norm of $A_2$ eigenvalue, which is $\sqrt{4\epsilon^{-1}(h - \epsilon\gamma/2)^2 + 4 - \epsilon\gamma^2}$, leads to $h = 0$ again.

Therefore, the equality of norms of $A_1$, $A_2$ eigenvalues means

$$-(2 - h\gamma - h\sqrt{-4 + \gamma^2}) = 2 - h\gamma + h\sqrt{-4 + \gamma^2} = \sqrt{4\epsilon^{-1}(h - \epsilon\gamma/2)^2 + 4 - \epsilon\gamma^2}.$$

The first equality gives $h\gamma = 2$, which, together with the second equality, gives $h = \pm\sqrt{\frac{2\epsilon}{1+\epsilon}}$. Selecting the positive value of optimal $h$, we also obtain optimal $\gamma = \sqrt{2(1 + \epsilon)}\epsilon^{-1/2}$, which is $\leq 2\epsilon^{-1/2}$ and thus satisfying our assumption ($2 \leq \gamma \leq 2\epsilon^{-1/2}$). The corresponding rate of convergence is thus

$$\frac{1}{2}\left(2 - h\gamma + h\sqrt{-4 + \gamma^2}\right) = \sqrt{\frac{1 - \epsilon}{1 + \epsilon}}.$$

Summary of (i) Since $\sqrt{\frac{1-\epsilon}{1+\epsilon}} < \sqrt{1 - \epsilon}$, the ULD Euler-Maruyama discretization converges the fastest when

$$h = \sqrt{\frac{2\epsilon}{1 + \epsilon}}, \qquad \gamma = \sqrt{2(1 + \epsilon)}\epsilon^{-1/2},$$

and the corresponding discount factor of convergence (i.e. base of exponential convergence) is

$$\sqrt{\frac{1 - \epsilon}{1 + \epsilon}}, \qquad \text{where } \epsilon = 1/\kappa \text{ with } \kappa \text{ being Hessian's condition number.} \tag{23}$$

(ii) Now consider the HFHR setup. Let's first state a result: when

$$\gamma = \frac{\sqrt{4c^2\epsilon^4 + 8c^2\epsilon^3 + 4c^2\epsilon^2 + \epsilon^2 - 2\epsilon + 1} + \epsilon + 3}{2c\epsilon^2 + 2c\epsilon} > 0, \tag{24}$$

$$\alpha = \frac{-\sqrt{4c^2\epsilon^4 + 8c^2\epsilon^3 + 4c^2\epsilon^2 + \epsilon^2 - 2\epsilon + 1} + 3\epsilon + 1}{2c\epsilon^2 + 2c\epsilon} > 0, \qquad h = c\epsilon \tag{25}$$

for any $c > 0$ independent of $\epsilon$, the iteration (20) converges with discount factor

$$\frac{1}{\sqrt{2}(1 + \epsilon)}\sqrt{(1 - \epsilon)\left(1 - \epsilon + \sqrt{4c^2\epsilon^4 + 8c^2\epsilon^3 + (4c^2 + 1)\epsilon^2 - 2\epsilon + 1}\right)}. \tag{26}$$

While the exact expression is lengthy, it can proved that the HFHR non-optimal discount factor (26) is strictly smaller than the ULD optimal discount factor (23) for not only small but also large $\epsilon$'s.

For some quantitative intuition, discount factors respectively have the following Taylor expansions in $\epsilon$:

HFHR non-optimal: $\qquad\qquad 1 - 2\epsilon + \left(\dfrac{c^2}{2} + 2\right)\epsilon^2 + \mathcal{O}\left(\epsilon^3\right)$ $\qquad\qquad$ (27)

ULD optimal: $\qquad\qquad 1 - \epsilon + \dfrac{\epsilon^2}{2} + \mathcal{O}\left(\epsilon^3\right)$ $\qquad\qquad$ (28)

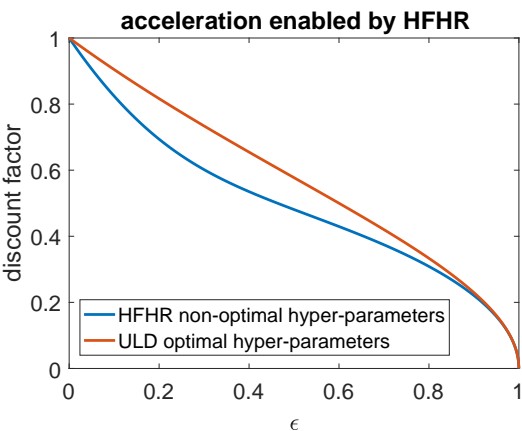

Figure 4: Acceleration of HFHR algorithm over ULD algorithm (despite of an additional constraint $\alpha$ may place on $h$) for multi-dimensional quadratic objectives. $1/\epsilon$ is the condition number.

The exact expressions of discount factors are also plotted in Fig.4 ($c = 1$ was arbitrarily chosen) and one can see acceleration for any (not necessarily small) $\epsilon$.

(**ii details**) How were values in (25) chosen? Following the idea detailed in (i), we consider a case where $A_1$ eigenvalues are both real, $A_2$ eigenvalues are complex conjugates, and all their norms are equal. Note there are 3 more cases, namely real/real, complex/real, and complex/complex, but we do not optimize over all cases for simplicity — the real/complex case is enough for outperforming the optimal ULD.

This case leads to at least the following equations

$$\begin{cases} \mathrm{tr}A_1 & = 0 \\ \det A_1 + \det A_2 & = 0 \end{cases} \qquad (29)$$

One can solve this system of equations to obtain $\alpha$ and $\gamma$ as functions of $h$. Following the idea of choosing $h$ small enough to resolve the stiffness of the ODE

$$\begin{cases} \dot{q}_2 & = p_2 - \alpha\epsilon^{-1}q_2 \\ \dot{p}_2 & = -\epsilon^{-1}q_2 - \gamma p_2 \end{cases},$$

pick $h = c\epsilon$. Then (29) gives

$$\gamma = \frac{\sqrt{4c^2\epsilon^4 + 8c^2\epsilon^3 + 4c^2\epsilon^2 + \epsilon^2 - 2\epsilon + 1} + \epsilon + 3}{2c\epsilon^2 + 2c\epsilon}$$

$$\alpha = \frac{-\sqrt{4c^2\epsilon^4 + 8c^2\epsilon^3 + 4c^2\epsilon^2 + \epsilon^2 - 2\epsilon + 1} + 3\epsilon + 1}{2c\epsilon^2 + 2c\epsilon}$$

or

$$\gamma = \frac{-\sqrt{4c^2\epsilon^4 + 8c^2\epsilon^3 + 4c^2\epsilon^2 + \epsilon^2 - 2\epsilon + 1} + \epsilon + 3}{2c\epsilon^2 + 2c\epsilon}$$

$$\alpha = \frac{\sqrt{4c^2\epsilon^4 + 8c^2\epsilon^3 + 4c^2\epsilon^2 + \epsilon^2 - 2\epsilon + 1} + 3\epsilon + 1}{2c\epsilon^2 + 2c\epsilon}$$

The former is our choice (25) because it can be checked that the latter leads to $\det A_1 > 0$ which violates the assumption of a pair of plus and minus real eigenvalues.

It is possible to find optimal $\alpha, \gamma, h$ for HFHR for the Gaussian cases. One has to minimize $\det A_2$ under the constraint $\det A_2 > 0$ in addition to (29). And then do similar calculations for the other 3 cases, and then finally the best among the 4 cases. Doing so however does not give enough insights to determine optimal hyperparameters for sampling general distributions.

## F RANDOMIZED MIDPOINT DISCRETIZATION OF HFHR

### F.1 THE ALGORITHM

HFHR is based on a continuous dynamics that adds HFHR corrections to the Underdamped Langevin Dynamics (ULD). It can be turned into a sampling algorithm via either a low-order time discretization (e.g., HFHR Algorithm 1) or a more accurate one. To complement the main text, this section demonstrates the latter, based on a powerful recent progress in discretizing ULD, known as Randomized Midpoint Algorithm (RMA) (Shen & Lee, 2019), and shows that the acceleration created by the HFHR correction terms persists.

More specifically, RMA is a high-order discretization scheme for ULD that achieved a better $\mathcal{O}(d^{\frac{1}{3}})$ dimension dependence of mixing time than first-order discretization of ULD, e.g., 1st-order KLMC (Dalalyan & Riou-Durand, 2020). Although RMA is originally designed specifically for ULD only, it is a general idea and already adapted to overdamped Langevin (He et al., 2020). Here we show it can be easily adapted to HFHR as well, as illustrated by the following Algorithm 2. Red highlights algorithmic changes we made to account for the HFHR corrections of ULD.

---

**Algorithm 2** Randomized Midpoint Algorithm from Shen & Lee (2019), adapted for HFHR

---

1: **Input**: potential function $f$ and its gradient $\nabla f$, damping coefficients $\alpha$ and $\gamma$, step size $h$, initial condition $(\boldsymbol{q}_0, \boldsymbol{p}_0)$

2: **procedure** RMA-HFHR$(f, \nabla f, \alpha, \gamma, h, \boldsymbol{q}_0, \boldsymbol{p}_0)$

3:      $k = 0$ and initialize $\begin{bmatrix} \boldsymbol{q}_0 \\ \boldsymbol{p}_0 \end{bmatrix}$

4:      **while** not converged **do**

5:          Generate an independent uniform random variable $\theta_k \sim U(0,1)$

6:          Generate Gaussian random vectors $\left( \boldsymbol{W}_{k+1}^1, \boldsymbol{W}_{k+1}^2, \boldsymbol{W}_{k+1}^3 \right) \in \mathbb{R}^{3d}$ as in (Shen & Lee, 2019, Appendix A)

7:          Generate Gaussian random vectors $\boldsymbol{B}_{k+1}^1, \boldsymbol{B}_{k+1}^2 \in \mathbb{R}^d$ as described by (31)

8:          $\boldsymbol{q}_{k+\frac{1}{2}} = \boldsymbol{q}_k + \frac{1}{\gamma}(1 - e^{-\gamma\theta_k h})\boldsymbol{p}_k - \frac{1}{\gamma}\left( \theta_k h - \frac{1}{\gamma}(1 - e^{-\gamma\theta_k h}) \right)\nabla f(\boldsymbol{q}_k) + \boldsymbol{W}_{k+1}^1$ $-\alpha\theta_k h \nabla f(\boldsymbol{q}_k) + \sqrt{2\alpha}\boldsymbol{B}_{k+1}^1$

9:          $\boldsymbol{q}_{k+1} = \boldsymbol{q}_k + \frac{1}{\gamma}(1 - e^{-\gamma h})\boldsymbol{p}_k - \frac{1}{\gamma}h(1 - e^{-\gamma(h-\theta_k h)})\nabla f(\boldsymbol{q}_{k+\frac{1}{2}}) + \boldsymbol{W}_{k+1}^2 -$ $\alpha h \nabla f(\boldsymbol{q}_{k+\frac{1}{2}}) + \sqrt{2\alpha}(\boldsymbol{B}_{k+1}^1 + \boldsymbol{B}_{k+1}^2)$

10:         $\boldsymbol{p}_{k+1} = \boldsymbol{p}_k e^{-\gamma h} - h e^{-\gamma(h-\theta_k h)}\nabla f(\boldsymbol{q}_{k+\frac{1}{2}}) + 2\boldsymbol{W}_{k+1}^3$

11:         $k \leftarrow k + 1$

12:      **end while**

13: **end procedure**

---

The red parts basically correspond to two Euler-Maruyama time-steppings of an auxiliary dynamics that contains only the HFHR correction terms

$$d\boldsymbol{q} = -\alpha \nabla f(\boldsymbol{q})dt + \sqrt{2\alpha}d\boldsymbol{B}_t, \tag{30}$$

first over a $\theta_k h$ timestep, and then over an $h$ timestep. These two steps originate from an operator splitting treatment of the full HFHR dynamics (eq.6), which is split into ULD and (30). Therefore, it is natural to see that

$$\boldsymbol{B}_{k+1}^1 = \int_{hk}^{h(k+\theta_k)} d\boldsymbol{B}_t, \qquad \boldsymbol{B}_{k+1}^2 = \int_{h(k+\theta_k)}^{h(k+1)} d\boldsymbol{B}_t,$$

and therefore $B_{k+1}^1$ and $B_{k+1}^2$ are, when conditioned on $\theta_k$, centered Gaussian vectors independent from each other and the $\boldsymbol{W}$'s, each being $d$-dimensional with i.i.d. entries, and they can be generated via

$$\boldsymbol{B}_{k+1}^1 = \sqrt{\theta_k h} \boldsymbol{\xi}_{k+1}^1, \qquad \boldsymbol{B}_{k+1}^2 = \sqrt{h - \theta_k h} \boldsymbol{\xi}_{k+1}^2, \qquad (31)$$

where $\boldsymbol{\xi}_{k+1}^1$ and $\boldsymbol{\xi}_{k+1}^2$ are i.i.d. standard d-dimensional Gaussian vectors.

**Remark F.1.** *In the original RMA (Shen & Lee, 2019, Algorithm 1), the uniform random variable for the midpoint's proportional location was denoted by $\alpha$. However, since we have already used this letter for the HFHR correction coefficient, we use instead $\theta$ to denote this uniform random variable.*

**Remark F.2.** *From the red text, it is easy to see that if $\alpha = 0$, Algorithm 2 degenerates to RMA for ULD. Nevertheless, Algorithm 2 is again just one RMA discretization of HFHR but not the only one.*

### F.2 NUMERICAL RESULTS: HFHR AGAIN ACCELERATES

To numerically compare the RMA discretization of HFHR dynamics and ULD dynamics (note we don't compare 1st-order HFHR Algorithm 1 with RMA-ULD as we'd like to compare apple with apple), we conduct an experiment very similar to that in Sec.6.1, with the same nonlinear potential function. We run both RMA for ULD and RMA for HFHR with dimension $d = 10$, initial value $(100 \times \mathbf{1}_d, \mathbf{0}_d)$, $h = 1$ (chosen to be near the stability limit of RMA-ULD), a family of $\gamma \in \{0.1, 0.2, 0.5, 1, 2, 5, 10, 20, 50, 100\}$ and $\alpha \in \{0, 0.001, 0.002, 0.005, 0.01, 0.02, 0.05, 0.1, 0.2, 0.5, 0.55, 0.6, 0.65, 0.7, 0.75, 0.8, 0.85, 0.9, 0.95, 1, 2, 5, 10, 20, 50, 100\}$. For each algorithm and each set of parameter values, we run 1,000 independent realizations to compute statistics and estimate the mean time of reaching $\varepsilon = 0.1$ neighborhood of the target distribution. Then, for each $\alpha$ (including $\alpha = 0$, which is the original RMA), we optimize over $\gamma$ choices to get the best results. To further reduce variance, we also repeat the experiment with 100 different random seeds.

Too large $\alpha$ values with which Algorithm 2 fails to reach $\epsilon$-neighborhood are not plotted and the final results are shown in Figure 5. It clearly suggests that with appropriated chosen $\alpha$ ($\alpha = 0.5$ in our case), RMA discretized HFHR dynamics requires fewer iterations than RMA discretized ULD, which suggests a better iteration complexity.

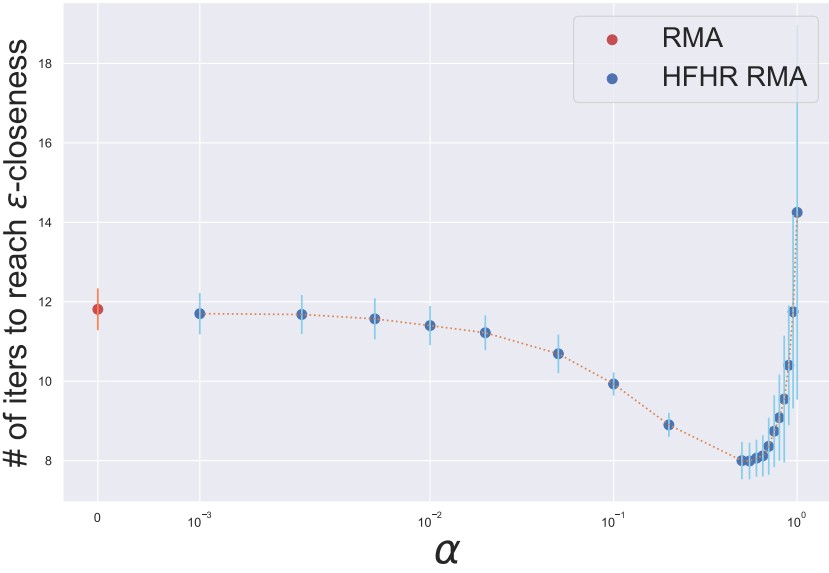

Figure 5: Improvement of RMA for HFHR (Algorithm 2) over the original RMA (for ULD) in iteration complexity. (vertical bar = 1 std.)

