# OpenReview forum: "Hessian-Free High-Resolution Nesterov Acceleration for Sampling"
_ICLR.cc/2022/Conference — ICLR 2022 Submitted_

### Official Review · Reviewer_xkFf · 2021-10-17

**Correctness:** 4
**Technical Novelty And Significance:** 3
**Empirical Novelty And Significance:** 2
**Recommendation:** 3
**Confidence:** 4

**Main Review:**

Strengths:
The HFHR process seems novel and has interesting connections with NAG. The paper is mostly clear and easy to follow.

Weakness:
1. HFHR has noise injected into position directly

This property is actually harmful for discretization because directly coupling Brownian motion into $q_t$ makes the trajectory of $q_t$ extremely non smooth and make it hard to estimate $\nabla f(q_t)$.

The ULD avoid directly inject noise into position to achieve acceleration compared to the overdamped LD.
HFHR, unlike from ULD, has noise term $dW_t$ in position. However it still achieve same asymptotic convergence as certain discretized ULD at the price of an extra condition on higher order derivative.

2. Rely on additional third-order growth condition

Although this condition is weaker than Hessian Lipschitz and has been used in prior art to accelerate the convergence of LD [Li et al., 2021], it still makes the convergence guarantee weaker than many discretized ULD, which have same or better asymptotic convergence speed and don't require third-order growth condition.

3. The condition number $\kappa$ dependence and third-order growth constant $G$ dependence in iteration complexity is unclear

The main paper doesn't specify the asymptotic dependence of $\kappa$ and $G$ but hide it in the constant $C$. Including that information in the main paper would be appreciated.

According the appendix page 20, where C is defined, I guess that the $\kappa$ dependence is $O(\kappa^{3/2})$ and $G$ dependence is $O(G)$. Could the author help to verify the above statements?

4. Iteration complexity is worse than certain discretization of ULD

Although the authors cited [Shen & Lee, 2019], they didn't discuss methods in this paper at all.

The randomized midpoint method is a special discretization of ULD, which achieves $\widetilde{O}(d^{1/3}/\varepsilon^{2/3})$ iteration complexity and doesn't require third-order growth condition.

5. The method is not well motivated

According to the paper, HFHR is motivated by one question "how to appropriately inject noise to NAG algorithm in discrete time". We should note that, although there are underlying connections between optimization and sampling, injecting noise into a good optimization method doesn't necessarily yield a good sampling algorithm.
Unfortunately, this paper seems like this is the case. More specifically, discrete HFHR with $\alpha \neq 0$ have worse iteration complexity compared to certain discretization of ULD.

6. Points in Figure 2 is biased estimation

If I understand correctly, the y-axis value in Figure 2 is smallest number $k$ such that $\lVert \mathbb{E} {\mu_k} q-\mathbb{E} \mu q \rVert \leq \varepsilon$.
In order to estimate $\mathbb{E} {\mu_k} q$, $\frac{1}{n} \sum_i^{n} q_k^{(i)}$ is calculated for $n=1000$ samples (according to line 3 in generate_config.py) which are generated and run in parallel for $k$ iterations.

$\frac{1}{n} \sum_i^{n} q_k^{(i)}$ is indeed unbiased estimation of $\mathbb{E} {\mu_k} q$. However
$\min k$ $s.t. \lVert \frac{1}{n} \sum_i^{n} q_k^{(i)}-\mathbb{E} \mu q \rVert \leq \varepsilon $
 is a biased estimation of
$\min k$ $s.t. \lVert \mathbb{E} {\mu_k} q-\mathbb{E} \mu q \rVert \leq \varepsilon$.

Typo:

dt is missing in eq 6.
Bracket error on 3rd-to-last line on page 8.

**Summary Of The Paper:**

This paper introduces a stochastic process called HFHR. The SDEs of HFHR is derived by rewriting Nesterov’s Accelerated Gradient (NAG) into phase-space representation, formulating it as ODEs, and then injecting noise into both position and momentum variables. HFHR can be used for sampling because it has stationary distribution as the target distribution. A discretization of HFHR is given by operator splitting and Euler integration. A mixing time bound of $\widetilde{O}(\sqrt{d}/\varepsilon)$ is obtained for sampling log-strong-concave-and-smooth target distribution with extra third-order growth condition.

**Summary Of The Review:**

The method is novel in the sense that it is derived based on the idea "inject noise into Nesterov’s Accelerated Gradient".
However the derived theory is not significant as it rely on extra assumptions and has worse iteration complexity.

---

> ### Author Response · Authors · 2021-11-23
> **Response to Reviewer xkFf (part 4 of 4)**
>
> > Rely on additional third-order growth condition
>
> We completely agree and thank the reviewer for bringing this to discussion. Regarding this criticism, may we respond in three steps? (1) While many discretized ULD do not need this condition, adding it seems to us not significantly improving their convergence either. Therefore, HFHR could still be useful for sufficiently regular potentials. (2) We're not sure whether our requirement of this additional regularity condition is a necessity or a limitation of our proof techniques. Without it we can still prove convergence, but bounds become worse. Nevertheless, this condition didn't have an effect in our empirical experiments, so it is not necessarily a drawback of our method, but admittedly it is a weakness of our theory component. (3) However, that was just for our analysis of a 1st-order scheme; we're not sure if other schemes, such as RMA-HFHR that we added to Appendix F, still requires it. Unfortunately we were unable to provide a full nonasymptotic analysis to each discretization we introduced due to space and time limitations.
>
> > The condition number $\kappa$ dependence and third-order growth constant $G$ dependence in iteration complexity is unclear. Are they $\mathcal{O}(\kappa^{3/2})$ and $\mathcal{O}(G)$ judging from page 20?
>
> Thanks for an important question. The third-order growth constant dependence is indeed $\mathcal{O}(G)$, but the condition number dependence should be better than $\mathcal{O}(\kappa^{3/2})$. We did not focus on explicit $\kappa$ dependence in our analysis, but it might be at most $\mathcal{O}(\kappa)$ judging from Remark 5.6.
>
> > Figure 2 uses a biased estimation of mixing time, because although $\frac{1}{n}\sum_i^n q_k^{(i)}$ is an unbiased estimation of $\mathbb{E}_{\mu_k}q$, $\min k$ s.t. $| \\frac{1}{n} \\sum_i^n q_k^{(i)} - \\mathbb{E} _{\\mu_k} q | \\leq \\varepsilon$ is a biased estimation of $\min k$ s.t. $| \\mathbb{E} _{\mu_k}q - \\mathbb{E} _{\mu}q | \\leq \\varepsilon$.
>
>
> We are thankful for the expert comment. This is very interesting and we agree. We haven't found any fix of this bias in the literature and it seems the estimate we used is very commonly adopted. Any suggestion would be greatly appreciated. Meanwhile, are we correct in guessing that if $1/\sqrt{n} \ll \varepsilon$ then this bias is not too much an issue? Our experiments do satisfy $1/\sqrt{n} \ll \varepsilon$.
>
> > Summary: The method is novel in the sense that it is derived based on the idea "inject noise into Nesterov’s Accelerated Gradient". However the derived theory is not significant as it rely on extra assumptions and has worse iteration complexity.
>
> We sincerely hope we explained why our iteration complexity is not worse but better, once apple-to-apple comparisons are made. Regarding the additional regularity condition, it is indeed required by our analysis for our 1st-order discretization; however, we're not sure if it is required by the method per se, or if all discretizations require it. If our explainations and efforts could be found reasonable, we would deeply appreciate a reconsideration.

---

> ### Author Response · Authors · 2021-11-23
> **Response to Reviewer xkFf (part 3 of 4)**
>
> > HFHR has noise injected into position directly. This is harmful. ULD doesn't.
>
> We completely understand why the reviewer had this impression. We, like many people, also had the same feeling, and we know this is one of the motivations behind, for instance, higher-order momentum for sampling [Mou et al. JMLR 2021]. Nevertheless, as far as we know, the hand-waiving intuition of "having noise in position is harmful" is not yet precisely formulated as a conjecture, let alone proved. We see at least four aspects in which the "harm" of having noise on position could be discussed:
>
> 1. Does having noise on position make popular existing discretizations inferior? We're no longer sure. For example, we used to believe *Euler-Maruyama discretization* of ULD has better dimension dependency than *that* of overdamped Langevin, partly due to that the latter has noise on the state variable itself, but recent results including [Chewi et al. COLT 2021] and [Li et al. arXiv 2109.03839] also, roughly speaking, gave $\tilde{O}(\sqrt{d})$ for the latter, which is the same as the best known dimension-dependency of the former.
>
>     Note in this comment we were focusing on comparing the dynamics per se, meaning that we won't be comparing a low order discretization of dynamics A with a high order discretization of dynamics B, because that would obscure the source of advantage. Instead, discretizations of the same order are discussed. However, this does lead to the next aspect:
> 2. Does additive noise on position make it harder to design good discretization? Maybe, because intuitively it gives bad regularity due to Ito stochastic integral, but we still don't know anything definite yet. Is randomized midpoint (the original version in [Shen & Lee 2019]) a great sampler because it combines the great idea of appropriate randomization and the fact that ULD has no noise on position? We're not entirely certain. For example, [He, Balasubramanian, and Erdogdu. NeurIPS 2020] showed one can also construct randomized midpoint method for overdamped Langevin (which has noise on position) and obtain improved rates too. We tend to think the success of randomized midpoint lies in the fact that it is a high order method (see e.g., [Li et al. NeurIPS 2019]), and its high-orderness is achieved, primarily via the great idea of randomization instead of merely having no noise on position.
> 3. Is the lower regularity created by having noise on position always bad? We don't think so, even if it did make discretization harder. Here is an example reason: There are at least two equally popular ways of using MCMC to generate i.i.d. samples. One is to simulate an ensemble of many independent realizations till fixed, long enough time, and collect independent values at the final time. The other is to simulate just one realization till very long time and then collect values along the tail of the time horizon as the desired samples. The latter approach doesn't really give independent samples as there will be **temporal correlation**. It's natural to think that smoother trajector corresponds to, in general, stronger correlation in time. In this sense, having noise on position may actually help the quality of samples. In short, we feel that, without quantitative comparisons, qualitative intuition may not be completely reliable.
> 4. We do feel that a lower regularity created by having noise on position, in general, poses extra challenges to analysis, especially tight bounds are desired. Existing results often make additional assumptions on the regularity of the potential to achieve improved bounds. We haven't seen much progress on whether these are neccessities or artifacts due to the analysis techniques. (Kindly see our response to the next point for more discussion on this.)
>
> In all cases, independent of all our speculations above or insights others may have, we do have provided both theoretical proofs and empirical evidence, that adding the HFHR corrections to the position improves the convergence rates of ULD. This is for both the continuous-time dynamics, and discretizations of the same type/order (the original submission is based on 1st-order discretization, and in the revision we added empirical comparisons based on randomized-midpoint discretization of HFHR and ULD dynamics).

---

> ### Author Response · Authors · 2021-11-23
> **Response to Reviewer xkFf (part 2 of 4)**
>
> > The method is not well motivated. although there are underlying connections between optimization and sampling, injecting noise into a good optimization method doesn't necessarily yield a good sampling algorithm. Unfortunately, this paper seems like this is the case. More specifically, discrete HFHR with $\alpha\neq 0$ have worse iteration complexity compared to certain discretization of ULD.
>
> Thanks for the opportunity of clarification. We apologize for any possible misunderstanding. We didn't state that injecting noise into a good optimization method generally yields a good sampling algorithm. Instead, we asked ourselves whether this could be the case for the specific algorithm of NAG-SC, and then discovered both theoretical and empirical evidence of a positive answer.
>
> Regarding "worse iteration complexity", hopefully our reply to the previous point clarified that comparing low-order discretization of HFHR dynamics with high-order discretization of ULD can be misleading, but HFHR does have better iteration complexity when its 1st-order discretization is compared with 1st-order discretizations of ULD, and when its RMA discretization is compared with RMA discretization of ULD too.

---

> ### Author Response · Authors · 2021-11-23
> **Response to Reviewer xkFf (part 1 of 4)**
>
> We thank the reviewer for taking the time to review our paper and providing many valuable feedback. Below is our itemized reply to the concerns (randomized midpoint is moved to the first because some other replies will also relate to this important point):
>
> > Iteration complexity is worse than randomized midpoint discretization of ULD
>
> Thank you very much for raising this concern. This is very important, and perhaps also a major factor of the reviewer's assessment, but fortunately we feel it can be addressed. In short, once we compare apple with apple and orange with orange, this concern hopefully resolves. Here are detailed explanations:
>
> (1) Our main objective was to suggest that the extra HFHR-correction terms, namely $- \alpha \nabla f(q_t) dt + \sqrt{2\alpha} dW_t$, are advantageous. For this, we quantified the acceleration they enabled both in continuous time and in a discretization. The purpose of our time-discretized analysis is mainly to show that the acceleration enabled by HFHR is a genuine one which persists after discretrization, unlike illusive accelerations created by trivialities such as time-rescaling, which will disappear after discretization. For that purpose, we had to choose some discretization, and a most fair choice would be to use similar discretizations for both HFHR and ULD, so that there is minimized distraction from the difference between discretization accuracies. Therefore, we chose a 1st-order scheme, because we thought 1st-order discretizations such as LMC/unadjusted-Langevin-algorithm and KLMC are still popular in the community, so we hoped to illustrate the acceleration that HFHR can bring, in a 1st-order setup. In Sec.1 of the original submission, we wrote "it was known that high-order discretizations can improve statistical accuracy and even the speed of convergence, although such improvements often come with more computations per iteration. The discretization considered here is just a simple first-order scheme that uses one (full-)gradient evaluation per step". Note RMA uses 2 gradient evaluations per step, so there are still situations in which simple 1st-order schemes are useful.
>
> (2) Meanwhile, we completely agree that the iteration complexity of a 1st-order discretization may not be as good as that of a higher-order discretization. When compared to Randomized Midpoint Algorithm (RMA) of ULD (i.e. HFHR dynamics with $\alpha=0$), which is a high-order discretization, the iteration complexity of our 1st-order HFHR discretization is indeed worse. However, we hope to emphasize that this comparison can be rather misleading, because it is between **a low-order discretization of dynamics A** and **a high-order discretization of dynamics B**, which per se is not a very good indicator of if dynamics A is better or worse than B. That is why we instead compared a low-order discretization of dynamics A with two **same**-order discretizations of dynamics B (one being KLMC and the other being the $\alpha=0$ version of the HFHR algorithm in the original submission). We now realized that these were not expressed clearly enough in the original submission, and have revised accordingly.
>
> (3) Nevertheless, thanks to the reviewer, we also investigated the effect of HFHR corrections on RMA, as it is a great high-order discretization. More precisely, we added another comparison between HFHR dynamics and ULD dynamics based on RMA discretization, in Appendix F (i.e., **a high-order discretization of dynamics A** v.s. **a same-order discretization of dynamics B**). There we provided both a description of how we generalized RMA to discretize HFHR dynamics, and empirical results that clearly show that HFHR corrections again lead to extra acceleration over ULD, when both are discretized by the high-order scheme of RMA. This makes the picture more complete, as now we have both low- and high-order discretizations, in addition to continuous time results, all of which confirm the nontrivial acceleration of HFHR.

---

> > ### Comment · Reviewer_xkFf · 2021-11-23
> > **How do we define "fair"?**
> >
> > > fair choice would be to use similar discretizations for both HFHR and ULD
> >
> > This is not fair.
> > 1. This is not how we compare ULD and LD.
> > When we say "ULD is better than LD" we compare KLMC and ULA.
> > Although ULD also has a Euler discretization, we almost never use it because we know there is a better choice.
> > 2. We should consider it as advantage if a process has higher order integrator, instead of hiding it.
> >
> > Based on the above points, I think the following comparison is fairer:
> > 1. Compare optimal discretization for two process. If we don't know optimal discretization, then
> > 2. Compare best known discretization for two process.

---

> > > ### Author Response · Authors · 2021-11-24
> > > **Response to Reviewer xkFf round 2**
> > >
> > > Thank you very much for the discussion.
> > >
> > > In the revised version, we added a Randomized Midpoint discretization of HFHR dynamics, and compared it with the Randomized Midpoint discretization of ULD dynamics. In this comparison, extra acceleration created by the HFHR correction was again observed, just like in the 1st-order case. These were also summarized in the round 1 response above.
> > >
> > > We don't think the optimal discretization of ULD is known, let alone that of HFHR, which is just proposed. Assuming we correctly understood the comment, that Randomized Midpoint should be the discretization to use in a comparison, we do have this discretization and comparison.
> > >
> > > If HFHR only admitted low-order discretization(s) but ULD admitted high-order one(s), we would completely agree that comparing low-order discretization of HFHR with low-order discretization of ULD would be unfair, and in that case we should do low-order HFHR v.s. high-order ULD. However, that is *not* the case, and since both HFHR and ULD admit 1st-order and Randomized Midpoint discretizations, can we consider it to be fair to compare 1st-order HFHR with a good 1st-order discretization of ULD, and Randomized Midpoint HFHR with Randomized Midpoint ULD? Kindly note that by 1st-order we didn't mean Euler-Maruyama (the next bullet describes more details).
> > >
> > > Here are additional itemized responses:
> > > > This is not how we compare ULD and LD. When we say "ULD is better than LD" we compare KLMC and ULA. Although ULD also has a Euler discretization, we almost never use it because we know there is a better choice.
> > >
> > > We completely agree. Now we explained that both HFHR and ULD have low- and high-order discretizations, and we also hope to clarify that, by "comparing similar discretizations", we meant comparing discretizations of the same order and based on the same number of gradient evaluations per step (sorry these details were probably buried in our round 1 response; we will discuss them again in the next bullet). Under these constraints, exactly like the reviewer suggested, we should choose the best available method. That's why we never used Euler discretization of ULD, and had we been comparing ULD and LD, KLMC v.s. ULA would fit the constraints. (Two discretizations of ULD involved in our discussion are KLMC and 1st-order HFHR algorithm with $\alpha=0$, but no Euler)
> > >
> > > > We should consider it as advantage if a process has higher order integrator, instead of hiding it.
> > >
> > > We agree and we did not hide it. In fact, our revision constructed randomized midpoint discretization of HFHR, and compared it with randomized midpoint discretization of ULD. Our round 1 response also described this fact.
> > >
> > > Meanwhile, as we also mentioned, there are still situations in which low-order methods are useful, because they use less gradient evaluation(s) per step. More precisely, they suit downstream applications in which gradient evaluation is computational expensive and thus the dominant part of the computational cost (e.g., Bayesian neural network, Bayesian inverse problems with nontrivial forward models such as PDE solves) and yet sampling accuracy requirement is low.
> > >
> > > To see this concretely, consider a standard nonasymptotic sampling error bound
> > > $$ e_k \leq \exp(-\beta k h) e_0 + C h^p, \qquad \text{as long as } h\leq h_0.$$ Suppose, for example, that we'd like reach $e_k \leq \epsilon=0.2$ accuracy using as few steps as possible, and we have both a $p=1$ low-order method and a $p=2$ higher-order method. For concreteness and simplicity, assume contraction rate $\beta=1$, initial error $e_0=1$, and the same order constant $C=1$ and stability limit $h_0\leq 0.1$ for both methods. Simple constrained optimization over $h$ values shows:
> > > 1. When $p=1$, optimum $e_{23} \approx 0.2003$ and optimum $e_{24} \approx 0.1907$, i.e., 24 steps are needed to reach $\epsilon$ accuracy.
> > > 2. When $p=2$, optimum $e_{16} \approx 0.2119$ and optimum $e_{17} \approx 0.1927$, i.e., 17 steps are needed to reach $\epsilon$ accuracy.
> > >
> > > If the $p=1,2$ methods respectively use 1 and 2 gradient evaluations per step (recall 1st-order HFHR and KLMC use 1 each while randomized-midpoint HFHR and randomized-midpoint ULD uses 2 each), then the low-order method needs 24 gradients, but the high-order method needs 34. Thus, the low-order method is computationally more efficient in this case, because it is an $\epsilon=0.2$ low accuracy problem.
> > >
> > > If we have a high accuracy problem instead, e.g., $\epsilon=0.01$, then under the same setup the low- and high-order methods respectively need 764 and 140 gradient evaluations (details omitted). Now the high-order method clearly wins.
> > >
> > > That is why we added randomized midpoint (high-order) discretization of HFHR to the revision (thanks to the reviewer) but also kept our 1st-order discretization. We feel both have their own playgrounds.

---

> > > > ### Comment · Reviewer_xkFf · 2021-11-24
> > > > **Questions on Algorithm 2**
> > > >
> > > > > If HFHR only admitted low-order discretization(s) but ULD admitted high-order one(s), we would completely agree that comparing low-order discretization of HFHR with low-order discretization of ULD would be unfair, and in that case we should do low-order HFHR v.s. high-order ULD.
> > > >
> > > > I am glad that we have reached the same understanding of fair comparison.
> > > >
> > > > Now the following issues are in focus:
> > > >
> > > > **Is Algorithm 2 really a high-order discretization?**
> > > >
> > > > More specifically, does it has improved $\varepsilon$ dependence such as $O(\varepsilon^{-2/3})$ (corrected) in RMA?
> > > >
> > > > The authors seems to claim that Algorithm 2 is a high-order discretization because they said:
> > > >
> > > > > "now we have both low- and high-order discretizations"
> > > > > "both HFHR and ULD have low- and high-order discretizations"
> > > >
> > > > However, they don't provide any proof of this claim. Moreover, the authors don't empirically measure the step size dependence as in Figure 1a for Algorithm2.
> > > >
> > > > I hope authors could provide either theoretical guarantee or empirical evidence that Algorithm2 could improve the $\widetilde{O}(\varepsilon^{-1})$ rate of low-order discretization.
> > > >
> > > > Actually, I would be quite surprised if Algorithm2 could improve the $\varepsilon$ dependence.
> > > > [1] gives $O(\varepsilon^{-1})$ lower bound for overdamped Langevin.
> > > > Given that HFHR has noise on position which is similar to overdamped Langevin, I suspect same lower bound should apply.
> > > >
> > > > > Does additive noise on position make it harder to design good discretization? Maybe, because intuitively it gives bad regularity due to Ito stochastic integral, but we still don't know anything definite yet.
> > > >
> > > > We do know that lower bound for any possible discretization of overdamped Langevin is worse than existing upper bound and lower bound for ULD, and this fact directly depends on that overdamped Langevin has noise on position.
> > > >
> > > > > there are still situations in which low-order methods are useful
> > > >
> > > > I fully agree.
> > > > However, comparing these constants could be very hard, as they could change in different problem. Therefore I prefer to seek for best asymptotic behaviour, then optimize constants in special situation.
> > > >
> > > > [1] Clark, John MC, and R. J. Cameron. "The maximum rate of convergence of discrete approximations for stochastic differential equations." In Stochastic differential systems filtering and control, pp. 162-171. Springer, Berlin, Heidelberg, 1980.

---

> > > > > ### Author Response · Authors · 2021-11-26
> > > > > **Response to Reviewer xkFf round 3a (Re: Algorithm 2)**
> > > > >
> > > > > > I would be quite surprised if Algorithm2 could improve the $\varepsilon$ dependence. [Clark and Cameron 1980] gives $\mathcal{O}(\varepsilon)$ lower bound for overdamped Langevin. Given that HFHR has noise on position which is similar to overdamped Langevin, I suspect same lower bound should apply.
> > > > >
> > > > > The $\mathcal{O}(\varepsilon)$ lower bound is well known as an order barrier of numerical SDE integration. The specific barrier the reviewer referred to exists, but only for discretizations that are based on deterministic oracles that allow each step to only query gradient and increments of the driving Wiener process, and then use a deterministic function of their current and past values. See, in addition to the important reference of Clark and Cameron (1980), also Rüemelin (1982). However, methods involving multiple stochastic integrals (e.g.,  Kloeden & Platen (1992); Milstein & Tretyakov (2013); Rößler (2010)) and/or **randomization** (e.g., Shen & Lee (2019)) do not have this barrier. In fact, as described in our round 1 response, He, Balasubramanian, and Erdogdu (NeurIPS 2020) already constructed randomized **over**damped Langevin that broke this barrier.  Even prior to that, Li, Wu, Mackey, and Erdogdu (NeurIPS 2019) already obtained $\tilde{\mathcal{O}}(\epsilon^{-2/3})$ discretization of **over**damped Langevin using high-order stochastic Runge-Kutta discretization (which uses double stochastic integral). Our Algorithm 2 is randomized too, and thus the $\mathcal{O}(\varepsilon)$ lower bound is irrelevant.
> > > > >
> > > > > > Is Algorithm 2 really a high-order discretization?
> > > > >
> > > > > Here is the sketch of a proof: we use the mean-square analysis framework (Li et al. 2019, Li et al. 2021) to transfer local numerical *integration* error to global nonasymptotic *sampling* error. More precisely, we only need to prove that our RMA-discretization of HFHR dynamics has local strong integration error of order $p_2 > 1.5$ and local weak integration error of order $p_1 \geq p_2+0.5$. Then, since the continuous HFHR dynamics is already proved to be geometric ergodic (Thm.5.1 of our submission) and contractive (Lemma D.5 of our submission), mean-square analysis (Li et al. 2019, Li et al. 2021) gives a bound on the W2 distance between the target distribution and the distributio at any step $k$, hence a quantification of the sampling error. To show that we actually have $p_2 = 2$ and $p_1 \geq 2.5$, which will give an $\epsilon^{-2/3}$ dependence (**not** $-3/2$ as you wrote by the way; RMA-ULD's dependence is $\epsilon^{-2/3}$), simply use computations analogous to those in (Shen and Lee. NeurIPS 2019) and (He, Balasubramanian, and Erdogdu. NeurIPS 2020).
> > > > >
> > > > > Admittedly this is not a full proof, which would be similiar to our existing one and take another $\sim 20$ pages. We've already provided substantial additional materials that the reviewer requested in our revised version, and unfortunately, based on ICLR timeline, at this point we don't have the access or enough time to update the manuscript and provide additional substantial materials (i.e., detailed full proof and/or additional large ensemble experiments which would take many days) requested again by the reviewer. We appreciate the scholarship, as it is what we deeply value too. Meanwhile, we wonder if this is really why a rating of 3 --- let us back up and pretend we didn't have the above evidence of the high-orderness of RMA-HFHR. Still, we've already provided empirical results that RMA-HFHR exhibits significantly improvement over RMA-ULD. According to the original review, comparing within the same order is not that necessary and we just need to improve upon RMA-ULD, and we did improve and submitted a revised version when we can. In addition, the original proof is already $\sim$ 20 pages and it is for a discretization that we already explained to be useful, but we still added another discretization because we believed that the reviewer is helping us improve the paper. However, adding another $\sim$ 20 page proof for the 2nd discretization may just diffuse the attention --- we just want to show that having additional HFHR terms in the dynamics helps; in addition, we seldom saw top ML conference papers that provide 40-page proofs of both two methods that are almost parallel.
> > > > >
> > > > > > However, comparing these constants could be very hard, as they could change in different problem.
> > > > >
> > > > > We agree. But not many experts we know would assert that low-order methods are completely useless.

---

> > > > > > ### Comment · Reviewer_xkFf · 2021-11-27
> > > > > > **Is Algorithm 2 really a high-order discretization?**
> > > > > >
> > > > > > Unfortunately, I cannot take any claim for granted without enough evidence, especially when the result is exceptionally good.
> > > > > >
> > > > > > I am glad authors cite these analysis frameworks to reduce the problem into local integration error.
> > > > > >
> > > > > > > To show that we actually have $p_2=2$ and $p_1 \geq 2.5$, which will give an $\varepsilon^{-2/3}$ ... simply use computations analogous to those in (Shen and Lee. NeurIPS 2019) and (He, Balasubramanian, and Erdogdu. NeurIPS 2020).
> > > > > >
> > > > > > I think this claim is rather critical and not that simple. I cannot directly apply logic in two cited paper, as continuous process and assumptions are different.
> > > > > >
> > > > > > Could authors provide a proof for just local integration error? Although it doesn't constitute a full proof of global convergence, I believe establishing $p_2=2$ and $p_1 \geq 2.5$ is enough to convince me that RMA-HFHR is a high order integrator.

---

> > > > > > > ### Author Response · Authors · 2021-11-29
> > > > > > > **Response to Reviewer xkFf round 4a (Re: questions on Algorithm 2)**
> > > > > > >
> > > > > > > Thanks for the continued discussion.
> > > > > > >
> > > > > > > > Could authors provide a proof for just local integration error?
> > > > > > >
> > > > > > > If you don't mind taking another look at our proofs of the local error of 1st-order HFHR (this is what we initially proposed, which is useful for low-accuracy applications), it can been seen that the proofs are lengthy calculations (they are given in Appendix D.5-D.7, page 24-33). Therefore, like stated in response round 3a, we do not have the space or time to provide a full proof, even just for the local error of RMA-HFHR (which is not in our initial submission, but constructed and tested as a response to the review).
> > > > > > >
> > > > > > > Do you not trust the empirical evidence in Fig.5? We can step back and think about it in this way: given an algorithm that has a parameter $\alpha$, which degenerates to RMA-ULD when $\alpha=0$ and discretizes continuous dynamics that converges faster with some nonzero $\alpha$, why is it not believable that one can get better results than RMA-ULD if one tunes $\alpha$?
> > > > > > >
> > > > > > > > I cannot take any claim for granted without enough evidence, especially when the result is exceptionally good.
> > > > > > >
> > > > > > > If we understood this sentence correctly, shouldn't having good results be a good thing? Even though everything in our original submission came with proofs, we agree that what we added in response to the reviewer's comment is without sharp rigorous guarantees; however, we saw a lot of good results in machine learning that do not come with full proofs. We also posted rebuttal codes for everyone to check if our added empirical evidence is true. Even though the reviewer still questions our Fig.1a (which merely verifies results that are already rigorously proved), these rebuttal codes are for Fig.5.
> > > > > > >
> > > > > > > ---
> > > > > > > In any case, we feel we might not be able to convince the reviewer no matter what we do given this rebuttal period, but we do want to thank the reviewer, and all other reviewers and the chairs for your time.

---

> > > > > > > > ### Comment · Reviewer_xkFf · 2021-11-29
> > > > > > > > **No evidence on improved step size dependence**
> > > > > > > >
> > > > > > > > I will discuss what authors has provided and why I think evidence is not enough.
> > > > > > > >
> > > > > > > > When I look into theory, authors refuse to provide proof of improved local integration error (due to space and time issue).
> > > > > > > >
> > > > > > > > When I look into experiment, authors only provide Figure 5, which doesn't directly measure step size dependence. If Algorithm 2 has worse asymptotic dependence, comparison in Figure 5 might change when $\varepsilon$ is small enough.
> > > > > > > > I don't know why authors only empirically measure step size dependence for Algorithm 1 in Figure 1a but don't apply the same experiment to Algorithm 2. Moreover, during my attempt to extend experiment in Figure 1a to Algorithm 2 by myself, I find Figure 1a cannot be reproduced.
> > > > > > > >
> > > > > > > > Given all evidence at hand, I find it hard to convince myself that "Algorithm 2 is a high order integrator".

---

> > > > ### Comment · Reviewer_xkFf · 2021-11-24
> > > > **Questions on Experiment**
> > > >
> > > >  * `code/verifyDependenceOnStepSize/verify_dependence_on_h_nonlinear_potential.py`could not reproduce Figure 1a.
> > > >
> > > > Line 184 of this file is `n_samples = int(1e6)`, which indicates that $10^6$ samples are used to estimate the mean.
> > > >
> > > > I find this value quite critical for reducing the bias and variance of the estimation.
> > > > I have tried both origin value `1e6` and larger value `1e8` (with parallelization).
> > > > Interestingly, it seems even `1e8` is not sufficient to give accurate estimation.
> > > >
> > > > In order to give more accurate estimation, I also use time averaging.
> > > > The best estimation of "error of mean" $\lVert\mathbb{E}\_{\mu\_\infty} X-\mathbb{E}\_{\mu} X\lVert$ is around $10^{-6}$. However, if I increase sample size, it seems this value could decrease further.
> > > >
> > > > In summary, I find it quite challenging to estimate $\lVert\mathbb{E}\_{\mu\_\infty} X-\mathbb{E}\_{\mu} X\lVert$, and I am curious how authors manage to generate Figure 1a. More detailed questions:
> > > >
> > > > 1. How many samples, or equivalently, what `n_samples ` do authors use for Figure 1a?
> > > > 2. Are all datapoints in Figure 1a estimated with same number of samples?
> > > > 3. What is the estimated bias and variance of "error of mean", i.e. y-axis value in Figure 1a?
> > > > 4. When I uses `n_samples = int(1e8)` plus time averaging, even for `h=1`, I could obtain estimated error of mean as small as $10^{-6}$. However, Figure 1a show error of mean larger than $10^{-4}$ for `h=1`.
> > > > Does that mean that estimation in Figure 1a is too imprecise?
> > > >
> > > > * In the recent revision, authors append a new algorithm and new experiment. Could authors provide related new code?

---

> > > > > ### Author Response · Authors · 2021-11-26
> > > > > **Response to Reviewer xkFf round 3b (Re: questions on experiment)**
> > > > >
> > > > > We're a little surprised that the reviewer questions Fig.1. It is only for providing additional (numerical) validation of theoretical results that are already rigorous proved. Nevertheless, we stand by reproducibility and checked again carefully, and here are itemized responses.
> > > > >
> > > > > > code/verifyDependenceOnStepSize/verify_dependence_on_h_nonlinear_potential.py could not reproduce Figure 1a.
> > > > >
> > > > >
> > > > > This script provides the funtionality to generate Figure 1a. However, as we used large ensemble and hence computational expensive simulations to generate Figure 1a, we parallelized computation jobs on a high performance cluster. Each run of the script gave one portion of data and then we aggregated and averaged the results to produce Fig 1a.
> > > > >
> > > > > The last line in `code/verifyDependenceOnStepSize/verify_dependence_on_h_nonlinear_potential.py`
> > > > > ```
> > > > > np.save(f'results/sample_mean|h={h}|d={d}|n_samples={n_samples}|part={seed}.npy', sample_mean)
> > > > > ```
> > > > > together with line 176
> > > > > ```
> > > > > parser.add_argument("--seed", type=int)
> > > > > ```
> > > > > both prove that we parallelized the jobs with multiple seeds.
> > > > >
> > > > > > How many samples, or equivalently, what n_samples do authors use for Figure 1a?
> > > > >
> > > > > We used `1e8` independent samples for each data point in Figure 1a. This number, however, is not equivalent to the variable `n_samples` in the script. As explained above, we used `n_samples=int(1e6)` for each single job, but run the script with 100 different seeds. So altogether `1e8` independent samples are used.
> > > > >
> > > > > > Are all datapoints in Figure 1a estimated with same number of samples?
> > > > >
> > > > > Yes, each data point in Fig. 1a uses $10^8$ independent samples.
> > > > >
> > > > > > What is the estimated bias and variance of "error of mean", i.e. y-axis value in Figure 1a?
> > > > >
> > > > > We're not sure if we understood correctly. An approximation based on empirical average is an unbiased estimator of the mean, and its variance is $\mathcal{O}(1/n)$.
> > > > >
> > > > >
> > > > > > When I uses n_samples = int(1e-8) plus time averaging, even for h=1, I could obtain estimated error of mean as small as $10^{-6}$. However, Figure 1a show error of mean larger than  $10^{-4}$ for h=1. Does that mean that estimation in Figure 1a is too imprecise?
> > > > >
> > > > > It is good to know and learn that with additional time averaging, the reviewer can obtain estimated error of mean as small as $10^{-6}$. However, the purpose of Fig.1a is to validate our theory, which doesn't have time averaging, so we did not do time averaging to reduce the error. Therefore, we're unsure if there is anything wrong with an error of $\approx 3\times 10^{-4}$ at $h=1$, or Fig.1a in general.
> > > > >
> > > > > > In the recent revision, authors append a new algorithm and new experiment. Could authors provide related new code?
> > > > >
> > > > > We will try to figure out how to share additional codes in a way compliant with the ICLR policy. If we find it, we'll reply to this thread as soon as we can.

---

> > > > > > ### Author Response · Authors · 2021-11-27
> > > > > > **Response to Reviewer xkFf round 3c (Re: questions on experiment) - part I**
> > > > > >
> > > > > > Unfortunately, we are not able to update supplementary materials for now. Instead, we will post our implementation of both RMA and HFHR-RMA here. Please insert the two code snippets below at line 181in `verifyIterationComplexity/utils.py`  and modify `verify_iteration_complexity.py` accordingly to run the new HFHR-RMA algorithm.

---

> > > > > > ### Author Response · Authors · 2021-11-27
> > > > > > **Response to Reviewer xkFf round 3c (Re: questions on experiment) - part II**
> > > > > >
> > > > > > ```
> > > > > >     def RMA(self, h, gamma):
> > > > > >         dist, N, D = self.dist, self.N, self.D
> > > > > >
> > > > > >         def generate_H_G(theta, h):
> > > > > >             H1, H2, G1, G2 = np.empty((N, D)), np.empty((N, D)), np.empty((N, D)), np.empty((N, D))
> > > > > >
> > > > > >             # slow for loop implementation
> > > > > >             for i in range(N):
> > > > > >                 theta_ = theta[i][0]
> > > > > >
> > > > > >                 # H1, G1
> > > > > >                 cov1 = np.array([
> > > > > >                     [0.25*(np.exp(4*theta_*h)-1), 0.5*(np.exp(2*theta_*h)-1)],
> > > > > >                     [0.5*(np.exp(2*theta_*h)-1), theta_*h]
> > > > > >                 ])
> > > > > >                 M1 = sqrtm(cov1)
> > > > > >                 noise1, noise2 = np.random.randn(D), np.random.randn(D)
> > > > > >                 G1[i, :] = M1[0][0] * noise1 + M1[0][1] * noise2
> > > > > >                 H1[i, :] = M1[1][0] * noise1 + M1[1][1] * noise2
> > > > > >
> > > > > >                 # H2, G2
> > > > > >                 cov2 = np.array([
> > > > > >                     [0.25*(np.exp(4*h)-np.exp(4*theta_*h)), 0.5*(np.exp(2*h)-np.exp(2*theta_*h))],
> > > > > >                     [0.5*(np.exp(2*h)-np.exp(2*theta_*h)), h-theta_*h]
> > > > > >                 ])
> > > > > >                 M2 = sqrtm(cov2)
> > > > > >                 noise1, noise2 = np.random.randn(D), np.random.randn(D)
> > > > > >                 G2[i, :] = M2[0][0] * noise1 + M2[0][1] * noise2
> > > > > >                 H2[i, :] = M2[1][0] * noise1 + M2[1][1] * noise2
> > > > > >
> > > > > >             return H1, H2, G1, G2
> > > > > >
> > > > > >
> > > > > >         def generate_W(theta, h, H1, H2, G1, G2):
> > > > > >             W1 = H1 - np.exp(-2*theta*h) * G1
> > > > > >             W2 = (H1 + H2) - np.exp(-2*h) * (G1 + G2)
> > > > > >             W3 = np.exp(-2*h) * (G1 + G2)
> > > > > >             return W1, W2, W3
> > > > > >
> > > > > >         self.initialize()
> > > > > >         name = r'RMA($\gamma$={})'.format(gamma)
> > > > > >         self.record_stats(name)
> > > > > >         for it in range(1, self.n_iter + 1):
> > > > > >             theta = np.random.rand(N, 1) # make theta broacastable
> > > > > >             H1, H2, G1, G2 = generate_H_G(theta, h)
> > > > > >             W1, W2, W3 = generate_W(theta, h, H1, H2, G1, G2)
> > > > > >
> > > > > >             q_mid = self.q + 0.5*(1-np.exp(-2*theta*h))*self.p - 0.5*(theta*h-0.5*(1-np.exp(-2*theta*h)))*dist.dV(self.q) + W1
> > > > > >
> > > > > >             self.q = self.q + 0.5*(1-np.exp(-2*h))*self.p - 0.5*h*(1-np.exp(-2*(h-theta*h)))*dist.dV(q_mid) + W2
> > > > > >             self.p = self.p*np.exp(-2*h) - h*np.exp(-2*(h-theta*h))*dist.dV(q_mid) + 2*W3
> > > > > >
> > > > > >             self.record_stats(name)
> > > > > >             if not self.benchmark and self.stop:
> > > > > >                 self.iteration_complexity = it
> > > > > >                 break
> > > > > >
> > > > > >         self.results[name]['samples'] = self.q.copy()
> > > > > >         self.methods_to_compare.append(name)
> > > > > > ```

---

> > > > > > ### Author Response · Authors · 2021-11-27
> > > > > > **Response to Reviewer xkFf round 3c (Re: questions on experiment) - part III**
> > > > > >
> > > > > > ```
> > > > > >     def HFHR_RMA(self, h, gamma, alpha):
> > > > > >         dist, N, D = self.dist, self.N, self.D
> > > > > >
> > > > > >         def generate_H_G(theta, h):
> > > > > >             H1, H2, G1, G2 = np.empty((N, D)), np.empty((N, D)), np.empty((N, D)), np.empty((N, D))
> > > > > >
> > > > > >             # slow for loop implementation
> > > > > >             for i in range(N):
> > > > > >                 theta_ = theta[i][0]
> > > > > >
> > > > > >                 # H1, G1
> > > > > >                 cov1 = np.array([
> > > > > >                     [0.25*(np.exp(4*theta_*h)-1), 0.5*(np.exp(2*theta_*h)-1)],
> > > > > >                     [0.5*(np.exp(2*theta_*h)-1), theta_*h]
> > > > > >                 ])
> > > > > >                 M1 = sqrtm(cov1)
> > > > > >                 noise1, noise2 = np.random.randn(D), np.random.randn(D)
> > > > > >                 G1[i, :] = M1[0][0] * noise1 + M1[0][1] * noise2
> > > > > >                 H1[i, :] = M1[1][0] * noise1 + M1[1][1] * noise2
> > > > > >
> > > > > >                 # H2, G2
> > > > > >                 cov2 = np.array([
> > > > > >                     [0.25*(np.exp(4*h)-np.exp(4*theta_*h)), 0.5*(np.exp(2*h)-np.exp(2*theta_*h))],
> > > > > >                     [0.5*(np.exp(2*h)-np.exp(2*theta_*h)), h-theta_*h]
> > > > > >                 ])
> > > > > >                 M2 = sqrtm(cov2)
> > > > > >                 noise1, noise2 = np.random.randn(D), np.random.randn(D)
> > > > > >                 G2[i, :] = M2[0][0] * noise1 + M2[0][1] * noise2
> > > > > >                 H2[i, :] = M2[1][0] * noise1 + M2[1][1] * noise2
> > > > > >
> > > > > >             return H1, H2, G1, G2
> > > > > >
> > > > > >
> > > > > >         def generate_W(theta, h, H1, H2, G1, G2):
> > > > > >             W1 = H1 - np.exp(-2*theta*h) * G1
> > > > > >             W2 = (H1 + H2) - np.exp(-2*h) * (G1 + G2)
> > > > > >             W3 = np.exp(-2*h) * (G1 + G2)
> > > > > >             return W1, W2, W3
> > > > > >
> > > > > >         self.initialize()
> > > > > >         name = r'HFHR_RMA($\gamma$={}, $\alpha$={})'.format(gamma, alpha)
> > > > > >         self.record_stats(name)
> > > > > >         for it in range(1, self.n_iter + 1):
> > > > > >             theta = np.random.rand(N, 1) # make theta broacastable
> > > > > >             H1, H2, G1, G2 = generate_H_G(theta, h)
> > > > > >             W1, W2, W3 = generate_W(theta, h, H1, H2, G1, G2)
> > > > > >
> > > > > >             B1 = np.sqrt(theta * h) * np.random.randn(N, D)
> > > > > >             B2 = np.sqrt(h-theta * h) * np.random.randn(N, D)
> > > > > >
> > > > > >             # import pdb; pdb.set_trace()
> > > > > >             q_mid = self.q + 0.5*(1-np.exp(-2*theta*h))*self.p - 0.5*(theta*h-0.5*(1-np.exp(-2*theta*h)))*dist.dV(self.q) + W1 - theta*h*alpha*dist.dV(self.q) + np.sqrt(2*alpha)*B1
> > > > > >
> > > > > >             self.q = self.q + 0.5*(1-np.exp(-2*h))*self.p - 0.5*h*(1-np.exp(-2*(h-theta*h)))*dist.dV(q_mid) + W2 - h*alpha*dist.dV(q_mid) + np.sqrt(2*alpha)*(B1 + B2)
> > > > > >             self.p = self.p*np.exp(-2*h) - h*np.exp(-2*(h-theta*h))*dist.dV(q_mid) + 2*W3
> > > > > >
> > > > > >             self.record_stats(name)
> > > > > >             if not self.benchmark and self.stop:
> > > > > >                 self.iteration_complexity = it
> > > > > >                 break
> > > > > >
> > > > > >         self.results[name]['samples'] = self.q.copy()
> > > > > >         self.methods_to_compare.append(name)
> > > > > > ```

---

> > > > > > ### Comment · Reviewer_xkFf · 2021-11-27
> > > > > > **Biased Estimation Issue**
> > > > > >
> > > > > > > We're not sure if we understood correctly. An approximation based on empirical average is an unbiased estimator of the mean
> > > > > >
> > > > > > $ \lVert \frac{1}{n} \sum_i^{n} q_k^{(i)}-\mathbb{E} \mu q \rVert$ is used to estimate $\lVert \mathbb{E} {\mu_k} q-\mathbb{E} \mu q \rVert$. I think it is pretty clear that the former is a biased estimate of the latter (for any finite n).
> > > > > >
> > > > > > My experiment shows that for $n={10}^8$ the bias is still pretty large, and is much larger than estimated value $\lVert \mathbb{E} {\mu_k} q-\mathbb{E} \mu q \rVert$, such that no meaningful estimation could be achieved.
> > > > > >
> > > > > > I am curious how meaningful result in Figure 1a could be obtained.

---

> > > > > > > ### Author Response · Authors · 2021-11-29
> > > > > > > **Response to Reviewer xkFf round 4b (Re: questions on experiment)**
> > > > > > >
> > > > > > > Thanks for the continued discussion. We already provided codes and checked to make sure our results are reproducible, but in order to avoid going into a deadlock, here is an alternative response: It seems a key concern of the reviewer's is the one described in Biased Estimation Issue, quoted below:
> > > > > > >
> > > > > > > > $| \frac{1}{n} \sum_i^n q_k^{(i)} - \mathbb{E} _{\mu} q |$ is used to estimate $| \mathbb{E} _{\mu_k}q - \mathbb{E} _{\mu}q |$. $\cdots$ for $n=10^8$ the bias is still pretty large, and is much larger than the estimated value $| \mathbb{E} _{\mu_k}q - \mathbb{E} _{\mu}q |$, such that no meaningful estimation could be achieved.
> > > > > > >
> > > > > > > To understand the magnitude of this bias and it's signal to noise ratio, we considered a toy problem: Assume $\mu \sim \mathcal{N}(\mu,\sigma^2)$ and $\mu_k \sim \mathcal{N}(\mu+\delta \mu,(\sigma+\delta\sigma)^2)$, where $\mu, \delta\mu, \sigma, \delta\sigma$ are all scalars. Let $q_k^{(i)}$ be i.i.d. samples of $\mu_k$. Obviously we have $$ | \mathbb{E} _{\mu_k}q - \mathbb{E} _{\mu}q | = |\delta\mu|$$ $$\frac{1}{n} \sum_i^n q_k^{(i)} \sim \mathcal{N}(\mu+\delta\mu, (\sigma+\delta\sigma)^2/n I)$$
> > > > > > > Let $X$ be $X=\frac{1}{n} \sum_i^n q_k^{(i)} - \mathbb{E} _{\mu} q$, then $X\sim \mathcal{N}(\delta\mu, (\sigma+\delta\sigma)^2/n)$. The task of understanding the bias translates into understanding the difference between $\mathbb{E}|X|$ and $|\delta\mu|$. Therefore, we now obtain lower and upper bound of $\mathbb{E}|X|$.
> > > > > > > To lower bound it, assume without loss of generality that $\delta\mu\geq 0$ because we can otherwise replace $X$ by $-X$ without affecting the result. Then $$ \mathbb{E}|X| \geq \mathbb{E}X = \delta\mu = |\delta\mu|$$ To upper bound it, note $$ \mathbb{E}|X| \leq \mathbb{E}|X-\delta\mu|+|\delta\mu|$$ Using the standard Euclidean norm bound of centered sub-Gaussian random variable, we have $$\mathbb{E}|X-\delta\mu| \leq 4 \sqrt{\text{Var}(X-\delta\mu)}=4(\sigma+\delta\sigma)/\sqrt{n}$$
> > > > > > > Therefore, $$ \left| \mathbb{E} | \frac{1}{n} \sum_i^n q_k^{(i)} - \mathbb{E} _{\mu} q | - | \mathbb{E} _{\mu_k}q - \mathbb{E} _{\mu}q | \right| \leq 4(\sigma+\delta\sigma)/\sqrt{n} $$ The right hand side will get multiplied by $\sqrt{d}$ if we have a $d$-dimensional case.
> > > > > > >
> > > > > > > Let's now connect to the problem considered in our paper. Since results there were computed nearing convergence, $\sigma + \delta \sigma$ and $\sigma$ are on the same order of magnitude, which is determined by the target distribution and at the order of 1. $d=2$ for Fig.1a and $d=10$ for Fig.2.  Therefore, the numerical value of this bias is very small when $n=10^8$.
> > > > > > >
> > > > > > > However, like the reviewer stated, whether the bias is negligible is determined by how it compares to the "signal" which is $| \mathbb{E} _{\mu_k}q - \mathbb{E} _{\mu}q |=|\delta\mu|$. The requirement of having a meaningful result is thus
> > > > > > > $$ 4(\sigma+\delta\sigma)/\sqrt{n} \ll |\delta\mu|$$ Note $|\delta\mu| \approx \varepsilon$, where $\varepsilon$ is the accuracy cutoff we previously used, and thus the requirement is exactly what we wrote in our round 1 response part 4, namely $1/\sqrt{n} \ll \varepsilon$. Note constants are already considered and the $\ll$ sign means literal comparison of numerical values.
> > > > > > >
> > > > > > > This analogy is of course not rigorous though, because the this toy example is only for Gaussians. Admittedly we don't have enough time or space to provide all the details, but we believe concentration bounds could help us stay in the ballpark in terms of order of magnitude.
> > > > > > >
> > > > > > > Therefore we feel $n=10^8$ should be large enough for $\varepsilon=0.1$.

---

> > > > > > > > ### Comment · Reviewer_xkFf · 2021-11-29
> > > > > > > > **Enough for ε=0.1, but not enough for ε<0.00001**
> > > > > > > >
> > > > > > > > > Therefore we feel $n=10^8$ should be large enough for $\varepsilon=0.1$
> > > > > > > >
> > > > > > > > My question is "I am curious how meaningful result in Figure 1a could be obtained.", and Figure 1a contains result as small as $\varepsilon < 10^{-5}$.
> > > > > > > >
> > > > > > > > Do the authors feel $n=10^8$ is large enough for $\varepsilon < 10^{-5}$?

---

> > > > > > > > ### Comment · Reviewer_xkFf · 2021-11-29
> > > > > > > > **Reproducibility Issue**
> > > > > > > >
> > > > > > > > > We already provided codes and checked to make sure our results are reproducible
> > > > > > > >
> > > > > > > > However, I cannot reproduce it as mentioned in previous comment.
> > > > > > > >
> > > > > > > > I have specified the seed I used. Could authors reproduce my result in previous comment?
> > > > > > > >
> > > > > > > > Why the result I obtained is not even similar to Figure 1a?
> > > > > > > >
> > > > > > > > Are there any difference between authors' method and my method? I am using code provided by authors. I guess the only possible changes are seeds (which authors don't specify) and step size list (where I only evaluate for a single step size, and I will conduct experiment with original step size list later).

---

> > > > > > ### Comment · Reviewer_xkFf · 2021-11-27
> > > > > > **Cannot reproduce the experiment**
> > > > > >
> > > > > > > It is good to know and learn that with additional time averaging, the reviewer can obtain estimated error of mean as small as $10^6$.
> > > > > >
> > > > > > I really don't understand why authors think it is "good to know".
> > > > > > My result indicate that a correct implementation with $n={10}^8$ samples will have large bias that totally blur the result.
> > > > > >
> > > > > > What's more, I am not convinced that Figure 1a comes from a correct implementation, according to the argument below.
> > > > > >
> > > > > > I reuse authors' code and run it 100 times with different seeds, as authors directed.
> > > > > >
> > > > > > ```
> > > > > > # run.sh
> > > > > > for j in {1..100}; do printf '%d\n' j; done | parallel python verify_dependence_on_h_nonlinear_potential.py --seed {}
> > > > > > ```
> > > > > >
> > > > > > For $h=1$, I changed line 188 into `hs = [1]` to only calculate for $h=1$.
> > > > > > I got 100 output files such as `results/sample_mean|h=1|d=2|n_samples=1000000|part=1.npy`.
> > > > > > I retrieve last column for each npy file, since that is for last iteration.
> > > > > > I calculate mean of these 100 points.
> > > > > > I calculate the 2-norm difference to ground truth $[-0.5, -0.5]$.
> > > > > > I got result `4.691039182210643e-05`, which is almost one order of manitude smaller than point $h=1$ in Figure 1a.
> > > > > >
> > > > > > For $h=1/2^7$, with similar procedure, I got result `4.728168032762106e-05` which is almost one order of manitude larger than point $h=1/2^7<1e{-2}$ in Figure 1a, and almost same as $h=1$.
> > > > > >
> > > > > > These observations convince me that $n={10}^8$ samples is far from enough to generate meaningful estimation. Even if we plot the meaningless output from the program, the figure should be a flat line instead of Figure 1a which fits $O(h)$ well.

---

> ### Comment · Reviewer_xkFf · 2021-11-27
> **Summary of claims and my concerns:**
>
> Authors' (implicit and explicit) claim:
>
> 1. Continuous HFHR converge faster than continuous ULD.
>
> I think this is an interesting contribution.
>
> 2. (Comparing the constant,) Algorithm 1 with $\alpha\neq 0$ is better than Algorithm 1 with $\alpha=0$.
>
> I think this is an interesting contribution.
>
> 3. Under third-order growth condition, Algorithm 1 achieves $\widetilde{O}(\sqrt{d}/\varepsilon)$ iteration complexity.
>
> No concern in theoretical proof.
> But I have concerns on claim 11 which verifies this claim.
>
> 4. (Comparing the constant,) Algorithm 1 convergence provably faster than KLMC
>
> My concern:
> I didn't see the proof.
>
> 5. (Comparing the constant,) Algorithm 1 convergence empirically faster than KLMC
>
> Given that both methods have same asymptotic behaviour, this claim is supported by Figure 2.
>
> 6. Discretized HFHR is better than discretized ULD.
>
> This claim is ill formed as two terms cannot be both distributed.
> However, authors and I agree to interpret it as:
> The best known discretized HFHR is better than best known discretized ULD.
> This claim turns into claim 7.
>
> 7. Algorithm 2 converge faster than RMA for ULD.
>
> This claim can be separated into two claims 8 and 10.
>
> 8. Algorithm 2 has same asymptotic behavior as RMA for ULD.
>
> I agree that this claim could be justified either:
>
> Theoretically: If claim 9 is true.
>
> Empirically: If step size dependence of Algorithm 2 could be measured in same setting as Figure 1a. This experiment need claim 11 as a prerequisite.
>
> My concerns:
>
> Theoretically, I have concern on claim 9.
>
> Empirically, authors don't provide this experiment I request. Moreover, I have concerns on claim 11.
>
> 9. Algorithm 2 has $p_2=1$ and $p_1\geq 2.5$ in terms of local integration error.
>
> My concern:
> Authors' analogical proof is too vague.
>
> 10. Algorithm 2 has better constant than RMA for ULD.
>
> If claim 8 is justified, then claim 10 could be supported by Figure 5.
>
> 11. Figure 1a is correct.
>
> This claim requires claim 12 to be true.
>
> My concern:
> I cannot reproduce the result under authors' instruction.
>
> 12. `1e8` independent samples are enough to provide accurate estimation of "error of mean" under setting of Figure 1a.
>
> My concerns:
> "error of mean" only has biased estimation.
> The bias is typically very large.
> `n_samples=1e8` (or equivalently, `n_samples=1e6` with 100 different seeds) is not enough to provide meaningful result.

---

### Official Review · Reviewer_97Mf · 2021-10-21

**Correctness:** 2
**Technical Novelty And Significance:** 1
**Empirical Novelty And Significance:** 1
**Recommendation:** 1
**Confidence:** 1

**Main Review:**

I do not think the problem addressed by this paper does not make sense.  The gradient-based algorithm added by the noise in the convex case does not play the essentially different role with that without the noise.

**Summary Of The Paper:**

 This paper discuss the accelerated-gradient-based MCMC method and propose some rigorous proof.

**Summary Of The Review:**

Please the authors refer to

Bin Shi, Weijie Su and Michael I. Jordan,  On Learning Rates and Schrödinger Operators, https://arxiv.org/abs/2004.06977

Bin Shi, On the Hyperparameters in Stochastic Gradient Descent with Momentum, https://arxiv.org/abs/2108.03947

---

> ### Author Response · Authors · 2021-11-23
> **Response to Reviewer 97Mf**
>
> We appreciate the effort of the reviewer. However, we feel very unfortunate that a rating of "1: strong reject" was given based on a confidence of "1: You are unable to assess this paper and have alerted the ACs to seek an opinion from different reviewers."
>
> We also do not understand the Main Review: "I do not think the problem addressed by this paper does not make sense. The gradient-based algorithm added by the noise in the convex case does not play the essentially different role with that without the noise." If there is anything we could do, we'd be thankful if you could let us know.

---

### Official Review · Reviewer_r34U · 2021-10-26

**Correctness:** 4
**Technical Novelty And Significance:** 3
**Empirical Novelty And Significance:** 3
**Recommendation:** 6
**Confidence:** 3

**Details Of Ethics Concerns:**

No ethic concerns.

**Main Review:**

This paper builds a NAG-based MCMC sampler, which is named as Hessian-Free High-Resolution (HFHR($\alpha, \gamma$)) and generalizes the underdamped Langevin (also can be written as HFHR($0, \gamma$)) with an additional gradient drift and Brownian motion for the update of the position variable.

**Pros**: The most prominent feature of the extended algorithm yields much faster exponential convergence with **rate $O(L)$**  in continuous time than the standard underdamped Langevin algorithm  **(Cheng'18, rate $O(\frac{m}{L})$, Dalalyan'20, rate $O(\frac{m}{\sqrt{L}})$)**. Although the acceleration becomes less significant in numerical algorithms due to an increase of discretization error, some speed-up by a constant factor can be still achieved.

**Cons**:  the writing is not very clear, which affects the readability experience and made me hard to check the details. For example 1) the derivation of Formula (6) is slightly ad-hoc and more related work or motivations are suggested; 2) at the end of section B, some link on the definitions of notations may be suggested; 3) page 24, $H_t$ is derived based on Taylor expansion instead of saying nothing is better for presenting to the readers.

Remark: Theorem 5.1 is checked carefully; Theorem 5.2 is not; the logic seems to be reasonable.

Minor issues:

1. Experiments conducted on section 6.2 is non-convex and doesn't match the theory.

2. Missing part in related works: Replica-exchange (a.k.a parallel tempering):

[1]. Accelerating Nonconvex Learning via Replica Exchange Langevin Diffusion. ICLR'19

[2]. Non-convex Learning via Replica Exchange Stochastic Gradient MCMC. ICML'20


Cheng'18: Underdamped Langevin MCMC: A non-asymptotic analysis. COLT'18.

Dalalyan'20: A. S. Dalalyan and L. Riou-Durand. On sampling from a log-concave density using kinetic Langevin diffusions. Bernoulli



**Summary Of The Paper:**

The author proposed an accelerated-gradient-based MCMC method based on Nesterov's accelerated gradient (NAG). In continuous time, the algorithm is able to achieve a tremendous acceleration over the underdamped Langevin algorithm. Numerical schemes can propose a speed-up with a constant factor.


**Summary Of The Review:**

The NAG-based MCMC sampler extends the underdamped Langevin with a drift term and Brownian motion and is proved faster than the alternative. I would recommend this paper to be accepted.

---

> ### Author Response · Authors · 2021-11-23
> **Response to Reviewer r34U**
>
> We thank the reviewer for taking the time to review our paper and providing many valuable feedback. The positive verdict is also greatly appreciated. Below is our itemized reply to the concerns:
>
> > the writing is not very clear
>
> We revised and hopefully there is some improvement.
>
> > the derivation of Formula (6) is slightly ad-hoc and more related work or motivations are suggested
>
> Thanks very much for the opportunity of clarification. Formula (6) is our innovation and we're unaware of much related work. We took the optimization ODE (equation (5)) that we derived from NAG-SC, and identified the correct form of noise needed, so that adding these noise to equation (5) gives an SDE (eq.6) that has the desired target distribution as its limiting distribution. The correct form of noise is obtained from calculations based on Fokker-Planck PDE, and details can be found in Appendix B. We apologize for being unclear previously, but hopefully it can now be seen that formula (6) appears slightly unnaturally because it is an innovation. We would be really thankful if this could be appreciated.
>
> > at the end of section B, some link on the definitions of notations may be suggested
>
> Thank you. A link is added to Appendix B.
>
> >  page 24, explain that $H_t$ is derived based on Taylor expansion instead of saying nothing, so that it is better for presenting to the readers.
>
> Thanks for the helpful suggestion. This is now added.
>
> > Experiments conducted on section 6.2 is non-convex and doesn't match the theory
>
> Thank you very much for giving us an opportunity of explanation. The assessment is absolutely correct. Our intention was to have both experiments that satisfy the conditions of our theory (Sec.6.1) and that do not (Sec.6.2, which is nevertheless an important machine learning problem), so that, like in many great articles we learned from, the robustness and efficacy of the proposed method could be better convinced. Sorry that we should have mentioned this to avoid confusion. It is now included.
>
> > Missing part in related works: Replica-exchange (a.k.a parallel tempering)
>
> The references are greatly appreciated and added to the revision.
>
> ***
>
> We sincerely hope the reviewer could find the above responses reasonable. In case they are, a reconsideration would be deeply appreciated.

---

### Official Review · Reviewer_22pJ · 2021-11-09

**Correctness:** 3
**Technical Novelty And Significance:** 2
**Empirical Novelty And Significance:** 3
**Recommendation:** 6
**Confidence:** 4

**Main Review:**

This paper introduces a Hessian-free high-resolution Nesterov’s Acceleration approach which proves faster convergence than both Underdamped and Overdamped Langevin Dynamics. In addition, it also proves that the acceleration cannot be achieved by time-rescaling. Experiments results support the argument in optimization acceleration as well as the theoretical results.

Some questions the reviewer minds:

-The theorem 5.1 depends on the assumption on \gamma and \alpha. If such an assumption often holds in real applications.

-It is will be better to demonstrate the acceleration superiority in with a more complex network and a large-scale dataset.

**Summary Of The Paper:**

This stuyd focused on developing a Hessian-free high-resolution Nesterov’s Acceleration approach which proves converging faster and being  non-trivial (not a time-rescaling trick).

**Summary Of The Review:**

The reviewer thinks this paper contributes a new and effective gradient based optimization solver. Overall, the reviewer would like to recommend acceptance.

---

> ### Author Response · Authors · 2021-11-23
> **Response to Reviewer 22pJ**
>
> We thank the reviewer for taking the time to review our paper and providing many valuable feedback. The positive verdict is also greatly appreciated. Below is our itemized reply to the concerns:
>
> > Can the assumption on $\gamma$ and $\alpha$ needed by our theoretical result be satisfied in practice?
>
> This is a very important point and we appreciate the opportunity of clarification. $\gamma$ and $\alpha$ are hyperparameters chosen by the user, and indeed there exist choices in practice that satisfy the conditions of our theory. In this sense, our theory is not just a mathematical game but helpful for choosing hyperparameters and thus optimizing the algorithm's performance.
>
> To technically demonstrate the above statement, i.e., that one can always find $\gamma$ and $\alpha$ that satisfy the theoretical conditions, the following is an example: for any given $\alpha \ge 0$, we can pick any $\gamma \ge \gamma_+$ where $\gamma_+$ is the positive root of the equation $\gamma^2 - m\alpha \gamma - (L+m) = 0$, and such $\gamma$ will always satisfy the assumption in Thm.5.1. Many other constructive ways exist too.
>
> > it will be better to demonstrate the acceleration superiority with a more complex network and a large-scale dataset
>
> We completely agree. In cases when our theoretical contribution is under-appreciated, we hope empirical evidence could cast doubts away. One issue with larger-scale dataset though is, this work did not consider stochastic gradient, because the full-batch-gradient case is theoretically already rather nontrivial. Since our task is sampling rather than optimization, the additional variance introduced by stochastic gradient has to be correctly offset, and convergence speed will be very different from the full-batch-gradient case, for both ULD and HFHR based algorithms. Therefore, we only listed SG-HFHR as a future direction in the Conclusion and Discussion section. Interestingly, some of the latest progress suggested not to use stochastic gradient but full-batch-gradient for training Bayesian neural networks (BNN) [Izmailov, Vikram, Hoffman, and Wilson. ICML 2021]. Our BNN (Sec 6.2) was trained using full-batch, however only on small data set, because we do not have the computational resources to do full-batch with large-scale dataset. One thing with BNN is, multiple independent realizations have to be run if one wants to have a trustworthy quantification of its performance (we used 10,000 realizations in Sec 6.2); thus our computational capability is strained.
>
> ***
>
> We sincerely hope the reviewer could find the above responses reasonable. In case they are, a reconsideration would be deeply appreciated.

---

### Official Review · Reviewer_4H3p · 2021-11-10

**Correctness:** 2
**Technical Novelty And Significance:** 3
**Empirical Novelty And Significance:** 2
**Recommendation:** 3
**Confidence:** 4

**Main Review:**

This paper has solid theoretical analysis.
However the author missed an important comparison with randomized midpoint method (Shen & Lee, 2019).
I think the author proved an iteration complexity that is worse to the method in that paper. This invalidates author's claim on constant factor speed-up over underdamped Langevin dynamics.

Specific Comments:
1) In Assumption A1, $m \lVert y-x \rVert \leq \lVert \nabla f(y)-\nabla f(x) \rVert \leq L \lVert y-x \rVert$ seems insufficient to establish m-stronly-convex and L-smooth.

2) Does this paper consider "log-concave" case? In the introduction, author said " will consider the setup of log-concave / log-strongly- concave target distributions". However, it seems Assumption A1 require $m > 0$.

3) On page 9, author wrote "(strongly) log-concave assumptions required in Theorem 5.2", why there is a bracket? Does it imply that Theorem 5.2 could be applied in log-concave case?

4) In Remark 5.5, at the last line, the author claimed "steps needed by ULD can be halved by HFHR". Which algorithm is the author referring to for "ULD". Is it algorithm in (Cheng et al. 2018) or HFHR with $\alpha=0$? It seems these two algorithms are actually different.

5) In section 6.1, author proposed to use error of mean as a surrogate of 2-Wasserstein distance. Although error of mean is a lower bound, how tight it is? In figure 2, the mean value at $\alpha=0.5$ in the Figure 2 seems close to 1, which means HFHR achieves $\varepsilon\leq0.1$ accuracy in just one iteration. Clearly one iteration is not enough for burn-in. Does it means error of mean is a bad indicator of mixing?

6) In figure 1, could the author also add lines for ULD and randomized midpoint method?

**Summary Of The Paper:**

This paper introduces Hessian-Free High-Resolution SDE inspired by Accelerated Gradient (NAG). The author shows that continuous solution achieves an acceleration over the underdamped Langevin. A discrete algorithm also has speed-up with a constant factor.

**Summary Of The Review:**

I thinks this paper could be improved substantially, but currently would like to recommend reject.

---

> ### Author Response · Authors · 2021-11-23
> **Response to Reviewer 4H3p (part 2 of 2)**
>
> > Is the log-concave case also considered?
>
> Thank you very much for catching this error. Sorry we had a very misleading typo. This paper only considers the log-strongly-concave setup. This error has been corrected in the revised version.
>
> > What discretization of ULD was the one mentioned in Remark 5.5?
>
> Thank you very much for identifying this source of confusion. The one in Remark 5.5 is HFHR discretization with $\alpha=0$. We were trying to compare apple with apple, i.e., using similiar discretizations, so that the advantage of a HFHR correction (i.e., nonzero $\alpha$) can be seen in an unbiased way. We now realize this creates confusion because we also implemented another ULD discretization in the experiment section, namely the popular 1st-order KLMC [Dalalyan & Riou-Durand, 2020]. In the revised version, we carefully distinguish different discretizations to avoid the confusion, and hopefully this resolves the issue.
>
> > HFHR converges too fast in figure 2. Does it mean error of mean is a bad indicator of mixing?
>
> Thank you very much for making this interesting note. We don't think that is the case. It may indeed be uncommon for an MCMC algorithm to mix within two iterations; however, it is not impossible. In Appendix E Demonstration 1 we proved that for isotropic Gaussian, HFHR with appropriate $\alpha$ converges in 1 step, whereas without $\alpha$ the convergence is only asymptotic. This is analogous to Newton versus gradient descent in optimization, whereas the former needs only 1 step for quadratic loss while the latter technically speaking still needs infinite many steps.
>
> The potential in figure 2 is much more nonlinear than that for Gaussian, and we don't have convergence in 1 step any more, but indeed the convergence is still fast, about 2x faster if the optimal $\alpha$ performance is compared to the $\alpha=0$ case, which is consistent with Remark 5.5.
>
> Note also that $\epsilon=0.1$ here, which is not too small. We didn't have the computational resources to do $\epsilon=0.01$ because fig.2 is obtained over large ensemble simulations, and while the number of steps needed to reach $\epsilon=0.01$ accuracy is not an issue, in order to ensure an achieved 10x accuracy is trustworthy, 100x realizations are needed for empirical averages.
>
> We really hope the fast convergence could be seen as an evidence of the advantage of HFHR, not a drawback.
>
> > add numerical experiments for randomized midpoint
>
> We performed additional experiments to compare randomized midpoint discretization of ULD with randomized midpoint discretization of HFHR dynamics. Results are added to the Appendix F of the revision. An acceleration enabled by HFHR is also clearly seen.
>
> ***
>
> We hope the above responses and our revision clarified the concerns, and sincerely plead for a reconsideration.

---

> ### Author Response · Authors · 2021-11-23
> **Response to Reviewer 4H3p (part 1 of 2)**
>
> We thank the reviewer for taking the time to review our paper and providing many valuable feedback. Below is our itemized reply to the concerns:
>
> > iteration complexity is worse than randomized midpoint
>
> Thank you very much for raising this concern. This is very important, and perhaps also a major factor of the reviewer's assessment, but fortunately we feel it can be addressed. In short, once we compare apple with apple and orange with orange, this concern hopefully resolves. Here are detailed explanations:
>
> (1) Our main objective was to suggest that the extra HFHR-correction terms, namely $- \alpha \nabla f(q_t) dt + \sqrt{2\alpha} dW_t$, are advantageous. For this, we quantified the acceleration they enabled both in continuous time and in a discretization. The purpose of our time-discretized analysis is mainly to show that the acceleration enabled by HFHR is a genuine one which persists after discretrization, unlike illusive accelerations created by trivialities such as time-rescaling, which will disappear after discretization. For that purpose, we had to choose some discretization, and a most fair choice would be to use similar discretizations for both HFHR and ULD, so that there is minimized distraction from the difference between discretization accuracies. Therefore, we chose a 1st-order scheme, because we thought 1st-order discretizations such as LMC/unadjusted-Langevin-algorithm and KLMC are still popular in the community, so we hoped to illustrate the acceleration that HFHR can bring, in a 1st-order setup. In Sec.1 of the original submission, we wrote "it was known that high-order discretizations can improve statistical accuracy and even the speed of convergence, although such improvements often come with more computations per iteration. The discretization considered here is just a simple first-order scheme that uses one (full-)gradient evaluation per step". Note RMA uses 2 gradient evaluations per step, so there are still situations in which simple 1st-order schemes are useful.
>
> (2) Meanwhile, we completely agree that the iteration complexity of a 1st-order discretization may not be as good as that of a higher-order discretization. When compared to Randomized Midpoint Algorithm (RMA) of ULD (i.e. HFHR dynamics with $\alpha=0$), which is a high-order discretization, the iteration complexity of our 1st-order HFHR discretization is indeed worse. However, we hope to emphasize that this comparison can be rather misleading, because it is between **a low-order discretization of dynamics A** and **a high-order discretization of dynamics B**, which per se is not a very good indicator of if dynamics A is better or worse than B. That is why we instead compared a low-order discretization of dynamics A with two **same**-order discretizations of dynamics B (one being KLMC and the other being the $\alpha=0$ version of the HFHR algorithm in the original submission). We now realized that these were not expressed clearly enough in the original submission, and have revised accordingly.
>
> (3) Nevertheless, thanks to the reviewer, we also investigated the effect of HFHR corrections on RMA, as it is a great high-order discretization. More precisely, we added another comparison between HFHR dynamics and ULD dynamics based on RMA discretization, in Appendix F (i.e., **a high-order discretization of dynamics A** v.s. **a same-order discretization of dynamics B**). There we provided both a description of how we generalized RMA to discretize HFHR dynamics, and empirical results that clearly show that HFHR corrections again lead to extra acceleration over ULD, when both are discretized by the high-order scheme of RMA. This makes the picture more complete, as now we have both low- and high-order discretizations, in addition to continuous time results, all of which confirm the nontrivial acceleration of HFHR.
>
> > the assumption $m\|y-x\| \leq \|\nabla f(\boldsymbol{y}) - \nabla f(\boldsymbol{x})\| \leq L\|\boldsymbol{y} - \boldsymbol{x}\|$ seems insufficient to establish m-strong-convexity and L-smoothness.
>
> Thanks for the very helpful comment. The $\|\nabla f(\boldsymbol{y}) - \nabla f(\boldsymbol{x})\| \leq L\|\boldsymbol{y} - \boldsymbol{x}\|$ part is a common  definition of $L$-smoothness, but indeed we could have used a more standard definition for strong convexity rather than $m\|y-x\| \leq \|\nabla f(\boldsymbol{y}) - \nabla f(\boldsymbol{x})\|$. Regarding what we used, under the assumption that $f\in\mathcal{C}^2$, sending $y\to x$ shows $m$ is a lower bound of the Hessian everywhere, and hence $f$ is m-strongly-convex. Nevertheless, if $f$ is only 1-time differentible, it is indeed possible to see difference from the standard definition of $f(y)\geq f(x)+\langle \nabla f(x), y-x\rangle + m/2 \|y-x\|^2, \forall x,y$. Therefore, we replaced our nonstandard definition by the standard one in the revision. This did not affect any of our results.

---

### Official Review · Reviewer_awRv · 2021-11-10

**Correctness:** 3
**Technical Novelty And Significance:** 3
**Empirical Novelty And Significance:** 3
**Recommendation:** 5
**Confidence:** 4

**Main Review:**

The paper is very technical and not easy to follow. The main contribution of the paper is to provide an accelerated first order diffusion process in continuous time, yielding a novel sampling method after discretization. I have a few questions/concerns to be clarified regarding the claims in the paper:

1 Acceleration in continuous time
Comparing to the rate obtained in Dalalyan 2020, the proposed algorithm improves by a factor of $\kappa$. However, I am confused as no improvement is achieved for the function $f(x,y) = mx^2+Ly^2$, this seems contradictory to the claim, do I misunderstand anything? Is it related to the time scaling?

2 Discretization
When discretizing, the paper applies a second-order symmetric composition. What is the motivation behind this step? What would be the benefit compared to applying forward Euler discretization?

3 Acceleration in discrete time
Unless we show that the constant term is improved by a non-trivial factor (L, or $\kappa$ for example), I wouldn't call it an acceleration as those are only upper bounds in the analysis.

4 ULD = HFHR(0,$\gamma$)
I understand that the continuous ODE are equivalent by taking $\alpha = 0$, is the discretized algorithm still follows the same equivalence? (for instance, different discretization method may be applied)

5 Comments on Experiments
a. In figure 2, it seems surprising to me that only 2 iterations is enough to reach $\epsilon$ closeness, which may suggest the problem is too easy
b. In Figure 3, why does some figure only has $\alpha=0.1$ and the others has $\alpha = 0.1, 0.5, 1$


**Summary Of The Paper:**

The paper proposes an accelerated MCMC method for sampling, motivated by Nesterov's Accelerated Gradient (NAG) method. Starting from the high resolution ODE of NAG obtained in Shi et al, the paper applies a two stage mechanism to remove the Hessian-dependency. The obtained first order ODE system serves as the backbone of the proposed diffusion process, Hessian-Free High-Resolution (HFHR) dynamics. Discretization the HFHR dynamics leads to the proposed sampling method. Theoretical convergence are provided for both the continuous and the discretized variant, showing acceleration to existing underdamped Langevin method (ULD).

**Summary Of The Review:**

Overall, I believe the theoretical result of the paper has merit but the presentation need to be improved

---

> ### Author Response · Authors · 2021-11-23
> **Response to Reviewer awRv (part 2 of 2)**
>
> > In Figure 3, why does some figure only has $\alpha=0.1$ and the others has $\alpha=0.1, 0.5, 1$.
>
> Thanks for asking. We realize that our explanation, "Cases where $\alpha$ is too large for numerical stability are not drawn", was buried in the text. As discussed in the theoretical part, if $\alpha$ is too large, $h$ does need to be smaller for the discretization to still work, and thus when $h$ is larger, less values of $\alpha$ work. Note, however, that the role of $\alpha$ is nevertheless not a simple time rescaling, and optimal values do not satisfy $\alpha h =$constant; see e.g., Remark 5.5 for choices of $\alpha$ based on bounds, and Appendix E for those based on tight calculations.
>
> ***
>
> We hope the above responses and our revision clarified the concerns, and sincerely plead for a reconsideration.

---

> ### Author Response · Authors · 2021-11-23
> **Response to Reviewer awRv (part 1 of 2)**
>
> We thank the reviewer for taking the time to review our paper and providing many valuable feedback. Below is our itemized reply to the concerns:
>
> > no improvement is achieved for the function $f(x,y)=mx^2+Ly^2$, this seems contradictory to the claim
>
> Thank you for an opportunity of clarification. In fact, we apologize for the misunderstanding, but what we meant is, for this $f$ the existing ULD (Underdamped Langevin Dynamics) result gives $\sqrt{m}/(\sqrt{\kappa}+\sqrt{\kappa-1})$ and our analysis methodology (which is not the same as used in the existing result) gives $\sqrt{m}/(2\sqrt{\kappa})$, also for **ULD** (i.e. HFHR with $\alpha=0$). This serves only as a sanity check. And then we demonstrated the addition acceleration created by $\alpha\neq 0$, both at the continuous and the discrete level, afterwards.
>
> > benefit of a symmetric composition over forward Euler discretization?
>
> This question is very much appreciated as it makes us realize that we under-explained. For deterministic differential equations, a truly symmetric composition (i.e. Strang splitting) would result in a 2nd-order global integration error, as opposed to the 1st-order global integration error of forward Euler. Our setup is SDE instead and we're interested in sampling error instead of integration error, but nevertheless under reasonable assumptions a similar improvement of accuracy can be proved for a true Strang scheme. Meanwhile, the version of discretization described in Sec.5.2 is an approximate Strang scheme and its order drops back to 1, same $\mathcal{O}(h)$ error as Euler-Maruyama, but the constant in the big $\mathcal{O}$ bound is actually improved (Appendix D.5-D.7 have more details about the above statements). Importantly, this gain is almost for free because the number of gradient evaluation per step remains to be 1 in our case, same as that of Euler-Maruyama, and hence we chose it as a simple 1st-order discretization. Higher-order discretizations exist too and more details about them is also discussed in this rebuttal (e.g., Appendix F).
>
> > Unless we show that the constant term is improved by a non-trivial factor, I wouldn't call it an acceleration as those are only upper bounds in the analysis.
>
> We completely agree. Results in Appendix E, for example, are not based on comparing bounds but tight theoretical calculations, and they gave, for example, Remark 5.6, which is on improved $\kappa$ dependence of the convergence rate. The genuine acceleration was also numerically validated; see e.g., Fig.2 in the original submission, and the newly added Fig.5 in the revised submission.
>
> > ULD = HFHR(0,$\gamma$) at the continuous level; is the discretized algorithm still follows the same equivalence? (for instance, different discretization method may be applied)
>
> If we understood the question correctly, yes, given any HFHR discretization, it with $\alpha=0$ gives a ULD discretization. An ULD discretization can correspond to multiple HFHR discretizations though. Neither HFHR nor ULD has a unique discretization.
>
> > In figure 2, it seems surprising to me that only 2 iterations are enough to reach $\epsilon$ closeness, which may suggest the problem is too easy.
>
> Thanks very much for this interesting note. We chose this potential because it is highly-nonlinear and resemblant to expressions we encounter in game theoretic problems as well as Bayesian logistic regression. Indeed, with good $\alpha$ values HFHR is able to use very few iterations, but without $\alpha$ (degenerated to ULD) more iterations are needed, so the problem is not necessarily too easy. In fact, in Appendix E Demonstration 1 we showed that for isotropic Gaussian, HFHR with appropriate $\alpha$ converges in 1 step, whereas without $\alpha$ the convergence is only asymptotic. This is a little like Newton versus gradient descent in optimization, whereas the former needs only 1 step for quadratic loss while the latter technically speaking still needs infinite many steps.
>
> Note also that $\epsilon=0.1$ here, which is not too small. We didn't have the computational resources to do $\epsilon=0.01$ because Fig.2 is obtained over large ensemble simulations, and while the number of steps needed to reach $\epsilon=0.01$ accuracy is not an issue, in order to achieve a trustworthy 10x accuracy, 100x realizations are needed for empirical averages.
>
> We really hope this could be seen as an evidence of the advantage created by HFHR, instead of a drawback.
>
> Nevertheless, we agree that this potential is a toy when compared to practical learning problems, and we did have a 2nd example (Bayesian neural network), for which convergence is slow but still accelerated by HFHR. We did Example 1 because we wanted to experimentally validate our theoretical predictions via large scale ensemble tests (e.g., Fig.1 and 2); we do not have the computational resource to afford these for Example 2 (Bayesian neural network) though.

---

### Comment · Reviewer_xkFf · 2021-11-29
**Discussion on Reproducibility Issue**

I have not been able to reproduce Figure 1a. I would be very grateful if someone can confirm my result or correct my mistake.

---

### My attempt

1. Download code from Supplementary Material.
2. Go to folder `code/verifyDependenceOnStepSize`.
3. Run `mkdir results`.
4. Run `for j in {1..100}; do printf '%d\n' $j; done | parallel python verify_dependence_on_h_nonlinear_potential.py --seed {}`.
5. Extract result by a python script.

Cost:
Step 4 takes about 212 minutes for each process. Different processes can be paralleled, so the experiment can be reproduced within 4 hours with 128-Thread processor.

---

Step 5. Extract result by a python script
```
import os
import numpy as np

hs = sorted(list(set(f.split("|")[1].split("=")[1] for f in os.listdir("results"))))

means = [
    np.mean(
        [
            np.load(f"results/sample_mean|h={h}|d=2|n_samples=1000000|part={i}.npy")[-1]
            for i in range(1, 101)
        ],
        0,
    )
    for h in hs
]
groundtruth = -np.ones(2) / 2
errors = np.linalg.norm(means - groundtruth, axis=1)

fmt = "{:<10} {:<20}"
print(fmt.format("step size", "error of mean"))
print(fmt.format("x-value", "y-value"))
for h, error in zip(hs, errors):
    print(fmt.format(h, error))

# plot
import matplotlib.pyplot as plt

plt.plot(np.array([float(h) for h in hs]), errors)
plt.xlabel("step size")
plt.ylabel("error of mean")
plt.xscale("log")
plt.yscale("log")
plt.show()
```

---
### Result
```
step size  error of mean
x-value    y-value
0.0078125  6.155751193279832e-05
0.015625   3.24395818680134e-05
0.03125    0.00012469951134488403
0.0625     0.00011271019977054319
0.125      0.00012402092738382037
0.25       0.00012676214501006623
0.5        0.00013291926615112697
1          0.0003001348227226792
2          6.805572349740432e-05
```

---
### Explain the method

Step 4 is due to authors' comment:

> Each run of the script gave one portion of data and then we aggregated and averaged the results to produce Fig 1a.

> we used `n_samples=int(1e6)` for each single job, but run the script with 100 different seeds. So altogether `1e8` independent samples are used.

I don't know seeds used by authors, thus I arbitrarily choose 1 to 100.

Explanation of script in step 5:
Retrieve last column for each npy file, because that is for last iteration.
Calculate mean of these 100 points, because that is mean of total `1e8` samples.
Calculate the 2-norm difference to ground truth $[-0.5,-0.5]$, because that is "error of mean" reported as y-axis in Figure 1a.

---
### Explain the result

My plot is not even similar to Figure 1a, which has nice alignment with $O(h)$.

At small step size regime, my y-axis value is much larger than data in Figure 1a.
At large step size regime, my y-axis value is much smaller than data in Figure 1a.
In my result, there is not significant correlation between estimated "error of mean" and step size. However in Figure 1a, the experiment fit a power law precisely.

---

### Decision · Program_Chairs · 2022-01-20

**Decision:**

Reject

**Comment:**

The paper considers the high resolution continuous limit of Nesterov's Accelerated Gradient (NAG) algorithm and its connections to sampling (MCMC methods). The paper develops a Hessian-Free High Resolution (HFHR) ODE and injects noise into it to obtain an accelerated sampling algorithm. Further, the paper provides a discrete-time variant of the algorithm by appropriately discretizing HFHR using simple discretization schemes. For strongly log-concave potential functions (log-densities), the paper proves convergence of the order $\tilde{O}(\sqrt{d}/\epsilon)$ in Wasserstein-2 distance. In the asymptotic sense, the result matches the convergence of the underdamped Langevin algorithm; however, the paper argues that the constants in the proposed algorithm are smaller and empirically shows that the proposed algorithm is faster in practice. The main contributions of the paper are theoretical; however, the theoretical results are supplemented by numerical experiments.

Overall, the reviewers found the contributions interesting and the theoretical contributions of the paper technically sound. The main concerns that were not completely addressed were related to the presentation of the results and reproducibility of some of the numerical experiments. While both seem minor and possible to address, ultimately there was not enough support to recommend acceptance. However, the paper is solid and merits acceptance after suitable revisions. Thus, the authors are encouraged to revise the paper and resubmit it to one of the conferences in the equivalence class of ICLR.